# Skeletal stem and progenitor cells maintain cranial suture patency and prevent craniosynostosis

Siddharth Menon[1,2,3,4], Ankit Salhotra [1,2,3], Siny Shailendra[1,2,3], Ruth Tevlin[1,2,3], Ryan C. Ransom[1,2,3], Michael Januszyk[1,2,3], Charles K. F. Chan [1,2,3,4], Björn Behr[5], Derrick C. Wan[1,2,3], Michael T. Longaker [1,2,3,4✉] & Natalina Quarto [1,2,3,6✉]

Cranial sutures are major growth centers for the calvarial vault, and their premature fusion leads to a pathologic condition called craniosynostosis. This study investigates whether skeletal stem/progenitor cells are resident in the cranial sutures. Prospective isolation by FACS identifies this population with a significant difference in spatio-temporal representation between fusing versus patent sutures. Transcriptomic analysis highlights a distinct signature in cells derived from the physiological closing PF suture, and scRNA sequencing identifies transcriptional heterogeneity among sutures. Wnt-signaling activation increases skeletal stem/progenitor cells in sutures, whereas its inhibition decreases. Crossing $Axin2^{LacZ/+}$ mouse, endowing enhanced Wnt activation, to a $Twist1^{+/−}$ mouse model of coronal craniosynostosis enriches skeletal stem/progenitor cells in sutures restoring patency. Co-transplantation of these cells with Wnt3a prevents resynostosis following suturectomy in $Twist1^{+/−}$ mice. Our study reveals that decrease and/or imbalance of skeletal stem/progenitor cells representation within sutures may underlie craniosynostosis. These findings have translational implications toward therapeutic approaches for craniosynostosis.

[1] Hagey Laboratory for Pediatric Regenerative Medicine, Stanford University School of Medicine, Stanford, CA, USA. [2] Division of Plastic and Reconstructive Surgery, Stanford University School of Medicine, Stanford, CA, USA. [3] Department of Surgery, Stanford University School of Medicine, Stanford, CA, USA. [4] Institute for Stem Cell Biology and Regenerative Medicine, Stanford University School of Medicine, Stanford, CA, USA. [5] Department of Plastic Surgery, University Hospital Bergmannsheil Bochum, Bochum, Germany. [6] Dipartimento di Scienze Biomediche Avanzate, Universita' degli Studi di Napoli Federico II, Napoli, Italy. ✉email: longaker@stanford.edu; nquarto@stanford.edu

Cranial sutures are fibrous joints that comprise two approaching osteogenic bone fronts separated by intervening proliferative mesenchymal tissue. They represent the major sites of cranial morphogenesis and grow in close coordination with the rapidly developing brain[1–5]. Their anatomy is remarkably well conserved in both mammalian and non-mammalian species[6–9]. Murine and human cranial sutures are similar: both have an anterior-frontal suture (AF), located between the paired frontal nasal bones; a posterior-frontal suture (PF) (metopic, in humans), located between the paired frontal bones; a coronal suture (COR), located between the frontal and parietal bones; a sagittal suture (SAG), between the paired parietal bones; and a lamboid suture (LAM), between the occipital and parietal bones. The PF suture fuses in early life, through endochondral ossification, while the SAG, COR, and LAM remain patent through adulthood[10].

A delicate balance between cell proliferation, differentiation, migration, and apoptosis regulates the osteogenic fronts at the cranial sutures, ensuring a steady equilibrium of growth and separation[2,11,12]. Previous studies have indicated that cranial suture closure versus patency is governed by canonical Wnt (cWnt) signaling[13]. The molecular machinery coordinating cell proliferation with osteoblast differentiation must be under strict control: prolonged proliferation of the suture mesenchyme or delayed osteoblast differentiation results in pathological suture expansion, whereas insufficient proliferation or accelerated differentiation results in premature cranial suture fusion, known as craniosynostosis[11].

Craniosynostosis affects approximately one in 2500 live births, resulting in significant clinical sequelae including major craniofacial deformities[14–18]. Craniosynostosis can be associated with an underlying genetic abnormality (syndromic) or, more commonly, non-syndromic[19–26]. Unveiling the factors that give rise to non-syndromic craniosynostosis is therefore of relevance in determining the etiology of premature suture fusion.

Recently, the suture mesenchyme has been postulated to act as a niche for stem cells critical in suture development, cranial growth, and injury repair[27–30]. Similar to the growth plates of the long bone, cranial sutures are the sites of growth for the cranial vault. Fueled by our prior isolation of skeletal stem cells and progenitors from long bone[31,32], long bone fractures[33,34], and mandibles[35,36], we aimed to examine cranial suture biology through the lens of these cells.

We first confirm that skeletal stem/progenitor cells (referred to as CD51+;CD200+ cells) are resident in cranial sutures and validated their multipotency and ability to self-renew. CD51+; CD200+ cells are then prospectively isolated from physiologically fusing and patent sutures, in addition to cranial sutures harvested from animal models of syndromic and non-syndromic craniosynostosis. We identify that aberrancies in CD51+;CD200+ cell equilibrium may underlie both physiologic suture closure and pathologic craniosynostosis. Furthermore, we demonstrate that cWnt activation increases CD51+;CD200+ cell frequency, prevents suture fusion, and ultimately rescues the craniosynostosis phenotype. These findings suggest that CD51+;CD200+ skeletal stem/progenitor cells and Wnt3a may provide a combined cellular/molecular intervention to treat or even prevent craniosynostosis.

## Results

### In vivo profiling of CD51+;CD200+ cells in cranial sutures.
We initiated our study by confirming the presence of skeletal stem/ progenitor cells (referred to as CD51+;C200+ cells), defined by the following immunophenotype: **CD51+**, **CD200+**, CD45−,

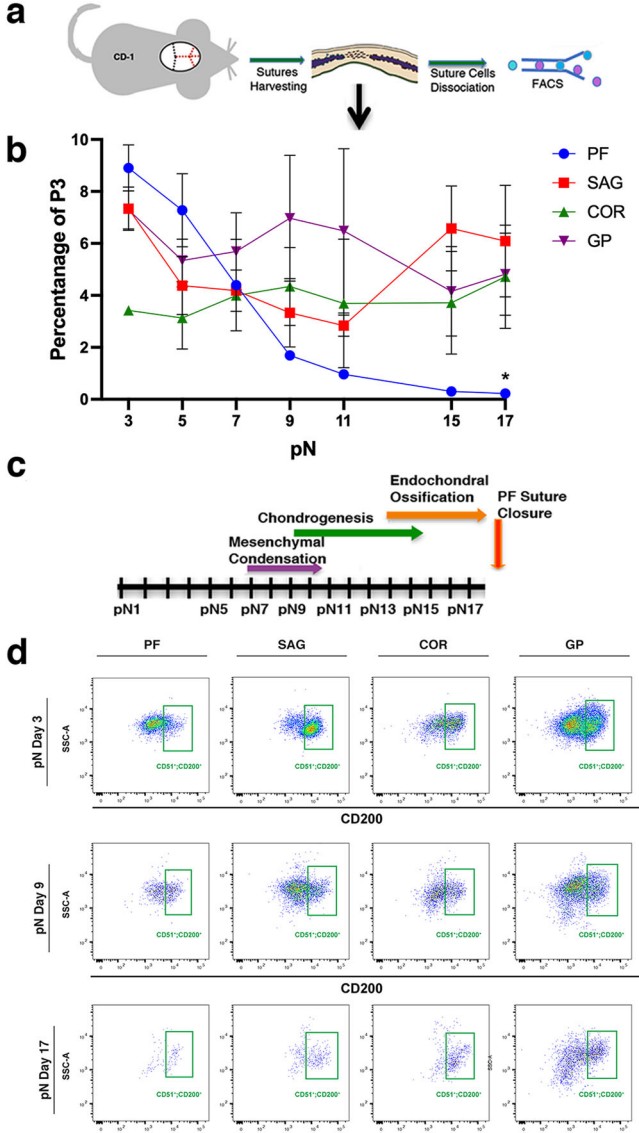

**Fig. 1 Temporal FACS profiling of CD51+;CD200+ cells in patent and fusing cranial sutures. a** Outline of calvarial suture harvesting and processing. **b** Time-course profiling of skeletal stem/progenitor cells isolated in vivo by flow cytometric analysis from freshly harvested posterior frontal (PF), sagittal (SAG), and coronal (COR) sutures. Tibial growth plate (GP) was used as reference. Values are given as a percentage of the P3 population, representing viable (propidium iodide-), non-hematopoietic (CD45−) cells isolated from the sutures. The fusing PF suture shows a dramatic decrease in the percentage of skeletal stem/progenitor cells between day pN7 and pN9, corresponding to the initiation of PF suture closure[10]. Conversely, the patent SAG and COR sutures and GP maintain a sustained percentage of these cells over time. Data are represented as means ± SEM; PF vs SAG $P = 0.002$, PF vs COR $P = 0.0147$, PF vs GP $P = 0.0179$, *$P ≤ 0.05$ unpaired, two-tailed student t-test were performed. $n = 60$ animals/timepoint, experiments were repeated 6 independent times for pN3, 4 for pN17 and 2 independent times for pN5, 7, 9, 11, and 15. **c** Diagram illustrating the timing of the endochondral differentiation through which the PF suture closes[10]. **d** Representative FACS plots of skeletal stem/progenitor (CD51+;CD200+) cells isolated in vivo from the PF, SAG, and COR sutures and GP at pN3, pN9, and pN17. CD51+;CD200+ cells decrease as a function of time in the physiologically fusing PF suture. Refer to Supplementary Fig. 1 for complete FACS gating and isolation strategy. Source data are provided as a Source data file.

Ter119[−], Tie2[−], Thy1.1[−], Thy1.2[−], 6C3[−], CD105[−31,32] in the mouse posterior frontal (PF), sagittal (SAG), and coronal (COR) sutures at postnatal day 3 (pN3) (Fig. 1a–d, Supplementary Fig. 1a–c). Following validation of their multi-potency and self-renewal capacity (Supplementary Fig. 2a–d), we then performed an in vivo time course profile of these cells in cranial sutures using FACS analysis (Fig. 1a–d). CD51[+];CD200[+] cells were analyzed in the PF, SAG, and COR sutures between pN3 and pN17 ($n = 60$ mice/time-point), a time frame spanning the period of physiological PF suture closure[10] (Fig. 1c). Cell numbers were normalized to the cells defined by the following immunopheno-type, [Pi[−]/CD45[−]], hereafter referred to as the P3 population, in our FACS gating scheme (Supplementary Fig. 1a–c). CD51[+]; CD200[+] cells from long bone growth plates (GP) were used as a reference[31,32].

The PF suture contained a higher percentage of CD51[+];CD200[+] cells at early time-points (pN3-5), when this suture is patent, followed by a sharp decrease in CD51[+];CD200[+] cells starting at pN7 (Fig. 1b). pN7 is prior to the onset of chondrogenesis which precedes suture closure (Fig. 1c)[10]. By pN17, when the PF suture is fused[10], CD51[+];CD200[+] cells represented only a small percentage of the P3 population. Conversely, CD51[+];CD200[+] cell frequency from the patent COR and SAG sutures were maintained throughout the time-course whereas their frequency in GP was relatively higher (Fig. 1b, d). Our data reveal significant differences in CD51[+];CD200[+] cell representation between patent versus fusing cranial sutures—specifically, the number of these cells in the PF suture decreases as a function of time in advance of physiological suture fusion.

**Transcriptomic signature of suture-CD51[+];CD200[+] cells.** Next, we compared the transcriptomes of CD51[+];CD200[+] cells resident in fusing and patent sutures. We performed bulk RNA-seq profiling of CD51[+];CD200[+] cells harvested from the PF, SAG, and COR sutures (hereafter referred to as PF-CD51[+]; CD200[+] cells, SAG-CD51[+];CD200[+] cells and COR- CD51[+]; CD200[+] cells respectively) of pN3 mice, again using GP-isolated CD51[+];CD200[+] cells (GP-CD51[+];CD200[+] cells) as controls ($n = 60$ mice/time point). pN3 was chosen based on two features shared at this time point by all three sutures: first, their patency; second, their comparable degree of endogenous active cWnt signaling (Supplementary Fig. 3a)[10,13].

Our analysis revealed significant transcriptomic differences among skeletal stem/progenitor cells from the different sutures. We identified 171 (out of 24,015 mouse mm9) genes whose expression levels were significantly different with a fold change greater than 3 between the different sutures (Fig. 2a). Hierarchical transcriptional analysis revealed a distinct clustering of transcripts that were up-or downregulated in CD51[+];CD200[+] cells from each suture. We identified a uniquely downregulated cluster of genes in PF-CD51[+];CD200[+] cells which are associated with craniosynostosis. Interestingly, the PF suture is the only suture that physiologically fuses during early life. Among these genes were *Cdc45*, *Huwe1*, *Efnb*, *Ahdc1*, *Dusp6*, *Jag1*, and *Ezh2*[37–42], while *Ptpn11*, a gene associated with gain-of-function mutations resulting in craniosynostosis[43], was upregulated in PF-CD51[+]; CD200[+] cells exclusively (Fig. 2b and Supplementary Fig. 3b, c). The expression profile of these genes was validated by RT-PCR (Supplementary Fig. 3b).

To further characterize these differences in transcriptional programming, we performed single-cell RNA-seq analysis on PF-CD51[+];CD200[+] cells, SAG-CD51[+];CD200[+] cells, and COR-CD51[+];CD200[+] cells isolated at pN3 (Fig. 2c). Single cell data for each of the sutures were pooled, and blinded partitional analysis identified five transcriptionally-defined clusters (Fig. 2d).

Interestingly, PF-CD51[+];CD200[+] cells and COR-CD51[+];CD200[+] cells exhibited similar distributions and together comprised the majority of clusters 0, 2, and 3, while the SAG-CD51[+];CD200[+] cells transcriptional profiles appeared more distinct (cluster 1 and 4) (Fig. 2e, f). Downstream analyses of cluster characteristics provided further insights into what may represent transcriptional meta-states among suture-derived CD51[+];CD200[+] cells, such as those with comparatively pro-osteogenic versus chondrogenic expression patterns. (Fig. 2f–e, Supplementary Fig. 4a–e, and Supplementary Fig. 5a–d). Moreover, these transcriptional programs appeared distinct from those of the periosteal stem cell (PSC) population previously identified[44] (Supplementary Fig. 6a) and comparably pure with our earlier CD51[+];CD200[+] cell population (Supplementary Fig. 6b)[45,46].

**Imbalance of CD51[+];CD200[+] cells marks craniosynostosis.** The different trends of resident CD51[+];CD200[+] cells between physiologically fusing versus patent sutures set the stage for investigating these cells in syndromic and non-syndromic craniosynostosis. Saethre-Chotzen syndrome, defined by *TWIST1* loss-of-function mutation, is the second most prevalent form of syndromic COR craniosynostosis[21,47]. *Twist1*[+/−] mice represent an established model to study this genetic syndrome[48,49], displaying premature COR fusion (Fig. 3a). We FACS-profiled CD51[+];CD200[+] cells to determine their representation in *Twist1* [+/−] COR sutures relative to COR sutures from wild-type mice. This analysis revealed a significant decrease of these cells in craniosynostotic *Twist1*[+/−] as compared to wild-type COR sutures (Fig. 3b).

The majority of human craniosynostoses are non-syndromic (85%); of these, synostoses of the SAG suture are the most common (40–55%). Mutations in the inhibitory *SMAD6* gene have been reported to lead to SAG craniosynostosis[50]. Inhibition of endogenous *TGF-β* has been shown to trigger downregulation of *SMAD6*, leading to enhanced activation of *BMP* signaling and induction of osteogenesis[51,52]. We thus hypothesized that SAG craniosynostosis could be phenocopied in wild-type SAG suture by inhibiting *TGF-β* signaling with the small molecule SB431542[51,52]. To test this approach, SAG sutures explanted from pN3 wild-type mice were treated with SB431542. After 8 days, treated sutures fused, whereas untreated SAG sutures remained patent (Fig. 3c). In treated SAG sutures, the suture mesenchyme was replaced by bony tissue bridging the osteogenic fronts, while in untreated sutures, mesenchyme remained present within the approaching osteogenic fronts. Consistent with previous reports[51,52], PCR analysis confirmed significant down-regulation of *Smad6* gene expression in SB431542-treated sutures (Fig. 3d). SB431542 treatment also triggered upregulated expression of *Id2* (Fig. 3e), a specific target of activated *BMP* signaling[53], and osteocalcin (*Bglap*), a marker of terminal osteoblast differentiation (Fig. 3f).

Having phenocopied a SAG model of non-syndromic craniosynostosis, we next performed FACS profiling of CD51[+];CD200[+] cells. We observed a significant decrease of these cells in SB431542-treated SAG sutures compared to untreated sutures (Fig. 3g). Collectively, these results demonstrate that a significant change in the ratio of skeletal stem/progenitor cell populations is a shared signature in our models of both syndromic and non-syndromic craniosynostosis.

**Modulation of cWnt signaling alters CD51[+];CD200[+] frequency.** We previously showed that physiologic closure of the PF suture is dependent on inactivation of endogenous cWnt signaling at the onset of endochondral ossification[13]. Conversely, patency of the SAG and COR sutures is maintained by sustained activation of

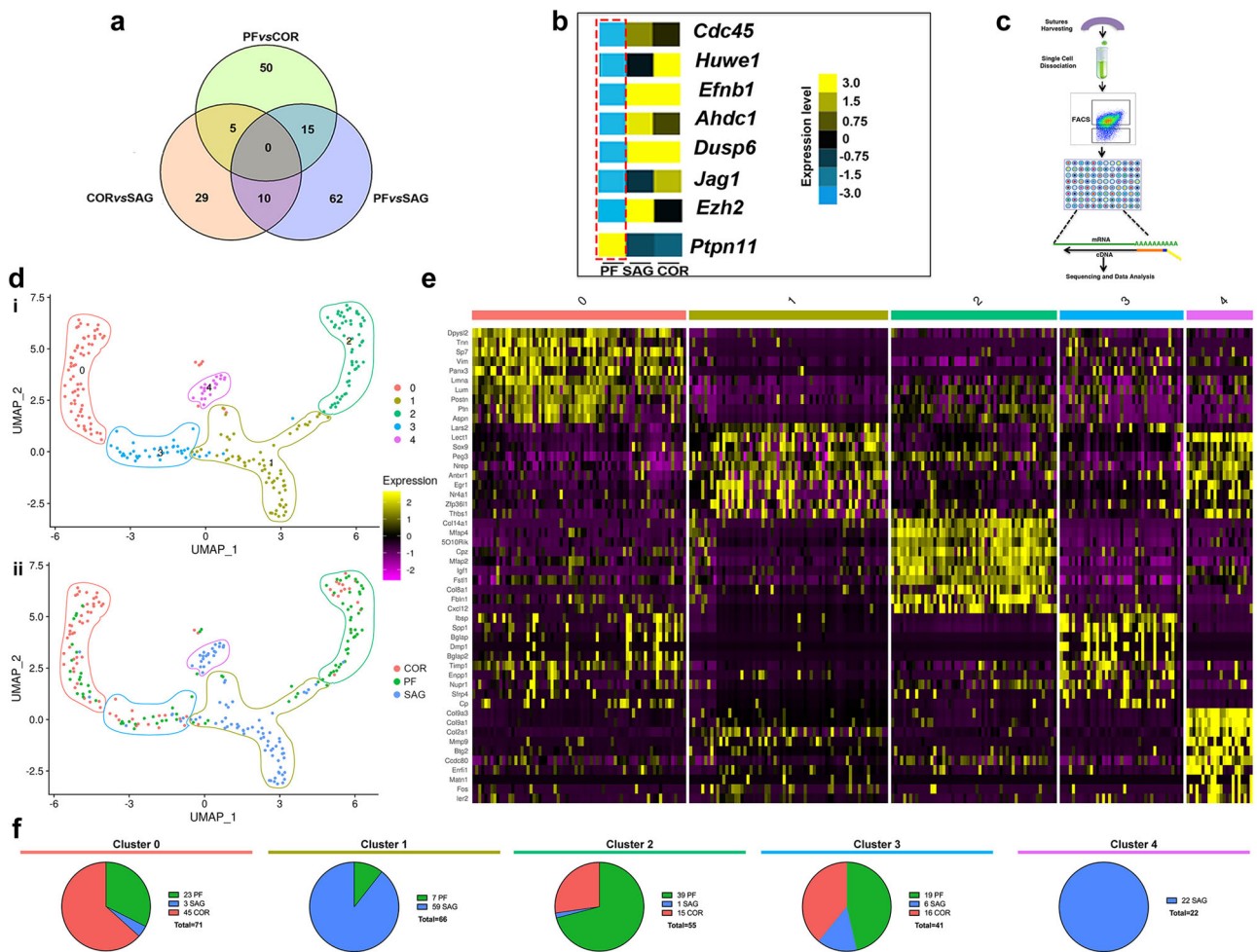

**Fig. 2 Transcriptomic profiling of CD51+;CD200+ cells from fusing and non-fusing sutures. a** Venn diagram of top differentially expressed genes (up- or downregulated by ≥3 fold) from bulk RNA-seq evaluation of skeletal stem/progenitor cells (CD51+;CD200+) isolated from different sutures. **b** Heatmap representation of top genes associated with syndromic craniosynostosis in CD51+;CD200+ cells isolated from PF sutures[37–43]. Yellow: upregulation; blue: downregulation. **c** Schematic outlining single-cell RNA analysis. **d** UMAP embedding showing the distribution of transcriptional programs for single CD51+; CD200+ cells isolated from PF, SAG, and COR sutures partitioned into five clusters. Cells are colored either by cluster (**i**) or suture of origin (**ii**). **e** Heat map of top 10 differentially expressed genes characteristic to each of the five transcriptionally-defined clusters. Yellow: upregulation; purple: downregulation. **f** Pie chart showing the representation of CD51+;CD200+ cells from each suture within each cluster. Sequencing data were deposited in the GEO database as GSE69909. All data can be accessed from the Gene Expressions Omnibus (http://www.ncbi.nlm.nih.gov/geo/) using accession number GSE138882. PF posterior frontal, SAG sagittal, COR coronal.

cWnt signaling over time[13]. Furthermore, manipulation of endogenous cWnt signaling is sufficient to reverse the fate of PF and SAG sutures[13]. These findings, along with the observation that CD51+; CD200+ cells sharply decrease prior to PF suture fusion (Fig. 1b), prompted us to investigate whether continuous activation of cWnt signaling (leading to PF suture patency) would affect the representation of these cells in the suture. PF sutures were treated with Wnt3a protein (Fig. 4a). Effective activation of cWnt signaling was confirmed by treating PF sutures of $Axin2^{LacZ/+}$ reporter mice (enabling detection of cWnt signaling through the X-gal reporter) with Wnt3a and performing X-gal staining as a readout of cWnt activity[13,54] (Supplementary Fig. 7a, b). As previously observed[13], an expanded suture mesenchyme and prevention of suture fusion was seen following exogenous activation of cWnt signaling in the PF suture (Fig. 4b). FACS analysis revealed significantly increased CD51+;CD200+ cells in PF sutures following Wnt3a treatment relative to PBS or control (Fig. 4c). This observation was also supported by FACS profiling performed on $Axin2^{LacZ/LacZ}$ mice endowed with enhanced activation of cWnt signaling due to targeted

disruption of $Axin2$, an inhibitor of cWnt signaling[54]. (Supplementary Fig. 7c).

To evaluate if Wnt3a treatment in vivo could lead to potential polyclonal proliferation, we analyzed the effects of Wnt3a treatment on PF sutures of $ActinCre^{ERT2}:Rainbow^{+/+}$ mice[31,35,55]. Our analysis suggested the possibility of an enrichment of clones, including but not exclusively composed of CD51+;CD200+ cells, within the suture mesenchyme of Wnt3a-treated sutures relative to untreated sutures. These data suggest increased cellular proliferation as a result of Wnt3a treatment (Supplementary Fig. 7d). Furthermore, we performed an in vitro proliferation assay of pN3 suture-derived CD51+; CD200+ cells, in order to determine if cell expansion was a direct effect triggered by Wnt3a, rather than solely a response of these cells to paracrine signaling from neighboring cells within the suture mesenchyme. Wnt3a treatment induced significantly higher EdU incorporation compared to untreated controls, thus inferring increased proliferation in Wnt3a-treated CD51+; CD200+ cells (Supplementary Fig. 7e).

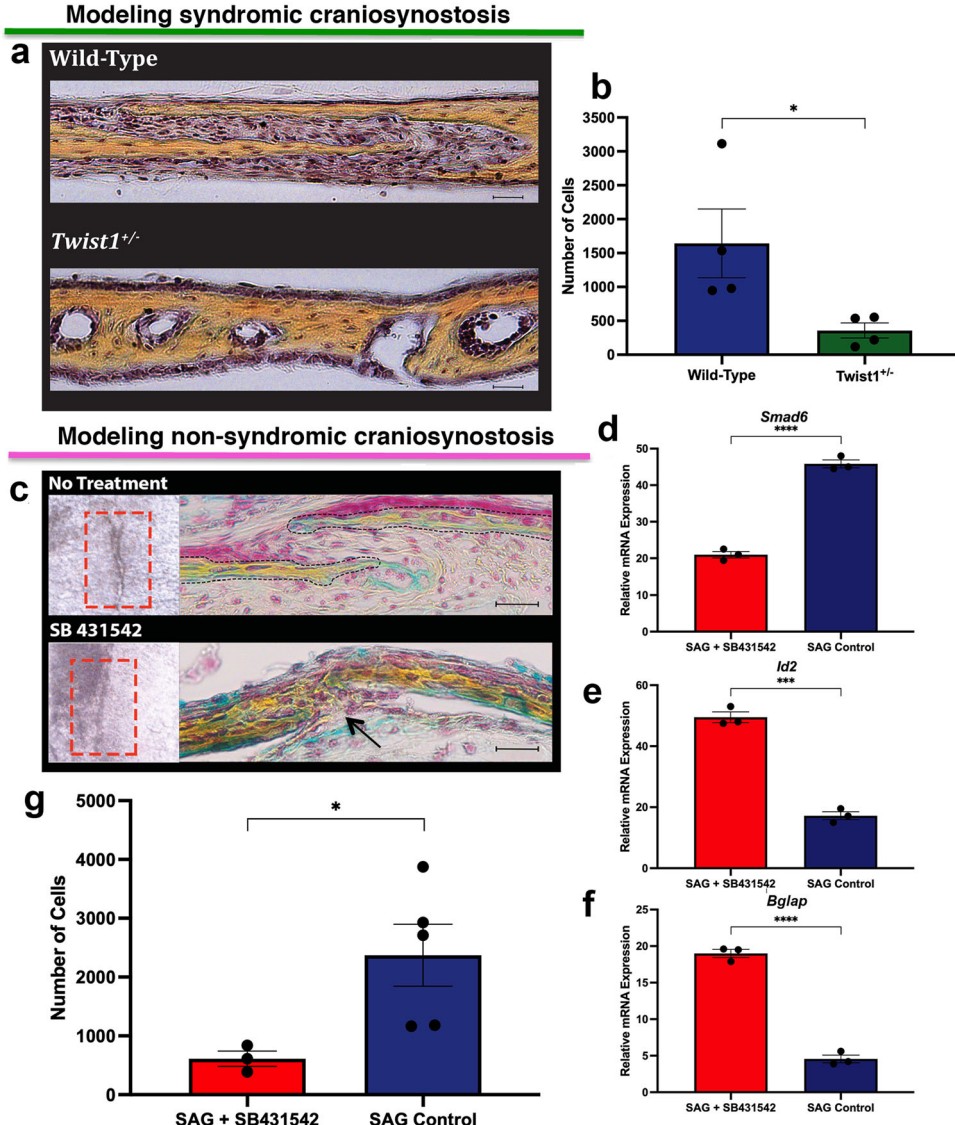

**Fig. 3 Imbalance of CD51⁺;CD200⁺ cells in syndromic and non-syndromic craniosynostosis. a** Representative Movat's pentachrome staining showing synostosis of COR suture in day pN15 *Twist1*⁺/⁻ mouse, compared to wild-type suture. Scale bars: 100 μm. Magnification at ×20. *n* = 3 animals/group, experiments were repeated 3 independent times. **b** FACS profiling of CD51⁺;CD200⁺ cells isolated from explanted wild-type and *Twist1*⁺/⁻ COR sutures at day pN3 reveals a significant decrease of these cells in *Twist1*⁺/⁻ COR sutures in comparison to wild-type. Values are normalized as total CD51⁺;CD200⁺ cells per 500,000 events. Data are represented as means ± SEM; *P* = 0.0482, *\*P* ≤ 0.05, unpaired, two-tailed student *t*-test were performed. *n* = 10 animals/timepoint, experiments were repeated 4 independent times. **c** Whole mount top view (left panel) and Movat's pentachrome staining (right panel) of coronal sections of untreated and SB431542-treated SAG suture explants (day pN3) harvested after 8 days in culture. SB431542-treated SAG suture lacks the suture mesenchyme, with bony tissue (black arrow) replacing the suture mesenchyme and fused. Scale bars: 200 μm. Magnification at ×40. *n* = 5 animals/group, experiments were repeated 3 independent times. **d–f** PCR analysis of *Smad6*, *Id2*, and *Bglap* expression in SB431542-treated and untreated SAG sutures confirming the effectiveness of SB431542 treatment and inhibition of TGFβ signaling paralleled by activation of BMP signaling. Values represent mean ± SEM; *Smad6*; *P* = < 0.0001, *Id2*; *P* = 0.0001, *Bglap*; *P* = < 0.0001, *\*\*\*P* ≤ 0.001, *\*\*\*\*P* ≤ 0.0001, unpaired, two-tailed student *t*-test were performed. *n* = 5 animals/group experiments were repeated 3 independent times. **g** FACS profiling of CD51⁺;CD200⁺ cells isolated from day pN3 untreated and treated sutures harvested after 8 days in culture shows a decrease of these cells in SB431542-treated SAG suture explants. Values are normalized as a total CD51⁺; CD200⁺ cells per 500,000 events. Values represent mean ± SEM; *P* = 0.0478, *\*P* ≤ 0.05, unpaired, two-tailed student *t*-test were performed. *n* = 20 animals/group, experiments were repeated 3 independent times. SAG sagittal, COR coronal. Source data are provided as a Source data file.

Conversely, inhibition of cWnt signaling in patent SAG sutures with *sFrp-1* and *Dkk-1* inhibitors markedly decreased CD51⁺; CD200⁺ cell frequency (Fig. 4d) and resulted in a nearly complete closure of the SAG suture (Fig. 4e). Clonal analysis of Wnt inhibitor-treated *ActinCre*^ERT2^*:Rainbow*⁺/⁺ sutures revealed a decrease in polyclonality relative to untreated and PBS-treated control groups (Supplementary Fig. 7f). These findings demonstrate that activation of cWnt signaling in the PF suture during the window of physiologic fusion enriches skeletal stem/ progenitor cells representation, whereas inhibition of cWnt signaling in the patent SAG suture decreases them and promotes pathologic fusion.

**Activation of cWnt signaling rescues craniosynostosis.** We next questioned whether there would be decreased polyclonal proliferation in *Twist1*⁺/⁻: *ActinCre*^ERT2^ *Rainbow*⁺/⁺ COR sutures mirroring the

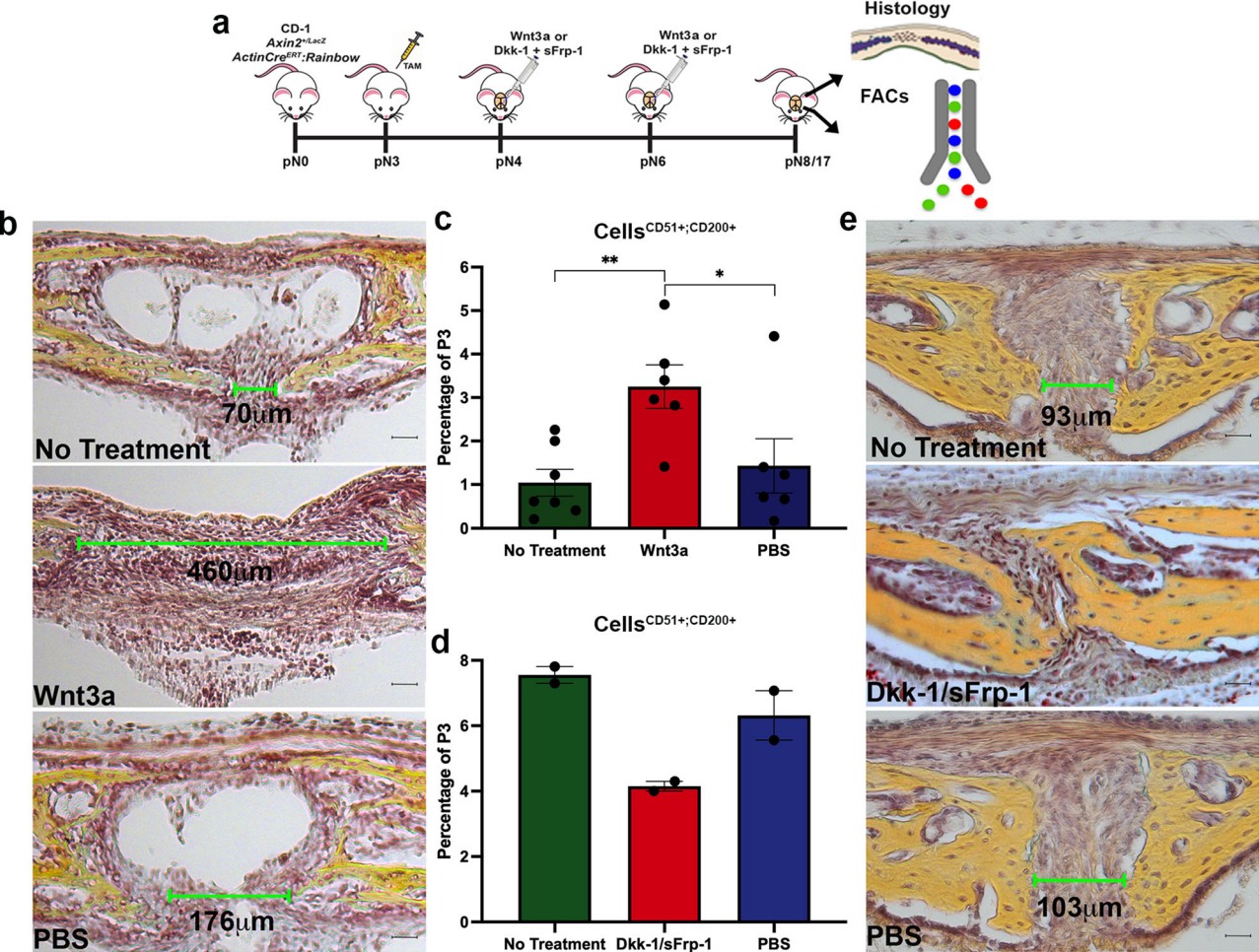

**Fig. 4 Modulation of cWnt signaling affects CD51+;CD200+ cell frequency in the suture mesenchyme. a** Schematic representation of the experimental procedures. TAM, Tamoxifen. **b** Movat's pentachrome staining of coronal sections from CD-1 mouse PF sutures treated in vivo at day pN4 with Wnt3a (150 ng) and harvested at day pN8 (normal onset of fusion). Movat's pentachrome staining shows a largely expanded suture mesenchyme in Wnt3a-treated sutures relative to PBS and untreated controls. Top: no treatment, middle: Wnt3a, bottom: PBS. Green bars mark the distance between the two osteogenic fronts. Scale bars: 100 μm. Magnification at ×20. n = 3 animals/group, experiments were repeated 3 independent times. **c** FACS profiling of freshly harvested Wnt3a-treated PF sutures at day pN8 reveals an increased representation of CD51+;CD200+ cells in these sutures relative to untreated and PBS controls. Values represent mean ± SEM; Wnt3a vs No Treatment, P = 0.0025, Wnt3a vs PBS, P = 0.046 *P ≤ 0.05, **P ≤ 0.01, unpaired, two-tailed student t-test were performed. n = 20 animals/group, experiments were performed 6 independent times. **d** FACS profile of freshly harvested SAG sutures at pN8 following in vivo treatment with cWnt signaling inhibitors sFrp-1 (2 μg) and Dkk-1 (2 μg) or PBS. Treatment with inhibitors decreased CD51+;CD200+ cells in comparison to PBS and untreated controls. Values represent mean ± SEM. n = 20 animals/group, experiments were performed two independent times. **e** Movat's pentachrome staining of SAG sutures treated with cWnt signaling inhibitors sFrp-1 (2 μg) and Dkk-1 (2 μg) and relative controls at day pN17. Green bars mark the distance between the two osteogenic fronts. Scale bars: 100 μm. Magnification at ×20. Top: no treatment, middle: Dkk-1/sFrp-1, bottom: PBS. n = 3 animals/group, experiments were repeated 3 independent times. PF posterior frontal, SAG sagittal. Source data are provided as a Source data file.

observed reduction of CD51+;CD200+ cells in *Twist1*+/− COR sutures (Fig. 3b). We observed decreased polyclonality in double transgenic *Twist1*+/−:*ActinCre*^ERT2^ *Rainbow*+/+ mice relative to control. By contrast, increased polyclonality was observed in COR sutures in *Axin2*^LacZ/+^:*ActinCre*^ERT2^ *Rainbow*+/+ mice, which display enhanced activation of cWnt signaling (Supplementary Fig. 8a–d). Given these findings, we next investigated whether sustained activation of cWnt signaling in a craniosynostotic background could restore the required balance of CD51+;CD200+ cells and thereby prevent/rescue the craniosynostosis phenotype.

We generated a double transgenic *Twist1*+/−:*Axin2*^lacZ/+^ mouse by crossing *Twist1*+/− mice (bearing coronal craniosynostosis) with *Axin2*^LacZ/+^ mice (Supplementary Fig. 9a–d). Pentachrome staining of COR sutures revealed suture patency and a large undifferentiated suture mesenchyme in both wild-type and *Twist1*+/−:*Axin2*^lacZ/+^ COR sutures as compared to synostotic *Twist1*+/− sutures (Fig. 5a).

Remarkably, FACS analysis revealed that in *Twist1*+/−:*Axin2*^lacZ/+^ COR sutures, CD51+;CD200+ cells were restored to wild-type levels (Fig. 5b). Ensuing screening of the *Twist1*+/−:*Axin2*^lacZ/+^ double transgenic COR sutures for complete unilateral or bilateral craniosynostosis revealed a frequency of 0% in wild-type mice, as expected; 83% in *Twist1*+/− mice; and 0% in *Twist1*+/−:*Axin2*^lacZ/+^ double transgenic mice. (Fig. 5c). Notably, a robust contribution of *Axin2*+ (LacZ+) cells was observed within the mesenchyme of the *Twist1*+/−:*Axin2*^lacZ/+^ COR suture at day pN3, day pN18, and 6 months (Fig. 5d and Supplementary Fig. 10). Moreover, immunostaining of activated β-catenin revealed levels of endogenous active cWnt signaling within the COR suture mesenchyme of *Twist1*+/−:*Axin2*^lacZ/+^ double transgenic similar to that of wild-type and *Axin2*^lacZ/+^ COR sutures (Supplementary Fig. 11a–d). Conversely, a faint immunostaining was detected in *Twist1*+/− COR sutures. Microcomputed tomography (μCT) confirmed COR suture patency

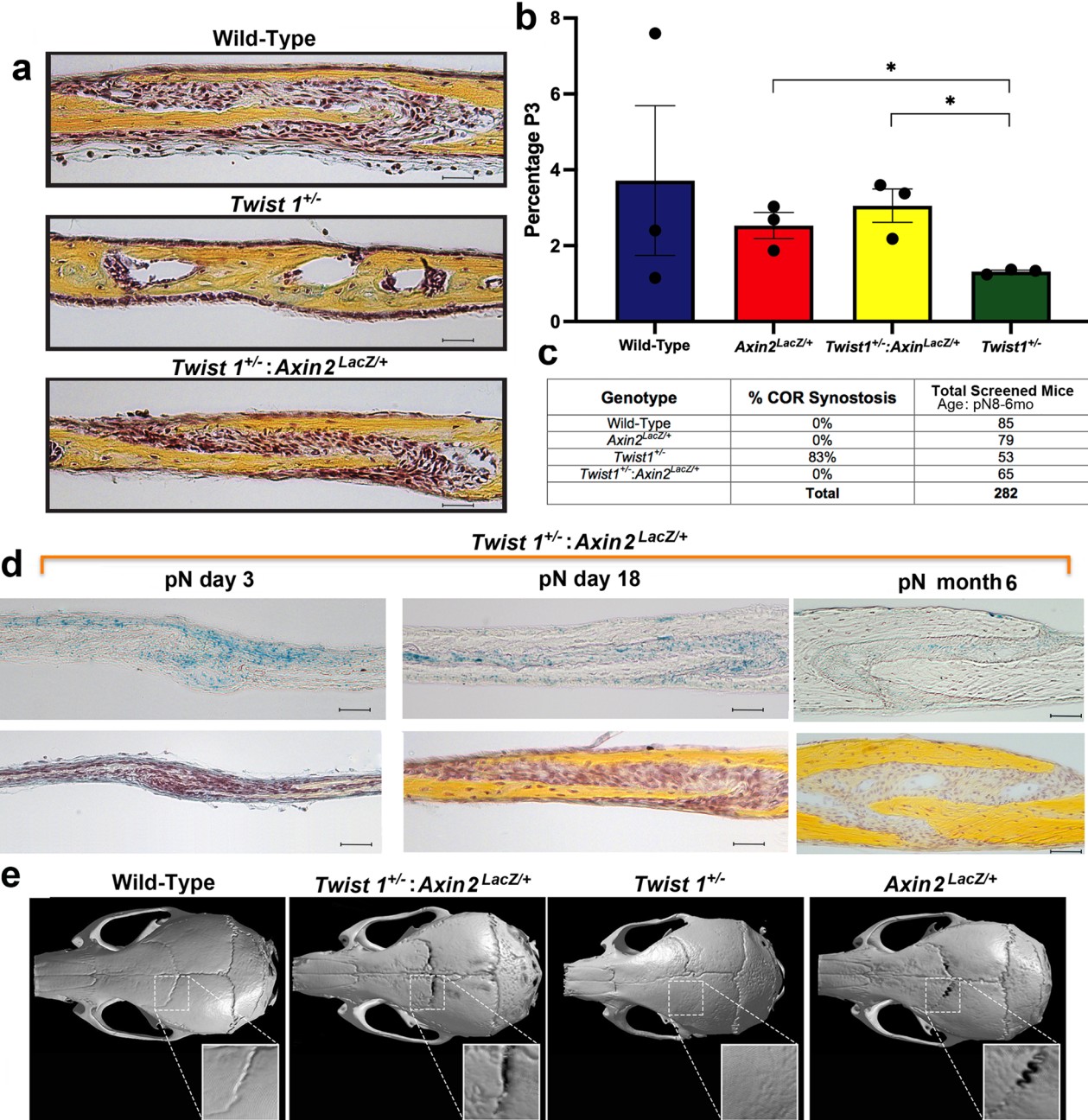

**Fig. 5 Sustained activation of cWnt signaling rescues COR suture craniosynostosis in *Twist1*$^{+/−}$ mice. a** Pentachrome staining and histological analysis of pN15 COR sutures from wild-type (top panel), *Twist1*$^{+/−}$ (middle panel) and *Twist1*$^{+/−}$:*Axin2*$^{LacZ/+}$ (bottom panel) mice. Similar to the wild-type suture, the *Twist1*$^{+/−}$:*Axin2*$^{LacZ/+}$ COR suture is patent with a clear undifferentiated tissue mesenchyme between the osteogenic fronts, thus demonstrating a rescue of the COR suture craniosynostosis. Conversely, the *Twist1*$^{+/−}$ COR suture is fused, showing a bony bridge and lack of a suture mesenchyme. Scale bars, 100 μm. Magnification at ×20. $n = 3$ animals/group, experiments were repeated 3 independent times. **b** FACS analysis of CD51$^+$;CD200$^+$ cells isolated from COR sutures of wild-type, *Axin2*$^{LacZ/+}$, *Twist1*$^{+/−}$, and *Twist1*$^{+/−}$:*Axin2*$^{LacZ/+}$ mice indicates increased representation of CD51$^+$;CD200$^+$ cells in the *Twist1*$^{+/−}$:*Axin2*$^{LacZ/+}$ double transgenic. Values represent mean ± SEM; *Axin2*$^{LacZ/+}$ vs. *Twist1*$^{+/−}$ $P = 0.025$, *Twist1*$^{+/−}$ vs. *Twist1*$^{+/−}$: *Axin2*$^{LacZ/+}$, $P = 0.0172$; *$P ≤ 0.05$, unpaired, two-tailed student *t*-test were performed. $n = 10$ animals/group, experiments were repeated 3 independent times. **c** Screening for coronal suture fusion in *Twist1*$^{+/−}$ versus *Twist1*$^{+/−}$:*Axin2*$^{LacZ/+}$ mice reveals a frequency of COR craniosynostosis in *Twist1*$^{+/−}$: *Axin2*$^{LacZ/+}$ mice restored to that of wild-type mice (i.e., patent COR suture phenotype), compared to the widespread fusion seen in *Twist1*$^{+/−}$. Screening was performed on mice ranging from day pN6 up to 6 months. $n = 282$ animals screened. **d** (top panel) X-gal staining identifies a of cWnt-activated cells within the mesenchyme of the patent COR suture in *Twist1*$^{+/−}$: *Axin2*$^{LacZ/+}$ double transgenic mice at day pN3, pN18, and 6 months postnatal. (bottom panel) Movat's pentachrome staining performed on adjacent slide sections as for top panels. Scale bars: 100 μm. Magnification at ×20. $n = 3$ animals/ group, experiments were repeated 3 independent times. pN post-natal. **e** Micro-CT analysis of wild-type, *Twist1*$^{+/−}$:*Axin2*$^{LacZ/+}$, *Twist1*$^{+/−}$, and *Axin2*$^{LacZ/+}$ skulls at 6 months postnatal reveals the presence of a wide patent COR suture in the *Twist1*$^{+/−}$:*Axin2*$^{LacZ/+}$ mouse, whereas a unilateral fused COR suture is observed in the *Twist1*$^{+/−}$ mouse. $n = 3$ animals/group, experiments were repeated 3 independent times. COR coronal. Source data are provided as a Source data file.

in $Twist1^{+/-}:Axin2^{lacZ/+}$ mice (Fig. 5e). Collectively, these data support that cWnt activation in the suture mesenchyme of the $Twist1^{+/-}:Axin2^{lacZ/+}$ COR suture may suppress COR suture craniosynostosis observed in $Twist1^{+/-}$ mice, possibly through expansion of the CD51$^+$;CD200$^+$ cell population.

**Preventing re-synostosis following suturectomy.** Conventional methods of treatment for craniosynostosis require complex surgical intervention to correct the fused suture, frequently resulting in re-fusion of the suture. Thus, there is a strong unmet need for cellular or molecular therapies to treat craniosynostosis and prevent resynostosis. Our data suggest that skeletal stem/progenitor cell equilibrium within the suture mesenchyme may maintain suture patency. Moreover, active cWnt signaling may promote cell expansion within the suture mesenchyme to levels sufficient for sustaining patency. Based on these findings we hypothesized that by transplanting skeletal stem/progenitor cells and Wnt3a protein following ablation of a synostotic suture, we could model a physiological suture mesenchyme, thereby preventing suture re-fusion. To test this hypothesis, we surgically ablated the synostotic COR suture of $Twist1^{+/-}$ mice via suturectomy (Fig. 6a), reflecting the surgical management of human craniosynostosis. CD51$^+$;CD200$^+$ cells co-seeded with Wnt3a in a collagen sponge carrier were transplanted in the suturectomy site, with untreated suturectomies acting as controls (Fig. 6a). The extent of re-fusion was assessed by microcomputed tomography (µCT) and histology over a 14-week period (Fig. 6b–d). After 14 weeks the control suturectomy sites re-fused completely, while suturectomies treated with CD51$^+$;CD200$^+$ cells and Wnt3a protein remained patent with the appearance of a restored mesenchyme (Fig. 6d). Additionally, transplanted GFP$^+$ CD51$^+$;CD200$^+$ cells were used to confirm their contribution to the suture mesenchyme (Fig. 6e). Interestingly, neighboring cells, in addition to the transplanted GFP$^+$ CD51$^+$;CD200$^+$ cells, appeared to participate in recreating the suture mesenchyme, thus reestablishing a "suture niche/microenvironment" (Fig. 6f). Conversely, mice treated with Wnt3a or CD51$^+$;CD200$^+$ cells alone, developed recurrent synostosis (Supplementary Fig. 12a–d).

## Discussion

This study examined cranial suture biology through the lens of a skeletal stem/progenitor cell population (namely CD51$^+$;CD200$^+$), demonstrating a link between, CD51$^+$;CD200$^+$ cell equilibrium and cranial suture patency, thus highlighting the potential significance of these cells in craniosynostosis. Following isolation of CD51$^+$;CD200$^+$ population and validation of their multi-potency and self-renewal capacity, we performed an in vivo time course profiling of these cells derived from mouse cranial sutures using FACS analysis. Our data revealed significant differences in skeletal stem/progenitor cell representation between patent versus fusing cranial sutures—specifically, the number of these cell in the PF suture decreases as a function of time in advance of physiological suture fusion. Furthermore, we identified that a significant decrease in skeletal stem/progenitor cell is a shared feature in our models of both syndromic and non-syndromic craniosynostosis.

As a dynamic representation of CD51$^+$;CD200$^+$ cells is implicated in patent versus fusing cranial sutures, we performed transcriptional profiling of the cells derived from the patent (COR, SAG) versus the fusing (PF) sutures. Interestingly, genes in which loss-of-function mutations are known to lead to craniosynostosis were almost exclusively downregulated in PF-derived skeletal stem/progenitor cells. The PF is the only suture physiologically fusing during early life, and we found an array of genes to be uniquely downregulated in PF-derived cells relative to COR-

and SAG-derived cells, suggesting a potential transcriptional signature of skeletal stem/progenitor cells fated for suture fusion. Furthermore, scRNA sequencing of cranial suture derived skeletal stem/progenitor cells identified multiple transcriptionally-defined clusters with gene expression profiles appearing to favor osteogenic versus chondrogenic cell fates. This would suggest that transcriptionally-defined clusters may represent transcriptional/functional primed cells. However, prospective isolation and evaluation of these putative subgroups would be required to determine whether they represent functionally distinct cell subpopulations. These findings highlight the transcriptional heterogeneity among suture-derived skeletal stem/progenitor cells, which may be reflective, in part, of differences in the niches wherein these cells reside and the complex tissue architectures within the cranial suture.

The importance of cWnt signaling in maintaining the undifferentiated state of stem cells by preventing them from progressing towards more differentiated lineages is well known[56,57]. Building on prior work by Behr et al.[13], we interrogated if cWnt signaling rescued craniosynostosis by acting directly on these skeletal stem/progenitor cells. Our data indicated that cWnt signaling not only induces the expansion of this population, both in vitro and in vivo, but can also prevent craniosynostosis. The observation that Wnt3a induces proliferation of CD51$^+$;CD200$^+$ cells in vitro provides direct evidence for the role of cWnt on cell expansion independent of other potential paracrine signaling transmitted by neighboring cells within the suture.

The role of cWnt signaling on skeletal stem/progenitor cell expansion and craniosynostosis was further investigated by crossbreeding a $Twist1^{+/-}$ mouse harboring COR craniosynostosis with an $Axin2^{LacZ/+}$ mouse, endowed with endogenous activation of cWnt signaling. The resulting double transgenic displayed suppression of the craniosynostosis phenotype and restoration of suture patency, possibly through the expanded skeletal stem/progenitor cell population observed. Furthermore, the preserved COR suture mesenchyme was largely composed of Xgal$^+$ (cWnt active) cells in a "niche" harboring an active cWnt signaling environment.

Having illustrated that activation of cWnt signaling together with the expansion of skeletal stem/progenitor cells are associated with maintenance of suture patency, we explored whether transplantation of these cells combined with Wnt3a protein would have a synergistic effect in preventing re-synostosis following suturectomy in $Twist1^{+/-}$ coronal craniosynostosis. To functionally test the ability of these skeletal stem/progenitor cells to maintain patency post-suturectomy, we developed an "engineered" suture mesenchyme by transplanting CD51$^+$;CD200$^+$ cells combined with Wnt3a protein into an ablated COR suture site of $Twist1^{+/-}$ mice affected by unilateral COR suture synostosis. Transplantation of skeletal stem/progenitor cells with Wnt3a led to patency of the craniosynostotic suture following its ablation, which may be due to suture mesenchyme reconstitution or indeed arise as a result of failure of healing of the bone defect. Our findings are supported by a recent study[58], showing that Gli-1$^+$ MSCs transplantation into a surgically ablated Twist-1$^{+/-}$ craniosynostotic suture prevented resynostosis, however, cWnt stimulation was not required. This difference could reflect ongoing autocrine cWnt signaling in Gli-1$^+$ MSCs, however, this remains a hypothesis. Based on GSE analysis of the cWnt pathway, our cells do not produce Wnt ligands, therefore require exogenous Wnt3a stimulation in order to activate the canonical signaling.

Taken together, the findings stemming from this study shift our focus towards viewing skeletal stem/progenitor cells through a lens magnifying these cells as an appealing tool suitable for preventing bone formation. Moreover, the findings from this study may lay the foundation for therapeutic options using

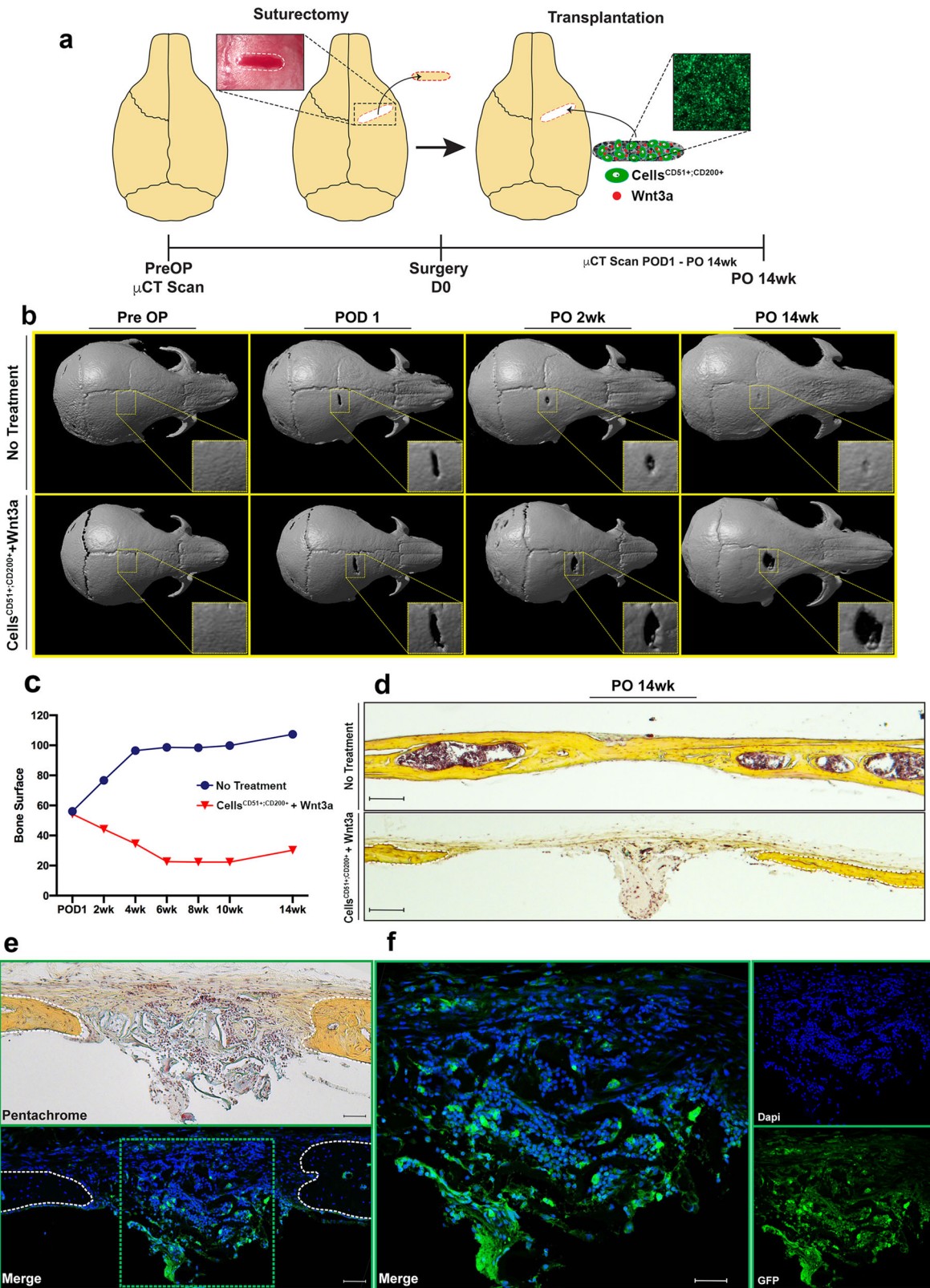

skeletal stem/progenitor cells coached with effectors of cWnt signaling as an early intervention therapy for craniosynostosis.

## Methods

**Animals.** All experiments using animals were performed in accordance with Stanford University Animal Care and Use Committee guidelines. This study complies with all relevant ethical regulations for animal testing and research under the guidance and approval of Stanford University's Administrative Panel on Laboratory Animal Care IACUC/APLAC. Animals were bred and housed in light (12 h light cycle: 7 am on, 7 pm off), temperature (69–79 °F) and moisture-controlled (30–70%) Research Animal Facility and were given food and water *ad libitum* in accordance with Stanford University guidelines, APLAC Protocol 8397 and 21067. CD-1 and C57BL/6 mice were purchased from Charles Rivers Laboratories Inc. *B6 Twist1*$^{+/-}$ (Stock# 2221) and *B6 ACTb-EGFP* (Stock# 3291) mice were purchased from Jackson Laboratory (Bar Harbor, Maine, USA). *Axin2*$^{LacZ/+}$ mice were as previously described[13]. *ActinCre*$^{ERT2}$:*Rainbow*

**Fig. 6 Transplantation of CD51$^+$;CD200$^+$ cells and Wtn3a prevents re-synostosis following surgical repair of craniosynostosis. a** Schematic representation of experimental procedure for COR suture ablation experiments. pN16 *Twist1*$^{+/−}$ COR suture synostosis was confirmed by preoperative micro-CT scans. Fused COR sutures were excised using a 0.3 mm drill and replaced by wild-type CD51$^+$;CD200$^+$ cells (3 × 10$^5$ cells) and recombinant Wnt3a protein (200 ng) loaded onto a collagen scaffold sponge, thus engineering a "suture mesenchyme". An empty suturectomy was used as control. The extent of re-fusion was evaluated over a 14-week period by micro-CT and histological analysis. POD post-operative day, PO post-operative. **b** Time-course micro-CT analysis of treated and untreated COR suturectomies over a 14-week period. Untreated suturectomy re-fuses in contrast to the treated suturectomy which remains patent through 14-weeks. *n* = 2 animals/group, experiments were repeated 3 independent times. **c** Quantification of COR suturectomy re-fusion by micro-CT over a 14-week period measuring image intensity in a standardized ROI encompassing the suturectomy using the Adobe Photoshop measurement tool. Untreated suturectomies (blue) show increased bone surface measurements over the 14-week period compared to suturectomy treated with a combination of CD51$^+$;CD200$^+$ cells and Wnt3a protein (red) confirming the untreated suturectomy is refusing over-time. *n* = 2 animals/group, analysis was performed one time. **d** Movat's pentachrome staining of sagittal sections of COR suturectomies at 14-weeks showing fusion of the untreated suturectomies compared to suturectomies treated with cells and Wnt3a protein. Magnification ×5, scale bars: 200 μm. *n* = 2 animals/ group, experiments were repeated 3 independent times. **e** wild-type GFP$^+$ CD51$^+$;CD200$^+$ cells (3 × 10$^5$ cells) coached with recombinant Wnt3a protein (200 ng) loaded onto a collagen scaffold sponge were transplanted as above to monitor their presence within the ablated COR suture at 10-week post-operative. (top panel), Movat's pentachrome staining of a COR suturectomies sagittal section. Magnification ×10, scale bar: 100 μm. (bottom panel), confocal micrograph of a sister slide of COR suturectomies sagittal section. Fluorescence indicates the presence of transplanted cells within the ablated suture. Magnification ×10, scale bar: 100 μm. **f** Magnification at ×20 of the boxed area in (**e**), suggests that transplanted CD51$^+$;CD200$^+$ cells participate in forming suture mesenchyme by recruiting neighboring cells Magnification ×20, scale bar: 50 μm. *n* = 2 animals/group, experiments were repeated 3 independent times. COR coronal. Source data are provided as a Source data file.

mice were a kind gift from the laboratory of Dr. Irv Weissman, Institute for Stem Cell Biology and Regenerative Medicine, Stanford University. *Twist1*$^{+/−}$ mice were crossed with *Axin2*$^{LacZ/+}$ mice to generate *Twist1*$^{+/−}$:*Axin2*$^{LacZ/+}$ mice. *Twist1*$^{+/−}$ mice were crossed with *ActinCre*$^{ERT2}$:*Rainbow* mice to obtain *Twist1*$^{+/−}$:*ActinCre*$^{ERT2}$:*Rainbow* and *Axin2*$^{LacZ/+}$ crossed with *ActinCre*$^{ERT2}$:*Rainbow* to obtain *Axin2*$^{LacZ/+}$:*Actin-Cre*$^{ERT2}$:*Rainbow* mice. *Axin2*$^{LacZ/+}$, *Twist1*$^{+/−}$, *ACTb-EGFP*, and *ActinCre*$^{ERT2}$:*Rainbow* mice were genotyped as previously described[54,55,59,60]. Genotyping primer sequences and expected results can be found in Supplementary Table 1.

**In-vivo suture harvesting and cell dissociation**. Animals (*n* = 60) were sacrificed by CO$_2$ asphyxiation for each postnatal (pN) time-point, pN3, pN5, pN7, pN9, pN11, pN15, and pN17 on the exact postnatal day determined by date of birth. The entire cranium was harvested, lightly washed in cold PBS and place in cold FACS buffer composed of PBS (Gibco-Life Technologies, Grand Island, NY, USA), 2% fetal bovine serum (Gibco-Life Technologies, Grand Island, NY, USA), 1% penicillin–streptomycin (Gibco-Life Technologies, Grand Island, NY, USA), 1% Pluronic F-68 (Gibco-Life Technologies, Grand Island, NY, USA). The PF, SAG and COR sutures were carefully dissected on ice with the aid of a dissection stereo microscope (Zeiss, Oberkochen, Germany). Approximately 0.5 mm by 1–3 mm (depending on pN time-point) region was carefully removed including the osteo-genic fronts of the cranial plates, suture mesenchyme, underlying dura mater and overlying periosteum (Suture Complex) using fine forceps and a small scalpel. Cell dissociation and digestion were performed as previously described[31,32]. Suture tissue was gently minced using fine tipped, dissecting scissor in a 2 ml microcentrifuge tube and chemically digested in 1.5 ml of serum-free Medium 199/EBS (HyClone, Logan, UT, USA) supplemented with 2.2 mg/ml Collagenase Type II (Sigma-Aldrich, St. Louis, MO, USA), 2 mg/ml Dispase II (Roche Diagnostics, Indianapolis, IN, USA), 1% Bovine Serum Albumin (Sigma-Aldrich, St. Louis, MO, USA), 1% Pluronic F-68 (Gibco-Life Technologies, Grand Island, NY, USA), 2% HEPES Buffer (Gibco-Life Technologies, Grand Island, NY, USA), 0.4% 2.5 M CaCl$_2$ (Sigma-Aldrich, St. Louis, MO, USA), 100 units/ml deoxyribonuclease I (Worthington Biochemistry, Lakewood, NJ, USA), incubated in a 37 °C water bath for 10 min followed by 2 digestions, each of 25 min, carried out in an orbital shaker at 275 rpm at 37 °C. Dissociated cells were filtered through a 70 μm filter and neutralized with 2× volume FACS buffer. The digestions were pooled together and total dissociated cells were pelleted at 1350 rpm at 4 °C, resuspended in FACS buffer, blocked with rat IgG and stained with fluorochrome-conjugated antibodies Ter119 (1:200, 15-5921-81), CD51 (AlphaV integrin) (1:50, 551187), CD105 (1:50, 13-1051-82), Thy1.1 (1:100, 47-0900-82), Thy1.2 (1:100, 47-0902-82), Tie2 (1:20, 14-5987-85), eBioscience, San Diego, CA, USA, CD45 (1:200, 103109), 6C3 (1:100, 108311) and CD200 (1:50, 123802), Biolegend, San Diego, CA, USA for 20 min under light agitation. Stained cells were pelleted at 1350 rpm at 4°, resuspended in 300 μl of FACS buffer and filtered through a 35 μm filter for fractionation by flow cytometry. For information regarding antibodies and media used for FACS isolation of skeletal stem/progenitor cells refer to Supplementary Tables 2 and 3.

**Fluorescence assisted cell sorting (FACS)**. Flow cytometry was performed on the FACS Aria II in the Lorey Lokey Stem Cell Institute Shared FACS Facility. Hematopoietic (CD45$^+$) and dead cells (Pi$^+$) were gated out and the remaining population (P3) was fractionated based on the following surface antigens as previously described[31,32]. Skeletal stem/progenitor cell (CD51$^+$;CD200$^+$): CD45$^−$, Ter119$^−$, Tie2$^−$, Thy1.1$^−$, Thy1.2$^−$, 6C3$^−$, CD105$^−$, CD51$^+$, CD200$^+$. Highly

pure, double sorted skeletal stem/progenitor cells were sorted directly into TRIzol Reagent (Ambion-Life Technologies, Carlsbad, CA, USA) for RNA isolation and extraction or alpha-MEM supplemented with 20% fetal bovine serum, 1% penicillin–streptomycin (Gibco-Life Technologies, Grand Island, NY, USA), and 0.1% ciprofloxacin HCl (bioWORLD, Dublin, OH, USA) for cell culture and dif-ferentiation assays. For information regarding antibodies and media used for FACS isolation of skeletal stem/progenitor cells refer to Supplementary Tables 2 and 4.

**FACS normalization methods**. All in-vivo FACS analysis were normalized by calculation of percentage of P3. Total number of CD51$^+$;CD200$^+$ cells encom-passing the skeletal stem/progenitor cell population (CD45$^−$, Ter119$^−$, Tie2$^−$, Thy1.1$^−$, Thy1.2$^−$, 6C3$^−$, CD105$^−$, CD51$^+$, CD200$^+$) were normalized to the P3 population (Pi$^−$/Ter119$^−$/CD45$^−$) for comparison among the three cranial sutures and various treatment groups. For in-vitro FACS analysis (Fig. 3), CD51$^+$;CD200$^+$ cells were normalization as total number of cells per 500,000 events. (Additional information on can be found in Supplementary Fig. 1). Analysis was performed using FACS Diva (BD) v8.0.1 and FlowJo (TreeStar) 10.1r5 software packages.

**Histology**. Whole cranium was harvested and fixed overnight at 4 °C, either in 0.4% or 4.0% PFA (Electron Microscopy Sciences, Hatfield, PA, USA) and dec-alcified in 19% EDTA at 4 °C for 2 days to 2 months (depending on the age of the specimen). Following appropriate decalcification, specimens were prepared for cryo-embedding by soaking in 30% (mass/vol) sucrose in PBS at 4 °C for 24 h and embedded in Tissue Tek O.T.C (Sakura Finetek, Torrance, CA, USA). The entire PF, SAG, and COR suture were cut in 10 μm sections. Sister slides were then selected and stained for X-gal and Pentachrome staining or confocal imaging for direct comparison of the different sutures. X-gal (Roche Indianapolis, IN, USA) and Pentachrome staining was performed as previously described[13]. Images are representative of at least 2 independent samples or experiments. Details of reagents used are listed in Supplementary Table 4.

**Tamoxifen induction of rainbow reporter system**. Rainbow mice[31,35,55] were employed for clonal analysis of the cranial sutures. Rainbow mice were crossed with the ubiquitous *ActinCre*$^{ERT2}$ driver to mark cells under the actin promoter after systemic tamoxifen injection. The Rainbow reporter ($R26^{VT2/GK3}$) is a mul-ticolor Cre-dependent marker system with a four-color reporter construct in the ROSA locus. Once recombination occurs, cells are randomly and genetically marked with one of ten possible color combination. The resulting progeny will be marked with the same color as the parent cell, creating a fluorescent mosaic pattern upon analysis. For the induction, *ActinCre*$^{ERT2}$:*Rainbow Twist1*$^{+/−}$:*ActinCre*$^{ERT2}$:*Rainbow* and *Axin2*$^{LacZ/+}$:*ActinCre*$^{ERT2}$:*Rainbow* mice were subcutaneously injected with 50 μl tamoxifen at 20 mg/ml at day pN3 and harvested at pN7 or pN8 depending on experimental design.

**Imaging analysis**. Laser scanning confocal microscopy was performed with a LEICA TCS SP8× confocal microscope (LEICA Microsystems, Buffalo Grove, IL, USA) with an objective lens (×10 HC PL APO, air, N.A. 0.40; ×20 HC PL APO IMM CORR CS2, H2O/Glycerol/oil, N.A. 0.75), located in the Cell Sciences Imaging Facility (Stanford University, Stanford, CA). Raw image stacks were imported into ImageJ (NIH) for further analysis. All clones were individually examined to confirm that they reported a single color. For visualizing individually labeled cells expressing the Rainbow reporter, the brightness and contrast were

adjusted accordingly for the green (eGFP), blue (mCerulean), orange (mOrange), and red (mCherry) channels and composite serial image sequences were assembled. Tiled images were stitched by a grid/collection stitching plugin. Clones were determined by analyzing the rendered composite image containing all four channels, using ImageJ (NIH) software. Cells that were visually determined to be clones were traced with a surface area measurement tool for analysis based on the pixel density of each image (300 pixels per inch). Clones with a surface area greater than or equal to 3.2 mm$^2$ were included. Values are representative of three independent fields analyzed. Experiments were performed two times.

**Transcriptomic analysis.** All RNA sequencing, bulk and single cell, were conducted at the Stanford Functional Genomics Facility (SFGF) core at Stanford University.

**Bulk RNA sequencing.** Skeletal stem/progenitor cells, freshly isolated from the PF, SAG, COR, and growth plate of pN3 CD-1 mice ($n = 60$) were double sorted directly into TRIzol Reagent (Ambion-Life Technologies, Carlsbad, CA, USA) using the FACS Aria II. RNA extraction was done using the Qiagen MiRneasy Kit (Cat#217084, Qiagen, Hilden, Germany). The quality of extracted RNA was evaluated using pico bio-analyzer chip on an Agilent bio-analyzer instrument. cDNA was synthesized and amplified using CloneTEch Ultra low input RNA kit v4 (Cat # 634888, CloneTEch, Mountain View, CA, USA) and fragmented using Covaris. Quality of the fragmented cDNA was assessed using high-sensitivity chip on an Agilent bio-analyzer. Libraries were then prepped using ClonTech Low Input Library Rep Kit v2 (Cat # 634899, CloneTEch, Mountain View, CA, USA). Equal nanomoler of each uniquely indexed library were pooled and sequenced on an Illumina HiSeq 4000 (purchased from NIH funds under award number S10OD018220). Details of reagents used for bulk RNA-sequencing are listed in Supplementary Table 4.

**Single-cell RNA sequencing (scRNA-seq).** Freshly isolated skeletal stem/progenitor cells from the PF, SAG, COR and growth plate of pN3 CD-1 mice ($n = 80$) were double sorted and single cells sorted into 96-well plates with 4 μl per well of lysis buffer consisting of 4 units of Recombinant RNase inhibitor (RRI) (CAT # 2313B, CloneTEch, Mountain View, CA, USA), 0.1% Triton X-100 (CAT# 85111, Thermo Fisher Scientific, Rockford, IL, USA), 2.5 mM dNTP (CAT# 10297018, Thermo Fisher Scientific, Rockford, IL, USA), 2.5 μM oligodT30VN (5′AAGCA GTGGTATCAACGCAGAGTACT30VN-3′, Integrated DNA Technologies, Skokie, IL, USA). Once sorted, cells were immediately spun down and frozen at −80 ° C. Single-cell RNAseq was performed via the Picelli method[61]. Lysis buffer plates were thawed on ice, then heated at 72 °C for 3 min in a Biorad C1000 Touch thermal cycler. First strand cDNA synthesis was performed in a 10 μl reaction with 100 Units of Clontech's Smartscribe reverse transcriptase (CAT# 639538, CloneTEch, Mountain View, CA, USA), 10 Units RRI, 1× First Strand Buffer (CloneTEch, Mountain View, CA, USA), 5 mM DTT, 1 M Betaine (CAT# B0300-5VL, Sigma, St. Louis, MO, USA), 6 mM MgCl$_2$, 1 μM Template Switch Oligo (TSO, (5′-AAGCAGTGGTATCAACGCAGAGTACATrGrG+G-3′, Exiqon/Qiagen, Hilden, Germany) at 42 °C for 90 min, 70 °C for 15 min. PCR pre-amplification was performed in a 25 μl reaction with 1× Kapa HiFi HotStart (CAT# KK2602, Kapa BioSystems, Wilmington, MA, USA), 0.1 μM ISPCR primer (5′-AAGCAGTGGTA TCAACGCAGAGT-3′, Integrated DNA Technologies, Skokie, IL, USA) at 98 °C for 3 min, then 25 cycles of 98 °C for 20 s, 67 °C for 15 s, 72 °C for 6 min, then 72 °C for 5 min. Reactions were cleaned with SPRI beads on a Biomek FX and eluted in 25 μl water, 0.2 μl aliquots were run on an Fragment Analyzer High Sensitivity NGS 1-6000 kit. Barcoded sequencing libraries were made using the miniaturized Nextera XT protocol of Mora-Castilla[62] in a total volume of 4 μl. Pooled libraries were sequenced on a Nextseq 500 High Output flow cell with 2 × 150 paired end reads. Details of reagents used for scRNA-sequencing can be found in Supplementary Table 4.

**Bulk RNA-sequencing data analysis.** Analysis were performed on the paired-ends fastqs using FastQC v0.11.5 prior to proceeding with the read mapping. Fastq reads were mapped to the mouse mm9 reference genome using STAR v2.5.1b in paired-end mode to yield BAM files sorted by coordinates. To estimate transcript abundances of gene features in terms of FPKMs, Cuffdiff was run on the various BAM files with the Cufflinks v2.2.1 suite using the Illumina iGenomes gene model. The web-tool Vennt v0.8.4 (http://drpowell.github.io/vennt/) was used to generate dynamic Venn diagrams from Cuffdiff differential gene expression data generated from bulk RNA-Seq experiments (Fig. 2a). The gene expression visualization tool Java TreeView[63] v1.1.6r4 was used to visualize gene expression heatmaps (Fig. 2b) following pairwise average-linkage hierarchical clustering of log2-normalized expression data using Cluster[64] v3.0, with mean-centering of genes and Euclidean distance as distance measure for clustering.

**Single-cell RNA-sequencing data analysis.** Raw mRNA counts were normalized on a per-cell basis with a scale factor of 10,000 and subsequently natural log transformed with a pseudocount of 1 in R v3.6.0 using the Seurat package v3.1.1[45]. Aggregated data were then evaluated using uniform manifold approximation and projection (UMAP) analysis over the first 15 principal components[65], with

resolution parameters of 1.0 for all primary and secondary cluster analyses with the exception of the sagittal (SAG) suture subset, for which a resolution parameter of 0.8 was used.

**Generation of characteristic cluster markers and enrichment analysis.** Cell-type marker lists were generated with two separate approaches. In the first approach, we employed Seurat's native *FindMarkers* function with a log fold change threshold of 0.25 using the ROC test to assign predictive power to each gene. However, in order to better account for the mutual information contained within highly correlated predictive genes, we also employed a characteristic direction analysis[66]. The 50 most highly ranked genes from this analysis for each cluster were used to perform gene set enrichment analysis in a programmatic fashion using EnrichR v2.1[67].

**Cross-platform comparison of scRNA-seq datasets.** Individual datasets were normalized using the SCTransform algorithm in Seurat followed by label-based anchor transfer with default parameters. Principal component analysis was performed on the resulting integrated dataset, and the top 30 components were used for neighbor finding and UMAP dimension reduction and embedding[45].

**Purity analysis of scRNA-Seq.** Entropy-based purity assessment of scRNA-seq samples using the Ratio of Global Unshifted Entropy (ROGUE)[46] toolkit was implemented to asses purity of our dataset, in addition to the publicly available datasets from Debnath et al., *Nature* 2018 (GSE106237) and Chan et al., *Cell* 2015 (GSE64447).

**RT-PCR validation of selected genes identified from bulk RNA-Seq analysis.** Total RNA was isolated from PF, SAG and COR derived skeletal stem/progenitor cells using the TRIzol method (Invitrogen, Carlsbad, CA, USA). Upon DNAse I (Ambion; Austin, TX, USA) treatment to clear genomic DNA, RNA was reverse transcribed using a SuperScript III First-Strand kit (Invitrogen, Carlsbad, CA, USA) as previously described[68]. Gene expression profile was analyzed by RT-PCR. The relative mRNA level in each sample was normalized to its *Gapdh* content. Values are provided as relative to *Gapdh* expression. Primers sequence and annealing temperature are described in Supplementary Table 5.

**Ex-vivo suture explants.** Sagittal sutures from day pN3 CD-1 mice and coronal sutures from day pN3 *Twist1*[+/−] mice were explanted and placed into 24-multiwell plates and cultured for eight days in DMEM GlutaMax supplemented with 10% fetal bovine serum, 1% penicillin–streptomycin Gibco-Life Technologies, Grand Island, NY, USA), and 10 μM SB431542 (Selleckchem.com, Houston, TX) where needed. After eight days in culture, explants were washed twice with PBS digested with StemPro Accutase cell dissociation reagent (Gibco-Life Technologies, Grand Island, NY, USA) for 30 min at 37 °C and prepared for FACS analysis as described above. For information regarding reagents used for suture explants refer to Supplementary Table 3 and Supplementary Table 4. Primers sequence and annealing temperature are described in Supplementary Table 5.

**cWnt activation and inhibition.** cWnt signaling activation and inhibition animal surgical procedures were conducted as previously described[13]. A skin incision was performed above the PF or SAG suture of anesthetized pN4 CD-1, *Axin2*[LacZ/+], or *ActinCre*[ERT2]:*Rainbow* mouse under the aid of a dissection stereo microscope Stemi 2000 (Zeiss, Oberkochen Germany). A 1.5 mm diameter Helistat collagen sponge (Integra LifeSciences, Plainsboro, NJ, USA) was either soaked in 150 ng of Wnt3a protein (R&D Systems, Minneapolis, MN, USA) and placed on the PF suture, or with a combination of 2 μg of Dkk-1 and 2 μg sFrp-1 (R&D Systems, Minneapolis, MN, USA) and placed on the SAG suture. PBS soaked collagen sponges were used as controls in both sutures. The incision was closed using a single 8-0 nylon suture and the animals were allowed to recover. This procedure was repeated every other day till pN8 or pN17 specifically, the incision was re-opened, the collagen sponge re-soaked with the corresponding factors. On day pN8 or pN17 animals were sacrificed and processed for histological or FACS analysis. For information regarding reagents used refer to Supplementary Table 4.

**Generation of the *Twist1*[+/−]:*Axin2*[LacZ/+] double transgenic mouse and screen procedure.** B6*Twist1*[+/−] (Stock# 2221) mice were purchased from Jackson Laboratory (Bar Harbor, Maine, USA) and *Axin2*[LacZ/+] mice on B6/CD-1 mixed background were obtained as previously described[13]. *Twist1*[+/−] male mice were crossed with *Axin2*[LacZ/+] female mice to generate *Twist1*[+/−]:*Axin2*[LacZ/+] double transgenic. For screen of bilateral or unilateral COR craniosynostosis in wild-type, *Twist1*[+/−] and *Twist1*[+/−]:*Axin2*[LacZ/+] mice, animals were sacrificed by CO$_2$ asphyxiation on the exact post-natal date determined by DOB. Entire cranial were dissected and a small portion of the tail harvested for genotyping as described above. Prior to genotyping, entire cranial were screened with the aid of a dissection microscope and scored by two independent investigators for complete bilateral or unilateral craniosynostosis ($n = 282$ mice screened). Screening was performed on mice ranging from day pN6 up to 6 months in age.

**Coronal suturectomy**. pN16-18 *Twist1*[+/−] mice (*n* = 2) were anesthetized by 2% isoflorane, shaved and disinfected prior to a longitudinal skin incision along the midline of the Cranial. The pericranium was removed and the synostosed coronal suture excised using a trephine drill with a 0.3 mm drill bit under constant irrigation and with meticulous care to avoid damaging the underlining dura mater. Preoperative microcomputed tomography (μCT) scans were used as a guide to identify the synostosed coronal suture for excision. Coronal suturectomies were treated with 1.5 mm diameter Helistat collagen sponge (Integra LifeSciences, Plainsboro, NJ, USA) seeded with ~3 × 10[5] skeletal stem/progenitor cells (see cell preparation for coronal suturectomy) resuspended in 2 μl of recombinant Wnt3a protein at a concentration of 100 ng/μl in PBS. Untreated and Wnt3a alone treated suturectomies were used as controls. The incision was closed using 8-0 nylon suture and the animals were allowed to recover. The extent of re-fusion of the excised coronal suture was assessed by μCT over a 14-week period and by histology 14-weeks postoperative. For information regarding reagents used refer to Supplementary Table 4.

**Microcomputed tomography (μCT) scanning and analysis**. Animals were anesthetized by 2% isoflurane prior to scanning. All scans were performed using a Bruker SkyScan 1276 at a resolution of 35 μm. μCT reconstructions were performed with NRecon software (Bruker, Billerica, MA, USA), and 3D solid volume images produced using CTVol (Bruker, Billeric, MA, USA)[36]. To assess the extent of healing, standardized region of interest was applied to the suturectomy area in μCT reconstructions from 1 biological replicates per condition and quantified using the measure tool in Photoshop (Adobe) as previously described[52]. Identification of bone tissue was based upon tissue densities via an automatic thresholding process in CTAn (Bruker, Billeric, MA, USA). Mean values were compared across POD 1, 2wk, 4wk, 6wk, 8wk, 10wk, 12wk, and 14wk time-points.

**Cell culture**. Skeletal Stem Cells were cultured in alpha-MEM GlutaMax supplemented with 10% fetal bovine 1% penicillin–streptomycin (Gibco-Life Technologies, Grand Island, NY, USA), and 0.1% ciprofloxacin HCl (bioWORLD, Dublin, OH, USA). Cells were incubated under low $O_2$ conditions (2% atmospheric oxygen, 7.5% $CO_2$) for 48 h and then moved to standard conditions (5% $CO_2$). For information regarding reagents used refer to Supplementary Table 3.

**Wnt3a proliferation assay**. The Click-iT EdU Imaging Kit (Invitrogen, Eugene, OR, USA) was used to assess the proliferation of skeletal stem/progenitor cells with Wnt3a treatment. Skeletal stem/progenitor cells from the three cranial sutures were isolated, pooled (as described above) and seeded at the density of 3 × 10[3] cells/well in a 24-well plate. Skeletal stem/progenitor cells were cultured in alpha-MEM GlutaMax supplemented with 10% fetal bovine serum, 1% penicillin–streptomycin (Gibco-Life Technologies, Grand Island, NY, USA), 0.1% ciprofloxacin HCl (bioWORLD, Dublin, OH, USA) and Wnt3a(50 ng/ml) (R&D Systems). Cells were incubated under low $O_2$ conditions (2% atmospheric oxygen, 7.5% $CO_2$) for 48 h and then moved to standard conditions (5% $CO_2$). Cells were incubated in EdU (10 μM) for 1.5 h under standard conditions. The Click-iT EdU Imaging Kit was used per the manufacturers protocol. The "Analyze Particle" tool on ImageJ imaging software was used for quantification of proliferating (GFP[+]) cells. The number of GFP[+], proliferating cells, as well as Dapi[+] nuclei was measured, and proliferation presented as a percentage of GFP[+] cells/Dapi[+] cells. For information regarding reagents used refer to Supplementary Tables 3 and 4.

**Cell preparation for coronal suturectomy**. Skeletal stem/progenitor cells were isolated from entire cranial and tibias from pN3-5 wild type (C57BL/6) or *EGFP*[+/−] mice by flow cytometry. Skeletal stem/progenitor cells were pelleted at 1350 rpm at 4 °C for 20 min, resuspended in growth media (as described above) supplemented with 50 ng/ml Wnt3a protein (R&D Systems, Minneapolis, MN, USA) and seeded at density of 3 × 10[5] cells/well into a 96-well plate. Cells were incubated under standard conditions (37 °C, 5% $CO_2$) overnight (8–12 h). The following morning, cells were lifted using StemPro Accutase Cell Dissociation Reagent (Gibco-Life Technologies, Grand Island, NY, USA), pelleted at 1350 rpm for 20 min at 4 °C. Pelleted cells were then resuspended in 2 μl of recombinant Wnt3a protein at concentration of 100 ng/μl in PBS for transplantation. For information regarding reagents used refer to Supplementary Tables 3 and 4.

**Differentiation assays**. Osteogenic and chondrogenic potential were evaluated using the Stem Pro osteogenesis or chondrogenesis differentiation kit (Gibco-Life Technologies, Grand Island, NY, USA). Freshly sorted skeletal stem/progenitor cells were seeded into a 96-well plate, pre-coated with 0.1% gelatin (EmbryoMax, Millipore, Burlington MA, USA) and seeded with a density of 3 × 10[3] cells/well (pooled sutures and growth plate) or 1 × 10[3] cells/well (sutures individually) for osteogenic differentiation assays. For chondrogenic differentiation assays, 1 × 10[5] cells/well (whole skull and growth plate) or 1 × 10[3] cells/well (sutures individually) were seeded into a 96-well plate. "Pooled sutures" were comprised of equal number (1 × 10[3] cells) of skeletal stem/progenitor cells from the PF, SAG, and COR sutures. After reaching confluency growth media was replaced by Stem Pro osteogenic or chondrogenic media according to the manufactures protocol and media was

changed every other day for 21 days (osteogenic) or 40 days (Chondrogenic). Osteogenic differentiation was assessed by alizarin red staining and chondrogenic by alcian blue staining as previously described[68,69]. For information regarding reagents used refer to Supplementary Table 3.

**Colony-forming units (CFUs) and self-renewal assay**. Colonies were assessed as previously described[31]. 500 freshly sorted skeletal stem/progenitor cells from pN3, CD-1 mice PF, SAG, and COR sutures and the GP were seeded into a 10 cm[2] plate pre-coated with 0.1% gelatin (EmbryoMax, Millipore, Burlington MA, USA) in alpha-MEM GlutaMax (supplemented with 10% fetal bovine serum 1% penicillin–streptomycin (Gibco-Life Technologies, Grand Island, NY, USA), and 0.1% ciprofloxacin HCl (bioWORLD, Dublin, OH, USA). Cells were incubated under low $O_2$ conditions (2% atmospheric oxygen, 7.5% $CO_2$) for two weeks. For evaluation of CFUs, colonies were stained with crystal violet. Cell colonies were photographed under an inverted microscope and colonies with >50 cells or more were counted. For evaluation for self-renewal ability, skeletal stem/progenitor cells were isolated by FACS and pooled from the PF, SAG, and COR sutures of pN3 CD-1 mice. 500 skeletal stem/progenitor cells were seeded into 10 cm[2] for two weeks as described above. After 2 weeks clones were lifted using Stem Pro Accutase (Gibco-Life Technologies, Grand Island, NY, USA) and skeletal stem/progenitor cells isolated by FACS and passaged onto 10 cm[2] plates as described above. For information regarding reagents used refer to Supplementary Table 3.

**Immunofluorescent staining for activated β-catenin**. Immunofluorescence was performed to evaluate activation of β-catenin signaling. Briefly, sagittal sections of COR sutures from wild-type, *Twist1*[+/−], *Axin2*[LacZ/+] and *Twist1*[+/−]:*Axin2*[LacZ/+] mice at day pN15 were allowed to equilibrate at room temp for 5 min followed by 2 × 5min washes in PBS and 20 min fixation in 4% PFA. After fixation, sections were washed for 3 × 5 min in PBST (PBS with 0.05% Tween-20) and blocked for 1 h at room temperature in 10% donkey serum in PBST. After blocking, sections were incubated overnight in a humid chamber at 4 °C with the primary antibody, Anti-Active-β-Catenin (EMD Millipore, Billerica, MA, USA, 05-665) at a 1:100 dilution in 2% donkey serum in PBST. Following incubation with the primary antibody, sections were washed for 5 × 5 min in PBST and incubated for 1 h in the secondary antibody, Goat anti-Mouse IgG (H + L), Alexa Fluor 488 (Thermo Fisher Scientific, Eugene, OR, USA, A-11001), at a 1:1000 dilution in 2% donkey serum. Following incubation with the secondary antibody, sections were washed 3 × 5min in PBST and 2 × 5 min in PBS. Sections were mounted with Fluoromount-G with DAPI and imaged using a LEICA TCS SP8 X confocal microscope (*n* = 2). Experiments were performed 3 independent times. To determine the nuclear localization of β-Catenin, EzColocalization[70] plug-in for ImageJ was used to map the localization of β-Catenin (GFP) to nuclei (Dapi). Results are presented as a heatmap demonstrating cells with colocalization of β-Catenin to the nucleolus. For information regarding reagents used refer to Supplementary Table 1.

**Statistics analysis and reproducibly**. Statistical significance was assigned for $P ≤ 0.05$ (*$P < 0.05$ to ****$P < 0.0001$ represent a significant difference). Statistical analysis was performed using an unpaired student's *t*-test with a two-tailed distribution and/or one-way analysis of variance (ANOVA). All statistical calculations were performed using the Prism 8 (GraphPad) software package. No statistical method was used to predetermine sample size. Figure panels show representative images of at least two independent experiments as indicated in the figure legend. Flow cytometry plots are representative of at least two individual experiments and up to ten. For all figures (n) indicate the number of animals used per each independent experiment, all experiments were performed at least three times unless otherwise indicated by the figure legend. For all figures data are presented as a percentage or absolute value. For all graphs data are presented as means or representative value with error bars representing ± the standard error.

**Reporting summary**. Further information on research design is available in the Nature Research Reporting Summary linked to this article.

## Data availability
All data to support the conclusions in this manuscript can be found in the figures. All source data for plots are available in the attached source data file and any other data can be requested from the corresponding authors. The mouse mm9 reference genome database was used for all RNA-seq experiments. Publicly available datasets were obtained from Gene Expression Omnibus under the following accession numbers; Debnath et al., *Nature* 2018 GSE106237 and Chan et al., *Cell* 2015 GSE64447. All RNA-seq and scRNA-seq data generated from this study can be accessed from the Gene Expressions Omnibus (http://www.ncbi.nlm.nih.gov/geo/) using accession number GSE138882. Source data are provided with this paper.

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

## Acknowledgements

The authors thank EY. Seo, CR. Duldulao and M. Lopez for their technical assistance and experimental support with flow cytometry experiments. We thank T. Doyle for his assistance with μCT scanning and analysis, supported by grant NIH S10 Grant 1S10OD02349701 to (T.D.), the Stanford Functional Genomics Facility (SFGF), Stanford Cell Science Imaging Facility (CSIF), and the Stanford Stem Cell Institute FACS Core. A special thank for the RNA-seq data curation and formal analysis to Dr. Ramesh V. Nair, Bioinformatics Director, Bioinformatics-as-a-Service (BaaS) under the Genetics Bioinformatics Service Center (GBSC), a Stanford School of Medicine service center operated by the Department of Genetics. We kindly also acknowledge the Diabetes Genomics Analysis Core (DGAC) and the grant #P30DK116074. This work was supported by National Institutes of Health (NIH) grants R01DE027323, R01DE026730, U24DE029463 to (M.T.L.), the Hagey Laboratory for Pediatric Regenerative Medicine, the Gunn/Olivier fund, the Johnson/Longaker fund. The authors wish to thank Dr. D. Sahar and Dr. B. Behr for their previous inspirational work to this study.

## Author contributions

S.M., N.Q., A.S., S.S., B.B., C.K.F.C., R.C.R., M.J., and D.C.W. performed experiments. S.M. generated figure panels 1c, 2c, 4a, S1a, S2d, S9a, and S9b; M.J. performed the bioinformatics analysis; N.Q., S.M., R.T., M.J., D.C.W., and M.T.L. wrote the manuscript. N.Q. supervised the project.

## Competing interests

The authors declare no competing interests.
