## [Peer Review File · Nature Communications]

Reviewers' comments:

Reviewer #1 (Remarks to the Author):

This manuscript claims to isolate "skeletal stem cells" (SSC) from different cranial sutures, based on a previous work published by the same group isolating such cells from tibia. However, I did not find either the isolation protocol used, or the characterisation of the cells obtained, provided convincing evidence that the authors had really isolated such cells, as opposed to a variety of cells at various stages of osteogenic or chondrogenic differentiation. Since the significance and impact of work described later in the manuscript is predicated on the bona fide isolation of SSC, this represents a serious flaw which undermines the conclusions of the work.

To elaborate, my first concern is that the protocol for isolating "SSC", which apparently follows the previous one (finally gating on CD51+ and CD105-) appears unconvincing. These cells are presented as being true SSC by line 2 of the results as a *fait accompli*, but I found no evidence presented that these cells actually have stem cell properties.

In Fig 1e, whereas the presumptive SSC in the GP group represent a distinct cell population, this is not seen for any of the sutures analysed. For the latter, there just appears to be a continuous "tail" from the bulk of the cells and it is not clear that the cells isolated are distinct from the continuum. Because the cut-off is essentially arbitrary this property makes it extremely difficult to quantify the true proportion of these supposed SSC in any of the experiments, making later conclusions based on these proportions (such as in Fig 3g, 4c and e) doubtful. Also, given these observations, in Fig 1c I do not understand why "percentage of P3" shown in Fig 1c is not consistently and substantially higher for GP than for the sutures.

A related major concern (perhaps not very surprising given the above) is that there are no data provided to support that these are actually stem cells, or what proportion of the population they might comprise. The single cell data presented in Fig. 2e-g indeed document that the populations isolated are quite heterogeneous and, based on several recognisable markers of cells committed to various lineages such as Sp7, Bglap, Bglap2, Sox9, Col9a1, a2, a3 etc, appear to be committed to various stages of differentiation. In summary I do not see evidence that the cells used are indeed a group of cells with specific functional characteristics of SSC.

As the remainder of the paper is reliant on having isolated such SSC, the impact of the remainder of the work is in my view substantially undermined by the conclusion that this is not the case. Whilst some of the results of suture manipulation are of interest, the experiments are not well enough controlled for specific conclusions to be reached. Many of the results obtained could be attributed to more non-specific effects on altering the balance of cell proliferation and differentiation. In the final experiment, use of two "treatments" together needs to be contrasted with each used separately, to understand the relative contributions (and potential synergism) of the two treatments.

Additional questions/issues for the authors to address:

There are a large number of grammatical errors and typos scattered around both the main text (including the abstract) and the Supplementary Methods and Tables. The senior authors need to thoroughly check the whole ms.

P7 :use of the word "sporadic" to distinguish non-syndromic from syndromic cases of craniosynostosis is very misleading, because an alternative meaning of this word is that the affected individual is the only one in the family, a situation that often arises in syndromic craniosynostosis.

P7: authors should check that they are using correct gene names. For example SMAD6 (human) and Smad6 (mouse) are correct, I-SMAD6 is not.

P9/Fig.5: Were the Twist1 and Axin2 mice on the same genetic background? In other words, did you exclude the possibly confounding effect of background in the amelioration of the craniosynostosis in the Twist1 mouse?

P12, referring to Fig 5d: "Remarkably, the reinstatement of the COR suture mesenchyme, largely composed of cells derived from Axin2LacZ/+ mice" Please explain: by definition, all cells will be doubly

heterozygous for Twist1 and Axin2lacZ. What is shown is that many cells are expressing LacZ (ie Axin2), This is described as a "significant contribution" in the Fig.5 legend but no controls are provided to attest to its significance.

Fig3a: The overlapping appearance of bones in this suture suggest that it is a coronal rather than a sagittal suture. Please check.

Finally I find the subtitle to this work "An Answer to Craniosynostosis" highly inappropriate. It implies a major breakthrough, which is not the case, and will only lead to false hopes and expectations on the part of any patients or relatives who get hold of this work when published.

Reviewer #2 (Remarks to the Author):

This is to elucidated behavior of skeletal stem cells (SSCs) during fusion of calvarial sutures. The authors previously reported that the posterior frontal (PF) suture in mice fuses in a physiologic condition through endochondral ossification and that SSCs in long bone growth plates can be isolated using cell surface identities such as 119-Ti32-Alpha+Thy-6C3-CD105-CS200+. As a next logical step, the authors hypothesize that the SSCs are also present in unfused sutures and become less abundant prior to their fusion. In this report, the authors demonstrate a dramatic reduction of SSCs in the physiologically fusing sutures (the PF suture) and pathologically fusing sutures (coronal sutures in Twist1 heterozygous mice). The authors demonstrate that increased canonical Wnt (cWnt) signaling can increase SSC population in the PF suture and results in more patency. This is also the case in coronal sutures in Twist1 heterozygous mice, since the authors show superimposition of Axin2 heterozygosity, an inhibitory gene for Wnt signaling, rescues coronal synostosis in Twist1 heterozygous mice. Lastly, the authors show implantation of SSCs with Wnt3A prevents from resynostosis after coronal suturectomy of Twist1 heterozygous mice.

This is a beautiful story to demonstrate presence of a similar type of SSCs that is critical for endochondral ossification in cranial sutures, which are surrounded by bones generated though intramembranous ossification. The authors successfully claim that balance of self-renewal and differentiation of the SSC population is critical for suture patency. It is striking that stem cell populations in each suture are very different, which potentially explains why only specific cranial sutures are affected by a given pathologic condition. Delay/prevention of resynostosis after suturectomy has a high clinical relevance. This reviewer, however, has a couple of questions to make the story clearer.

1. It is interesting to find a number of genes of which loss of function is known to cause craniosynostosis is down regulated in the PF suture (Fig 2b). Are these gene also downregulated in cells from the coronal suture of Twist1 het mice?
2. In Figure 3, the authors take an interesting approach to increase BMP signaling in the sagittal (SAG) suture by inhibiting TGFb signaling. Do authors see similar impacts on other sutures when they are cultured with the inhibitor? It is reported that increased BMP signaling results in the anterior frontal (AF) suture synostosis, thus one would expect suppression of TGFb signaling also has an impact on the AF suture.
3. Related to above, since both SAG and PF sutures are regarded as the midline sutures, genes shown in Fig 2b would also be changed in SB431542 treated SAG sutures.
4. In the section of "Cell Preparation for coronal suturectomy" in the supplemental method file, the authors describe that SSCs were isolated from entire calvaria and tibiae. Does it mean the SSCs used

for Fig 6 experiment are mixture of different origins? If this is the case, it would against a logic the authors try to establish in this story.

5. microCT images shown in Fig 6b bottom could be interpreted as a failure of damage healing rather than maintenance of suture patency due to much larger non-mineralized area than a normal suture. This notion is somewhat supported by histological image show in Fig 6d.

Reviewer #3 (Remarks to the Author):

In the manuscript by Menon et al., the authors explore the role of skeletal stem cells in craniosynostosis. The manuscript is well written and figures are of a good quality. However, I have several serious concerns about the study specified below. Additionally, I have not got access to the GSE data-set and, accordingly, do not review that part of the work.

Major comments

1. The authors explore CD45-, Ter119-, Tie2-, Thy1.1-, Thy1.2-, 6C3-, CD105-, Alpha V Integrin+ population of cells (as clarified in the method section) and name them skeletal stem cells (SSCs) based on the previous publication (Chan CKF et al., Cell 2015, 160:285-298). However, in that publication SSCs are defined as CD45- Ter119- Tie2- AlphaV+ Thy- 6C3- CD105- CD200+, whereas the alphaV+CD200- population is referred as pre-bone cartilage and stromal progenitors arising from alphaV+CD200+ stem cells (Chan CKF et al., Cell 2015, 160:285-298). Thus, I conclude that the manuscript is focused not on SSCs, but on a population of alphaV integrin positive multi-potent transient progenitors or a mixture of different sub-populations. Please clarify this issue.

2. The manuscript provides only a correlation between the number of integrinV alpha+ (below I call them CD51+, coding gene is ITGAV) cells and a suture fate. There is no causal relation proven in any single experiment. Accordingly, many claims presented throughout the manuscript are overstating and the title "Skeletal stem cells powering calvarial suture fate: An answer to craniosynostosis" is by far overstating taking the correlative nature of the manuscript into account.

3. Single cell transcriptomics provides a powerful instrument to get insight into the mechanisms regulating CD51+ cells in various sutures and well appreciated here. However, the analysis presented is very immature.

(i) There is no pathway analysis provided showing regulation of Wnt pathway in CD51+ cells, the pathway previously proposed by authors to explain fusion of a posterior-frontal (PT) suture, but not a coronal (CAR) or sagittal (SAG) sutures (Behr et al., Dev Biol 2010, 344:922-940).

(ii) The role of hedgehog pathway, previously implicated in suture physiology (Zhao et al., Nat Cell Biol 2015, 17:386-396), is also not explored.

(iii) Furthermore, cellular heterogeneity is not presented per suture, but as a pool of all cells including those from long bones, precluding comparison of different sutures and composition of their CD51+ population.

(iv) Finally, a comparison with recently published single cell data sets for other skeletal stem cells (Debnath et al., Nature 2018, 562:133-139; Tikhonova et al., Nature 2019, 569:222-228; Mizuhashi et al., Nature 2018, 563:254-258 with single cell data-set presented in Mizuhashi et al., JBMR 2019, 34:1387-1392) is also lacking.

Altogether, these limitations preclude considering presented data as a valuable part of the manuscript.

4. The only attempt to prove functional role of CD51+ cells is presented in Figure 6, but the authors compared combination of Wnt3a plus CD51+ cells versus untreated mice. Wnt3a alone is capable to prevent fusion of a suture (Behr et al., Dev Biol 2010, 344:922-940) and therefore the role of CD51+ cells in this experiment remains unanswered. I believe in this experiment the appropriate comparison would be Wnt3a vs Wnt3a plus CD51+ cells or all lin- cells vs CD51+ cells.

5. All the treatments and genetic manipulations employed are not-specific and may act via other cell populations. This applies to experiments with Twist1 and Axin2 gene manipulations as well as Wnt3a or SB431542 treatments. Accordingly, it is impossible to conclude if the gene/treatment affects CD51+ cells directly, via other cells or simply regulates ITGAV gene expression level.

6. The number of CD51+ cells is presented being normalized to their number at postnatal day 3. This normalization looks rather strange to me. It looks reasonable only for Figure 1C where aged dynamics is assessed, but in this case why it is 4-9% at day 3, but not 100%? For experiments presented in Figs. 4 and 5 such normalization is rather misleading and the number of CD51+ cells better be presented as a percentage of all lineage-negative cells in a suture. In Fig. 3 the number of CD51+ is not normalized at all. It is therefore unclear if the observed decrease in CD51+ cells in Twist1 +/- mice reflects overall decrease of cell number in the suture or reflects specific regulation of CD51+ cells.

7. The authors claim increased proliferation upon Wnt3a treatment and decreased proliferation upon blocking the pathway. This is based on clonal analysis with brainbow mouse strain (Fig. 4d and g). However, clonal size is not quantified and therefore conclusions are not supported. Furthermore, it is unclear how clones were defined. Utilization of non-specific Actin-CreER for clonal analysis also precludes concluding if the clone size reflects activity of CD51+ cells or any other cellular population in a suture. The authors interpreted the data as that "treatment (Wnt3a or Dkk-1) regulates proliferation within a suture". Similar conclusion was achieved by EdU labeling in their previous work (Behr et al., Dev Biol 2010, 344:922-940) and therefore experiments presented in Fig 4d and g provide no new insights.

8. The experiments with genetic combination of Twist1 heterozygous and Axin2-LacZ/+ mice are very interesting and abrogation of COR synostosis in double heterozygous mice is impressive. The authors attribute the observed phenotype to the increased activity of b-catenin pathway in Axin2-LacZ/+ mice. I believe that elevated activity of b-catenin pathway in Axin2-LacZ/+ has to be proven - elevated b-catenin activity would imply numerous pathologies but these mice are grossly normal, which rise a suspicion that their b-catenin signaling is rather normal.

Response to Reviewer#1

Dear Reviewer, we thank you very much for the time dedicated to review our manuscript. Your comments and suggestions are greatly appreciated.

Reviewer: 1. This manuscript claims to isolate “skeletal stem cells” (SSC) from different cranial sutures, based on a previous work published by the same group isolating such cells from tibia. However, I did not find either the isolation protocol used, or the characterisation of the cells obtained, provided convincing evidence that the authors had really isolated such cells, as opposed to a variety of cells at various stages of osteogenic or chondrogenic differentiation. Since the significance and impact of work described later in the manuscript is predicated on the bona fide isolation of SSC, this represents a serious flaw, which undermines the conclusions of the work. To elaborate, my first concern is that the protocol for isolating “SSC”, which apparently follows the previous one (finally gating on CD51+ and CD105-) appears unconvincing. These cells are presented as being true SSC by line 2 of the results as a fait accompli, but I found no evidence presented that these cells actually have stem cell properties.

Our answer: The experimental procedure used to purify SSCs is described under the Methods section, which is provided as supplementary data in the manuscript due to space constraint. In addition, we do also refer to two previous published papers describing the original purification procedure (see: Chan, C. K. *et al.* *Identification and specification of the mouse skeletal stem cell.* **Cell** 160, 285-298, (2015); Gulati, G. S. *et al.* *Isolation and functional assessment of mouse skeletal stem cell lineage.* **Nat Protoc** 13, 1294-1309, (2018). Moreover, the FACS step-wise purification procedure (gating strategy) of SSCs (finally gating on CD51+ CD200+ and CD105-) is also illustrated in the new **Supplementary Figure 1**. SSCs resident in the cranial sutures have been isolated using the same immunophenotypic profile and FACS program originally used to isolate SSCs from long bones (References above). We would like to remark that there is only one shared profile, which is currently in use to isolate mouse SSCs from several skeletal compartments (*e.g.* long bone, calvarial bone, calvarial suture, mandible) (in addition to the references above please see: Tevlin R. *et al.*, *Pharmacological Rescue of Diabetic Skeletal Stem Cell Niches.* **Sci Transl Med** 9, 1-23, 2017; *Mechanoresponsive stem cells acquire neural crest fate in jaw regeneration.* **Nature**, 563, 514-521, 2018; Jones R.E. *et al.*, *Skeletal Stem Cell-Schwann Cell Circuitry in Mandibular Repair* **Cell Rep**, 11, 2757-2766, 2019;); Murphy MP *et al.*, *Activation of skeletal stem cell in an injured osteoarthritic mouse model* (**Nature Med**, 2020 accepted for publication).

In our current study we show that SSCs isolated from calvarial suture share the same characteristics of SSCs isolated from different skeletal sources as previously described (above references). The new **Supplementary Figure 2** (previously **Supplementary Figure 1**) illustrates the multilineage fate specification ability of suture-derived SSCs to differentiate toward osteogenic and chondrogenic lineages as well their colony-forming (CFU) ability as previously reported by our group.

Taken together, these data confirm the identity of SSCs isolated from sutures as that of those previously described SSCs.

Reviewer: 2. In Fig 1e, whereas the presumptive SSC in the GP group represent a distinct cell population, this is not seen for any of the sutures analysed. For the latter, there just appears to be a continuous "tail" from the bulk of the cells and it is not clear that the cells isolated are distinct from the continuum. Because the cut-off is essentially arbitrary this property makes it extremely difficult to quantify the true proportion of these supposed SSC in any of the experiments, making later conclusions based on these proportions (such as in Fig 3g, 4c and e) doubtful. Also, given these observations, in Fig 1c I do not understand why "percentage of P3" shown in Fig 1c is not consistently and substantially higher for GP than for the sutures. A related major concern (perhaps not very surprising given the above) is that there are no data provided to support that these are actually stem cells, or what proportion of the population they might comprise.

Our answer: We understand the reviewer's question and therefore we would like to address it: The difference in appearance of the plots between the "GP" and the three calvarial sutures reflects differences in the cell density of SSCs isolated from each tissue region. The representative FACS plots are provided as density plots, and based on our experience, on average, 60 mice sutures would provide between 750-2000 (depending on each suture) SSCs whereas, 10 mouse growth plates would provide approximately 4000 SSCs. Ten mice were sufficient for the isolation of GP-SSCs. Therefore, if we would have used a larger a number of mice for sutures isolation of SSCs to compensate their under representation, a more distinct suture SSCs populations (similar to the plot of GP-SSCs) would have be present as well. We found the use of more than 60 mice/each time-point to be extremely expensive and most important time consuming in terms of collection and preservation of cell viability during the whole dissecting procedure.

The percentage of P3 is not higher in the GP because this is a normalized value to the total number of viable, non-hematopoietic cells isolated from the GP. If SSCs numbers were plotted (and not normalized) there would be a much higher number of SSCs isolated from the GP throughout this time course as the reviewer suggested.

We have isolated SSCs from sutures using exactly the FACS procedure as originally described by Chan, C. K. *et al. Identification and specification of the mouse skeletal stem cell. Cell* 160, 285-298, (2015). We have one shared profile, which is currently in use to isolate mouse SSCs from several sources (*e.g.* long bone, calvarial bone, calvarial suture, mandible, spinal column) Gate selection and cut off are not arbitrary and were determined based on the use of FMOs and unstained controls during the development of this current profile and were based on settings determined in the original publication from 2015.

We believe (and hope) that we have clarified and addressed the reviewer comments with our answers above.

In addition as mentioned above the data supporting that SSCs are stem cells are illustrated in the **new Supplementary Figure 2** (previously **Supplementary Figure 1**).

Reviewer: 3. The single cell data presented in Fig. 2e-g indeed document that the populations isolated are quite heterogeneous and, based on several recognisable markers of cells committed to various lineages such as Sp7, Bglap, Bglap2, Sox9, Col9a1, a2, a3 etc, appear to be committed to various stages of differentiation. In summary I do not see evidence that the cells used are indeed a group of cells with specific functional characteristics of SSC.

Our answer: The aim of performing scRNA analysis was indeed to unveil heterogeneity between SSCs isolated from sutures such as PF COR and SAG, which comprise tissues of different embryonic origin. As we stated in the **Discussion section**, calvarial sutures morphogenesis and architecture involve interaction of tissues derived from two developmental embryonic sources: neural crest and mesoderm. Therefore, heterogeneity is not an unexpected finding as these cells are residing in different niches influencing them through a diversity of autocrine/paracrine signaling. SSCs derived from different embryonic tissues origins cross-talk through autocrine/paracrine signaling and therefore feature heterogeneity.

An important aspect that should be remarked and kept in mind is that SSCs are postnatal skeletal stem cells committed to a skeletogenic lineage. SSCs are restricted to give rise to bone, cartilage but not fat (see **Appendix Figure 1, on page 7 of our response**, and Chan et al., **Cell** 160, 285-298, 2015), and display self-renewal/CFU ability as well (**as mentioned above and illustrated in Supplementary Figure 2**). Therefore, taken together, these properties define them as a skeletal stem cell.

Moreover, considering that these postnatal skeletal stem cells are committed toward chondrogenic and osteogenic lineage specification, the expression of genes such as Sp7, Sox 9, Col 9, Col2 is not unexpected because they control cell fate determination. The identified genes such as (e.g. Col2, Col9, Col14, Sp7, Mmp9, Panx3, Sox9, Bglap) are indeed known for their structural and functional roles in skeletal development and are expressed in other skeletal stem cells (and mesenchymal stem cells). The expression of these genes mark a skeletal stem cell, and have been previously described in different skeletal stem cells identified by other investigators (please see the **Appendix Figure. 2** illustrating a summary of selected genes proposed to enrich for mouse skeletal stem cells and multi-potent stromal cells.). Moreover, work by Greenblatt's group identified recently a periosteal stem cell mediating intramembranous bone formation and these cells, namely PSC, are transcriptionally defined by expression of Sox9, Sp7, Alpl, Runx2, Col2 (see **Extended data Figure 6, Panel b, from Debnath et al., Nature 562, 133-129, 2018**).

Reviewer: 4. As the remainder of the paper is reliant on having isolated such SSC, the impact of the remainder of the work is in my view substantially undermined by the conclusion that this is not the case. Whilst some of the results of suture manipulation are of interest, the experiments are not well enough controlled for specific conclusions to be reached. Many of the results obtained could be attributed to more non-specific effects on altering the balance of cell proliferation and differentiation. In the final experiment, use of two "treatments" together needs to be contrasted with each used separately, to understand the relative contributions (and potential synergism) of the two treatments.

Our answer: Thank you very much for this valuable comment/suggestion to test each treatment separately (Wnt3a alone and SSCs alone) to understand their relative contributions/potential synergism. We performed parallel experiments with groups of mice treated either with Wnt3a or SSCs alone but decided to not present due to space limitations. We also expected (to be sincere) that perhaps one of the reviewers would have asked for these experiments. Therefore, we are glad that the reviewer is giving us the opportunity to show these additional data, which can be found in the new **Supplementary Figure 11**.

Reviewer: 5. There are a large number of grammatical errors and typos scattered around both the main text (including the abstract) and the Supplementary Methods and Tables. The senior authors need to thoroughly check the whole ms.

Our answer: Thank you for kindly pointing this out to us, and I apologize. We have polished the manuscript and all grammatical errors and typos scattered in the Abstract and Supplementary Methods have been corrected. Corrections are highlighted in RED FONT.

Reviewer: 6. P7: use of the word "sporadic" to distinguish non-syndromic from syndromic cases of craniosynostosis is very misleading, because an alternative meaning of this word is that the affected individual is the only one in the family, a situation that often arises in syndromic craniosynostosis.

Our answer: Thank you for this comment, it is greatly appreciated, and indeed, reflective of the reviewer' outstanding and long-time expertise in human genetics and craniosynostosis!..... We have deleted the misleading word "sporadic" and instead used "non-syndromic".

Reviewer: 7. P7: authors should check that they are using correct gene names. For example SMAD6 (human) and Smad6 (mouse) are correct, I-SMAD6 is not.

Our answer: Thank you for pointing this out. We have accommodated the nomenclature using *SMAD6* and *Smad6* thoroughly the manuscript. Changes are in RED FONT.

Reviewer: 8. P9/Fig.5: Were the Twist1 and Axin2 mice on the same genetic background? In other words, did you exclude the possibly confounding effect of background in the amelioration of the craniosynostosis in the Twist1 mouse?

Our answer: This is a very good point, and we fully agree. *Twist1* and *Axin2* mice are not on the same genetic background: *Twist1*^{+/-} mice are on a B6 background and *Axin2* mice are on a mixed B6/CD-1 background as described (see please, under **Methods section, page 7**). Since we were aware of potential confounding effects due to the diversity of genetic background, therefore, when we started this experiment we also generated double mutant using mice with same genetic background: B6-*twist1* and B6-*axin2*. We observed same rescue of craniosynostosis also in these double mutant as observed in the other. We did choose to proceed for a large-scale breeding using *twist1* on B6 background (instead of mixed background) because this colony was largely expanded and ready for our experimental needs. To generate another large *twist1* colony either on a mixed background (B6/CD-1) or B6 *Axin2* colony, it would have significantly delayed our work. Of relevancy, the animal housing cost at Stanford is highly expensive, and therefore, expanding additional mouse-colonies on a mixed background would have been costly.

Reviewer: 9. P12, referring to Fig 5d: "Remarkably, the reinstatement of the COR suture mesenchyme, largely composed of cells derived from Axin2LacZ/+ mice" Please explain: by definition, all cells will be doubly heterozygous for Twist1 and Axin2lacZ. What is shown is that many cells are expressing LacZ (ie Axin2), This is described as a "significant contribution" in the Fig.5 legend but no controls are provided to attest to its significance.

Our answer: The X-gal positive cells observed in the suture mesenchyme of the double mutant COR suture are *Axin2* positive cells. We have removed the word *significant* and replaced with *robust* in the revised manuscript. **Figure 5 legend, Panel d**, reads as it follows: **d**, (Top panel) X-gal staining identifies a **robust presence** of cWnt-activated cells derived from *Axin2*^{LacZ/+} mouse within the mesenchyme of the patent COR suture in *Twist1*^{+/-}: *Axin2*^{LacZ/+} double mutant mice at day pN3, pN18, and 6 months postnatal. Moreover, we have also performed a side-by-side X-gal staining of COR sutures derived from both *Axin2*^{LacZ/+} and *Twist1*^{+/-}: *Axin2*^{LacZ/+} double mutant mice. Data gained from this comparative analysis show that in the COR suture of *Twist1*^{+/-}: *Axin2*^{LacZ/+} double mutant mice an endogenous active cWnt signaling niche is reconstituted, similar to that of *Axin2*^{LacZ/+} COR suture as revealed by a intense X-Gal staining. These data are illustrated in **Supplementary Figure 9**.

Reviewer: 10. Fig3a: The overlapping appearance of bones in this suture suggests that it is a coronal rather than a sagittal suture. Please check.

Our answer: Figure 3c, illustrates indeed a Sagittal suture the slight appearance of overlapping bone is due to the fact the suture was explanted at postnatal day 3 (pN3) and kept in culture for 8 days prior to being removed and embedded for histology. There was a variation in structure appearance due to time in culture and the lack of other in-vivo structures (like the brain) that help to maintain suture morphology/shape.

Reviewer: 11. Finally, I find the subtitle to this work “An Answer to Craniosynostosis” highly inappropriate. It implies a major breakthrough, which is not the case, and will only lead to false hopes and expectations on the part of any patients or relatives who get hold of this work when published.

Our answer: Our study, specific to the last part of it (**Figure 6** and **Supplementary Figure 11**), is a proof-of-concept introducing, in perspective, the idea/proposal of a potential skeletal stem cell-based therapy to treat craniosynostosis. We do apologize, for adopting this title for our manuscript. By doing that, we absolutely never meant to fuel false hopes and expectations to patients. We believe that the title could be slightly modified by adding a question mark at the end. Therefore, if the reviewer would be ‘kindly permissive’ with us, we would like changing the title as: *An Answer to Craniosynostosis?* such that it will not sound as a **bold statement**, but rather an open question. And a simple question never misleads or hurts anyone.....

Appendix Figure 1

Adipose-derived Stem Cells

Suture-SSCs

Legend: Adipogenic differentiation assay performed using Adipogenic differentiation medium (StemPro™ Adipogenesis Differentiation kit, Cat Number A1007001, Gibco, ThermoFisher) on adipose-derived stem cells (positive control as reference for the effectiveness of the adipogenic assay) and pool of Suture-SSCs. Oil red staining at day 12 identified lipid droplets in Adipose-derived Stem Cells as a result of their differentiation towards an adipogenic lineage. Conversely, suture-SSCs did not specify along the adipogenic lineage as already previously reported for growth plate (GP)-SSCs (Chan et al., **Cell** 160, 285-298, 2015).

[Redacted]

Response to Reviewer#2

Dear Reviewer, we thank you very much for the time dedicated to review our manuscript. Your comments and suggestions are greatly appreciated.

Reviewer: 1. It is interesting to find a number of genes of which loss of function is known to cause craniosynostosis is down regulated in the PF suture (Fig 2b). Are these gene also downregulated in cells from the coronal suture of Twist1 het mice?

Our answer: We thank you for pointing this out. Unfortunately, we don't know if the cluster of genes linked to syndromic craniosynostosis we saw downregulated only in PF suture-SSCs of wild type mice are also downregulated in COR suture-SSCs of *Twist* mice. However, we should consider that haploinsufficiency for *Twist1* gene is *per se* sufficient and uniquely responsible to determine craniosynostosis. All the genes found downregulated are responsible for other monogenic/syndromic craniosynostoses and it is unlikely that *Twist1* haploinsufficiency affects their expression. However, whether the transcription factor *Twist1* may or may not impact the expression of those genes found downregulated in PF suture-SSCs remains to be assessed.

Reviewer: 2. In Figure 3, the authors take an interesting approach to increase BMP signaling in the sagittal (SAG) suture by inhibiting TGFb signaling. Do authors see similar impacts on other sutures when they are cultured with the inhibitor? It is reported that increased BMP signaling results in the anterior frontal (AF) suture synostosis, thus one would expect suppression of TGFb signaling also has an impact on the AF suture.

Our answer: As mentioned above by the reviewer, we know as well the elegant work published by Mishina's group showing that increased BMP signaling leads to AF suture synostosis. We did not perform the experiment using the small molecule SB431542 on AF suture, because our goal was to "phenocopy" a non-syndromic Sagittal craniosynostosis (the most frequent occurring type of craniosynostosis). Therefore, taking advantage of a *SMAD6* "denovo" mutation causing SAG suture craniosynostosis published by Timberlake et al., during the time of our study, we decided to mimic this craniosynostosis by downregulating *Smad6* in SAG suture using SB431542. The investigation on the AF suture using our experimental approach (as mentioned by the reviewer) is indeed a valuable suggestion that we will keep in mind for our future studies.

However, based on the published literature, patients affected by *SMAD6* mutations can present either isolated sagittal (n = 113), metopic (n = 70) or combined sagittal and metopic craniosynostosis. In light of the reported literature and considering that the human metopic suture corresponds to the frontal suture in mouse (AF+PF

suture) we could predict that SB431542 treatment might cause fusion of the AF suture as well.

Reviewer: 3. Related to above, since both SAG and PF sutures are regarded as the midline sutures, genes shown in Fig 2b would also be changed in SB431542 treated SAG sutures.

Our answer: This would not necessarily be the case, and we can't predict it, since we have not performed the profile analysis of those genes in SB431542-treated SAG sutures. Just to remind the reviewer, the experiment was performed to downregulate *Smad6* in order to mimic the “*denovo*” mutation of *SMAD6* causing SAG suture craniosynostosis as published by Timberlake et al. The genes found downregulated in PF suture-SSCs are related to syndromic craniosynostosis, and their downregulation of course does not necessarily reflect the responsiveness of SSCs to an inhibition of TGF β signaling, but rather an intrinsic imprinting of PF-SSCs programmed to participate in suture fusion. We hypothesize that the unique gene-signature unveiled in PF-SSCs is probably under an epigenetic control. This is an interesting hypothesis that deserves to be investigated in the near future.

Moreover, it should be pointed out that bulk RNA-seq data shows that the expression level of *Smad6* is higher in PF-SSCs as compared to SAG-SSCs therefore, this observation would suggest that the unique downregulation in PF-SSCs of those genes linked to syndromic craniosynostosis (and not found in SAG-SSCs or COR-SSCs) does not correlate to decreased level of *Smad6* and therefore, inhibition of TGF β signaling in this context.

Reviewer: 4. In the section of “Cell Preparation for coronal suturectomy” in the supplemental method file, the authors describe that SSCs were isolated from entire calvaria and tibiae. Does it mean the SSCs used for Fig 6 experiment are mixture of different origins? If this is the case, it would be against a logic the authors try to establish in this story.

Our answer: We have initially tested SSCs isolated either from calvaria or tibia by transplanting them separately on a single mouse COR suturectomy, and no differences in the outcome were observed. Unfortunately, due to limitations in cell-yield and having the need to work on a large number of mice to isolate cells for each treatment group (in order to achieve statistical significance) as well to use freshly isolated cells, we were unable to perform the experiments with SSC-derived from one source only. We felt comfortable to combine SSCs derived from two different bone-sources, because they share the same FACS profile as well as fate lineages specification and CFU ability (see please **Supplementary Figure 1** and **Supplementary Figure 2**) and for tibia-SSCs (Chan, C. K. *et al. Identification and specification of the mouse skeletal stem cell. Cell* 160, 285-298, 2015). However, we would like to emphasize that the experiments illustrated in **Figure 6** and

Supplementary Figure 11 represent mainly a proof-of-concept. These data demonstrate the feasibility of a cell-based therapy as potential intervention to craniosynostosis.

Reviewer 5. microCT images shown in Fig 6b bottom could be interpreted as a failure of damage healing rather than maintenance of suture patency due to much larger non-mineralized area than a normal suture. This notion is somewhat supported by histological image show in Fig 6d.

Our answer: The current clinical intervention for craniosynostosis is a strip suturectomy in which the synostosis suture is excised, and the cranial vault reconstructed. This is accomplished by removing a large strip of bony tissue corresponding to the area of fused suture in order to prevent and/or delay the formation of bony tissue, thus avoiding the need for subsequent procedures. In our ablation experiments we have been assisted by a skilled plastic surgeon, which routinely performs suture ablation surgery (at Stanford University Packard Children Hospital) on children affected by craniosynostosis. He has defined the appropriate amount of bony tissue to remove. What is shown in **Figure 6d** is two well-defined osteogenic fronts separated by a suture mesenchyme, that can be better appreciated in Panel **e** and **f**. In these two panels we can observe GFP SSCs populating the suturectomy gap interspersed with resident cells (migrated probably from the underlying dura-mater) in a pseudo-suture mesenchyme-like appearing structure. We found the interaction between the transplanted GFP-SSCs and the endogenous cells to be very interesting, particularly in how they cooperate to rebuild a suture mesenchyme spanning between the osteogenic front thus, keeping them apart from each other.

Response to Reviewer#3

Dear Reviewer, we thank you very much for the time dedicated to review our manuscript. Your comments and suggestions are greatly appreciated.

Reviewer: 1. *The authors explore CD45-, Ter119-, Tie2-, Thy1.1-, Thy1.2-, 6C3-, CD105-, Alpha V Integrin+ population of cells (as clarified in the methods section) and name them skeletal stem cells (SSCs) based on the previous publication (Chan CKF et al., Cell 2015, 160:285-298). However, in that publication SSCs are defined as CD45-, Ter119-, Tie2-, Thy1.1-, Thy1.2-, 6C3-, CD105-, Alpha V Integrin+, CD200+, whereas the Alpha V+, CD200- population is referred as pre-bone cartilage and stromal progenitors arising from AlphaV+CD200+ stem cells (Chan CKF et al., Cell 2015, 160:285-298). Thus, I conclude that the manuscript is focused not on SSCs, but a population of alphaV integrin positive multi-potent transient progenitors or a mixture of different subpopulations. Please clarify this issue*

Our Answer: We do apologize for the apparent confusion about the nomenclature. We have isolated SSCs from sutures using exactly the FACS procedure as originally described by Chan, C. K. *et al. Identification and specification of the mouse skeletal stem cell. Cell* 160, 285-298, (2015); (we have only one shared profile, which is currently in use to isolate mouse SSCs from several sources (*e.g.* long bone, calvarial bone, calvarial suture, mandible). Dissociated cells from PF, COR, SAG sutures and GP were stained with fluorochrome-conjugated antibodies for Ter119, CD51, CD105, Thy1.1, Thy1.2, Tie2, CD45, 6C3 and CD200. CD200 was mistakenly left off the methods section and has been revised appropriately, (please see Methods section page 2, paragraph **Fluorescence assisted cell sorting (FACS)**). SSCs isolated from sutures share the same immunophenotype profile as that of SSCs previously described by our group in several published papers (Chan, C. K. *et al. Identification and specification of the mouse skeletal stem cell. Cell* 160, 285-298, 2015; Gulati, G. S. *et al. Isolation and functional assessment of mouse skeletal stem cell lineage. Nat Protoc* 13, 1294-1309, 2018; Tevlin R. *et al., Pharmacological Rescue of Diabetic Skeletal Stem Cell Niches. Sci Transl Med* 9, 1-23, 2017; *Mechanoresponsive stem cells acquire neural crest fate in jaw regeneration. Nature*, 563, 514-521, 2018; Jones R.E. *et al., Skeletal Stem Cell-Schwann Cell Circuitry in Mandibular Repair. Cell Rep*, 11, 2757-2766, 2019; Murphy MP *et al., Activation of endogenous skeletal stem cell in an injured osteoarthritic mouse model (Nature Med*, 2020, accepted for publication). To provide further clarification on our FACS isolation and gating method we have added in our revised manuscript a new **Supplementary Figure 1** illustrating the gating strategy to isolate SSCs.

Reviewer: **2.** *The manuscript provides only a correlation between the number of integrinV alpha+ (below I call them CD51+, coding gene is ITGAV) cells and a suture fate. There is no causal relation proven in any single experiment.. Accordingly, many claims presented throughout the manuscript are overstating and the title “Skeletal stem cells powering calvarial suture fate: An answer to craniosynostosis” is by far overstating taking the correlative nature of the manuscript into account.*

Our Answer: We believe that all the data presented in Figure 3, 4, 5 indicate a tight correlation between the decrease and/or increase in SSCs and suture fate. We show clearly that a decrease in SSCs parallels/correlates to suture fusion, *viceversa* increase in SSCs leads to suture patency.

It was not our intention to overstate the study by using the title: “Skeletal stem cells powering calvarial suture fate: An answer to craniosynostosis” but rather to pose a question. We have modify the title as it follows: “Skeletal stem cells powering calvarial suture fate: An answer to craniosynostosis”? Now, it is like asking a question without being bold or pretentious. We hope that the Reviewer would be “kindly permissive” giving us the opportunity to keep the title as it is. [:-),Thanks!].

Reviewer: **3.** *Single cell transcriptomics provides a powerful instrument to get insight into the mechanisms regulating CD51+ cells in various sutures and well appreciated here. However, the analysis presented is very immature. (i) There is no pathway analysis provided showing regulation of Wnt pathway in CD51+ cells, the pathway previously proposed by authors to explain fusion of a posterior-frontal (PT) suture, but not a coronal (CAR) or sagittal (SAG) futures (Behr et al., Dev Biol 2010, 344:922-940). (ii) The role of hedgehog pathway, previously implicated in suture physiology (Zhao et al., Nat Cell Biol 2015, 17:386-396), is also not explored. (iii) Furthermore, cellular heterogeneity is not presented per suture, but as a pool of all cells including those from long bones, precluding comparison of different sutures and composition of their CD51+ population. (iv) Finally, a comparison with recently published single cell data sets for other skeletal stem cells (Debnath et al., Nature 2018, 562:133-139; Tikhonova et al., Nature 2019, 569:222-228; Mizuhashi et al., Nature 2018, 563:254-258 with single cell data-set presented in Mizuhashi et al., JBMR 2019, 34:1387-1392) is also lacking.*

Our Answer: Our study focuses on two major points: 1). To provide evidence that a proper balance of skeletal stem cells is a crucial player in controlling patency and/or fusion of a cranial suture. 2). The relevant role of cWnt signaling in this process. In our previous study by Behr et al., we indeed demonstrated that a sustained endogenous active cWnt is a fundamental “prerequisite” for maintaining suture patency. Herein, we nail down our previous observation providing evidence that

cWnt signaling triggers an enhancement of SSCs population resident in the cranial sutures.

Geneset enrichment analysis (GSEA) data obtained from bulk-RNAseq (performed on postnatal pN3 mice) revealed no significant difference in the endogenous level/regulation of cWnt signaling between SSCs derived from different sutures (PF, COR and SAG). These data indicate that the endogenous activation of cWnt signaling in SSCs mirrors that of the suture mesenchyme of pN3 mice as we have previously observed using the *Axin2* transgenic, in which the Xgal staining is a read-out of endogenous activated signaling (*Behr et al., Dev Biol 2010, 344:922-940.*) Therefore, data gained from GSEA further support our choice to perform the bulk RNA-seq analysis at time postnatal 3 (pN3) to ensure that any difference unveiled by transcriptomic analysis would not reflect a significant differential degree in the extent of endogenous activation of cWnt signaling between the three sutures, but rather intrinsic characteristics of SSCs resident in different sutures (perhaps due to their different embryonic tissue origin). The GSEA data further validate and support what we stated in the Results paragraph on page 5, lines 3-5 as it follows: *“Day pN3 was chosen based on two features that all three sutures share at this timepoint: first, they are patent; and second, they exhibit a comparable degree of active endogenous cWnt signaling.”*

Another important aspect emerging from GSEA is that SSCs are endowed by low endogenous active cWnt signaling. This finding would suggest an absence of saturated activation of cWnt signaling that would make cells more responsive to the exogenous Wnt-stimuli applied. This is supported by our observation that when we culture SSCs *in vitro*, they promptly respond to exogenous added Wnt3a protein by proliferating highly, more than untreated cells. Of note, SSCs express frizzled (fzd) and Lrp5 receptors of cWnt signaling.

We would like to share with the reviewer the GSEA report illustrating a cWnt signaling profile in pN3 suture-derived SSCs. The data can be found as **Appendix Figure 1** and **Appendix Table 1** and **Table 2** attached below.

The transcriptomic analysis was performed on SSCs derived from each suture with the aim to highlight the degree of heterogeneity between SSCs resident in each of three cranial sutures formed by tissues of different embryonic origin as PF, COR and SAG sutures are.

Definitively, the transcriptomic analysis does not represent a major focus of this study. We are aware and indeed acknowledge the intellectual beauty of the several other papers mentioned by the reviewer. Unfortunately, due to space constraint we are not able to either discuss them or outline comparisons between our cells and those reported by other studies. However, we cited in the introduction section of our manuscript the works by Zhao et al. and Maruyama et al. (Reference 27 and 28) since they are the most closely related to our study.

In this context, we would like to anticipate that a comparison and relationship between our mSSCs and additional skeletal stem cells reported by other investigators should be accomplished in the near future. We have an independent

ongoing study focusing on a comparative analysis between mouse and human suture-SSCs. This ongoing study is exclusively centered on an in-depth transcriptomic and epigenetic profiling (omics profiling) of the different cranial suture SSC populations in mouse and humans and comparative analysis to others skeletal stem cells.

Reviewer: 4. The only attempt to prove functional role of CD51+ cells is presented in Figure 6, but the authors compared combination of Wnt3a plus CD51+ cells versus untreated mice. Wnt3a alone is capable to prevent fusion of a suture (Behr et al., Dev Biol 2010, 344:922-940) and therefore the role of CD51+ cells in this experiment remains unanswered. I believe in this experiment the appropriate comparison would be Wnt3a vs Wnt3a plus CD51+ cells or all lin- cells vs CD51+ cells.

Our Answer: Thank you very much for this valuable comment. We performed parallel experiments having also group of mice treated either with Wnt3a or SSCs, alone, but decided to not present due to space constraint. We also anticipated (to be sincere) that perhaps one of the reviewers would have asked. Therefore, we are glad that the reviewer is giving us the opportunity to show these additional data, which can be found in the new **Supplementary Figure 11**.

Additionally, we would like to remark that in our previous study (Behr et al., Dev Biol 2010, 344:922-940.) the experiment with Wnt3a treatment was performed in very different context, such as a patent PF suture (postnatal day 4), which later (without any treatment) would undergo to physiological fusion. In the current study we have performed the experiment in a “COR suture craniosynostotic context”, a suture already fused to which we apply Wnt3a upon its ablation. Our new results (**Supplementary Figure 11**) clearly demonstrated that Wnt3a alone in this context is not sufficient to prevent resynostosis upon COR suture ablation. These finding highlights novel aspects that definitively differ from those previously contemplated in Behr et al. work (Wnt3a treatment on non-fused PF suture). Importantly, we also show herein, that transplantation of SSCs alone is not sufficient to prevent resynostosis upon COR suture ablation. This strongly suggests that Wnt3a may trigger a proliferative response of SSCs and possibly of other neighboring cell populations, which in a concerted manner reconstitute a proper suture mesenchyme

Reviewer: 5. *All the treatments and genetic manipulations employed are not-specific and may act via other cell populations. This applies to experiments with Twist1 and Axin2 gene manipulations as well as Wnt3a or SB431542 treatments. Accordingly, it is impossible to conclude if the gene/treatment affects CD51+ cells directly, via other cells or simply regulates ITGAV gene expression level.*

Our Answer: Our FACS analyses clearly indicate that SSCs are responsive to each treatment we have performed, and their increase or decrease in percentage/number tightly correlate to patency or fusion of sutures. Of course, we expect (and can't rule out) that in addition to SSCs, other mesenchymal cells may respond to the stimuli we apply and therefore participate to the suture patency and/or fusion program. A clear suggestion of this emerges from earlier experiments performed by Behr showing that Wnt3a treatment expands markedly the PF suture mesenchyme which, of course is composed not only by "our identified SSCs".

We know that these skeletal stem cells are not universal and the "ONLY ONE" resident in the suture mesenchyme, there are others. For example, Gli1+ and Axin2 stem cell populations have also been identified in cranial sutures. However, our study is performed through the lens of this specific SSCs population and we believe that it highlights a "consistent" correlation between **activation of cWnt signaling-increased-SSCs-suture patency**; and conversely, **inhibition of cWnt-decreased-SSC-suture fusion**.

We don't believe that a treatment like Wnt3a simply targets the expression of ITGAV(CD51) and CD200 (instead of enhancing their proliferation) because we have always observed that SSCs cultured in presence of Wnt3a proliferate at higher rate than untreated cells. Furthermore, the concomitant occurrence of two independent events such as, up-regulation of ITGAV(CD51) and CD200 expression by Wnt3a protein is quite unlikely.

Additionally, our results illustrated in **Figure 3c-g** also provide good evidence of a correlation between **inhibition of TGF β signaling-decreased-SSCs-suture fusion**. Taken together, these finding demonstrate a direct effect of cWnt and TGF β signaling on SSCs.

Reviewer: 6. *The number of CD51+ cells is presented being normalized to their number at postnatal day 3. This normalization looks rather strange to me. It looks reasonable only for Figure 1C where aged dynamics is assessed, but in this case why it is 4-9% at day 3, but not 100%? For experiments presented in Figs. 4 and 5 such normalization is rather misleading and the number of CD51+ cells better be presented as a percentage of all lineage-negative cells in a suture. In Fig. 3 the number of CD51+ is not normalized at all. It is therefore unclear if the observed decrease in CD51+ cells in Twist1 +/- mice reflects overall decrease of cell number in the suture or reflects specific regulation of CD51+ cells.*

Our Answer: We apologize for the confusion, however as described either under the Materials section or Figure legend, **P3 does not refer to postnatal day 3 (pN3).**

P3 population represents viable, non-hematopoietic (or RBCs) cells isolated from each region (sutures or growth plate) and used to normalize to the number of SSCs isolated. **Percentage of P3** is the method we use for normalizing the representation of SSCs isolated from the three calvarial sutures and the growth plate. During the developmental timeline of our study there is a dynamic change in the size and structure of the calvarial sutures. To account for the differences in suture size and animal size over the time course and to provide a fair comparison to the growth plate, the number of SSCs isolated from each region (PF, COR, SAG or GP) were normalized to the P3 population in our FACS isolation scheme (described in methods - Fluorescence assisted cell sorting (FACS). In our isolation scheme after gating out debris and doublets (FSC vs. SSCs) the remaining population is fractionated for CD45/Ter119 vs Pi. Hematopoietic (CD45⁺/Ter119⁺) and dead cells (Pi⁺) were excluded and the remaining population, denoted P3, is further fractionated as described in **Supplementary Figure 1** and Materials section. Just to clarify one more time, P3 is not post-natal 3 (pN3), P3 is a population representing viable, non-hematopoietic (or RBCs) cells isolated from each suture and used to normalize to the number of SSCs isolated.

Reviewer: 7. *The authors claim increased proliferation upon Wnt3a treatment and decreased proliferation upon blocking the pathway. This is based on clonal analysis with brainbow mouse strain (Fig. 4d and g). However, clonal size is not quantified and therefore conclusions are not supported. Furthermore, it is unclear how clones were defined. Utilization of non-specific Actin-CreER for clonal analysis also precludes concluding if the clone size reflects activity of CD51+ cells or any other cellular population in a suture. The authors interpreted the data as that “treatment (Wnt3a or Dkk-1) regulates proliferation within a suture”. Similar conclusion was achieved by EdU labeling in their previous work (Behr et al., Dev Biol 2010, 344:922-940) and therefore experiments presented in Fig 4d and g provide no new insights.*

Our Answer: We have quantified the clonality in mice treated with Wnt3a and sFrp1 and Dkk1 inhibitors. These new data can be found in the new revised **Figure 4**. We used these mice to show difference in clonality and therefore proliferative

activity upon each treatment (Wnt3a and sFrp1+Dkk1). As illustrated in **Figure 4e** Wnt3a significantly increased the clonality whereas, sFrp1+Dkk1 treatment decreased the clonality **Figure 4i**. Additionally, we have also quantified the clonality in rainbow/*Twist1*^{+/-}, rainbow/*Axin2*^{lacZ/+} and rainbow wild-type mice **Supplementary Figure 7**. We would like to clarify that the aim of generating the transgenic rainbow/*Twist1*^{+/-} and rainbow/*Axin2*^{lacZ/+} mice was to highlighting differences in clonality and therefore an overall proliferative activity between the suture mesenchyme of a *Twist1*^{+/-} craniosynostotic COR suture and the COR suture mesenchyme of *Axin2*^{lacZ/+} mice that were the mice used to rescue normal COR suture phenotype in *Twist1*^{+/-}. **Supplementary Figure 7** illustrates a significant difference in clonality, with decreased clonality in COR suture of rainbow/*Twist1*^{+/-} as compared to COR suture of rainbow/*Axin2*^{lacZ/+} mice. Thus, the clonality profile of COR suture in these double transgenic mice mirrors that observed in PF and/or SAG suture of rainbow mice treated with Wnt3a and sFrp1+Dkk1 (**Figure 4e** and **4i**) respectively.

The procedure for quantification of clonality shown in **Figure 4e** and **4i** and **Supplementary Figure 7** is described under Methods.

Furthermore, we would like to remark that the generation of these rainbow mice was not meant to show a specific increase of SSCs as unfortunately, this is not feasible due to the lack of a mouse transgenic specific marker, which could allow a lineage tracing of SSCs. Therefore, we could only use the ubiquitous Actin-Cre^{ER} for clonal analysis of suture mesenchyme in fused versus patent suture, and/or upon activation and inhibition of cWnt signaling. Again, the scope of the rainbow double mutant mouse was to highlight differences in clonality between a COR craniosynostotic suture a normal phenotype COR suture through enhancing activation of cWnt signaling within the suture mesenchyme.

In our previous study (*Behr et al., Dev Biol 2010, 344:922-940*) Wnt3a was applied to a patent PF suture, which physiologically fuses, herein we have performed an experiment using a genetically double mutant and analyzed a craniosynostotic COR suture that otherwise would resynostose upon ablation. In our previous study we have shown that Wnt3 treatment enlarged the suture mesenchyme, suppressed the chondrogenic master regulator SOX9 leading to inhibition of chondrogenesis within the suture, ultimately preventing the endochondral ossification process though which the PF suture closes physiologically (please see: **Figure 4A-C, Behr et al., Dev Biol 2010, 344:922-940**). Here we show a “*de novo*” reconstruction of the suture mesenchyme by transplanting SSCs coached with Wnt3. Moreover, we demonstrated that Wnt3a alone is not sufficient in preventing resynostosis of a craniosynostotic COR suture upon its ablation (please see: **Supplementary Figure 11**). Therefore, we believe that the two experimental contexts as well as outcomes are quite different.

Reviewer: **8.** *The experiments with genetic combination of Twist1 heterozygous and Axin2-LacZ/+ mice are very interesting and abrogation of COR synostosis in double heterozygous mice is impressive. The authors attribute the observed phenotype to the increased activity of b-catenin pathway in Axin2-LacZ/+ mice. I believe that elevated activity of b-catenin pathway in Axin2-LacZ/+ has to be proven - elevated b-catenin activity would imply numerous pathologies but these mice are grossly normal, which rise a suspicion that their b-catenin signaling is rather normal.*

Our Answer: Immunofluorescence staining for activated β -catenin is higher in *Axin2^{lacZ/+}* COR suture than in wild-type COR suture thus, suggesting an increased activation of endogenous cWnt signaling in *Axin2^{lacZ/+}* COR suture. Importantly, staining for activated β -catenin performed on COR suture of *Twist1^{+/-}:Axin2^{lacZ/+}* reveals levels of active cWnt signaling similar to wild-type and *Axin2-LacZ/+* COR sutures. In contrast, activated β -catenin staining is very poor in *Twist1^{+/-}* COR sutures. These findings demonstrated high level of endogenous active cWnt signaling in the COR suture of double mutant resembling that of an unfused COR suture. These new data are presented in the new **Supplementary Figure 10**.

We would like to remark that an additional confirmation of substantial endogenous active cWnt signaling in the COR suture of double mutant mice is provided by the large number of X-gal staining observed in COR suture of these mice (**Figure 5d** and **Supplementary Figure 9**).

Targeted disruption of *Axin2* gene in mice induces malformations of skull structures only in homozygotic *Axin2^{-/-}* mice as reported in the article originally describing this transgenic mouse (Yu HM et al, 2005, *Dev Biol*). Moreover, our long-time experience working with these mice confirms this as well. In contrast, heterozygotic *Axin2^{+/-}* mice display a normal phenotype without any pathological features.

Appendix Figure 1

Appendix Figure 1 Legend. Gene Set Enrichment Analysis (GSEA) for the Canonical Wnt Pathway. Gene Set Enrichment Analysis was performed on bulk-RNA sequencing of SSCs isolated from pN3 PF, SAG and COR sutures. GSEA report for the cWnt pathway shows no significant enrichment at pN3 across the three sutures. Reported Nominal P-values are greater than 0.05 and False Discovery Rate (FDR) values greater than 0.30. See **Appendix Table 1** for report results., GSEA enrichment analysis for Canonical Wnt Signaling across the three sutures PF vs. SAG for SAG (**a**), PF vs COR for COR (**b**), COR vs SAG for COR (**c**). The bar-code indicates the position of each gene within the set; red and blue colors represent positive and negative Pearson correlations respectively. NOM P-Value, Nominal P-Value; FDR,

false discovery rate. **d**, Heat map of FPKM counts from cuffdiff analysis of Bulk-RNA sequencing of SSCs isolated from pN3 PF, SAG and COR sutures, demonstrates no significant difference in FPKM among the gene set identified by GSEA of the Canonical Wnt Pathway. Yellow: high counts; blue: low counts. Refer to **Appendix Table 2** for exactly values and statistical analysis.

Appendix Table 1

NAME	SIZE	ES	NES	NOM p-val	FDR q-val	FWER p-val	RANK AT MALEADING EDGE
PF vs SAG for SAG							
CANONICAL WNT SIGNALING I	20	-0.3382528	-0.8013068	0.8095238	1	1	4220 tags=30%, list=18%, signal=36%
PF vs COR for COR							
CANONICAL WNT SIGNALING I	18	-0.6204418	-1.4653069	0.06768559	0.3098134	1	5219 tags=56%, list=22%, signal=71%
COR vs SAG for SAG							
CANONICAL WNT SIGNALING I	20	0.5684354	1.3639884	0.08724833	0.5217169	1	4556 tags=45%, list=19%, signal=56%

Appendix Table 2

gene	sample_1	sample_2	status	value_1	value_2	log2(fold_change)	test_stat	p_value	q_value	significant
Apc	Cor-SSC	PF-SSC	OK	3.58613	1.5154	-1.24273	-1.3954	0.16575	0.999437	no
Axin1	Cor-SSC	PF-SSC	OK	17.9576	16.504	-0.121777	-0.171019	0.85745	0.999437	no
Cav1	Cor-SSC	PF-SSC	OK	79.3851	22.1647	-1.8406	-2.61634	0.0106	0.741529	no
Csnk1g1	Cor-SSC	PF-SSC	OK	3.15943	0.697207	-2.18001	-1.42783	0.1736	0.999437	no
Ctnnb1	Cor-SSC	PF-SSC	OK	263.262	174.127	-0.596362	-0.624235	0.5197	0.999437	no
Cul3	Cor-SSC	PF-SSC	OK	31.3487	11.2361	-1.48026	-2.01087	0.0498	0.999437	no
Dvl1	Cor-SSC	PF-SSC	OK	35.8943	25.335	-0.502623	-0.735922	0.45165	0.999437	no
Dvl2	Cor-SSC	PF-SSC	OK	13.5715	16.7256	0.301485	0.398099	0.6772	0.999437	no
Dvl3	Cor-SSC	PF-SSC	OK	15.5098	5.86499	-1.40298	-1.5904	0.1113	0.999437	no
Fzd5	Cor-SSC	PF-SSC	OK	1.24898	1.37611	0.13985	0.0970108	0.9287	0.999437	no
Gsk3a	Cor-SSC	PF-SSC	OK	21.3924	15.3305	-0.480692	-0.594524	0.55675	0.999437	no
Gsk3b	Cor-SSC	PF-SSC	OK	7.58337	4.64587	-0.706891	-0.947652	0.33375	0.999437	no
Klhl12	Cor-SSC	PF-SSC	OK	5.37203	2.84458	-0.917251	-0.771361	0.4356	0.999437	no
Lrp6	Cor-SSC	PF-SSC	OK	6.33434	11.2583	0.829726	1.21995	0.2233	0.999437	no
Nkd2	Cor-SSC	PF-SSC	OK	4.26733	3.39016	-0.331977	-0.28934	0.76655	0.999437	no
Pi4k2a	Cor-SSC	PF-SSC	OK	16.5009	15.841	-0.05888	-0.084547	0.93015	0.999437	no
Pip5k1b	Cor-SSC	PF-SSC	OK	0	2.03629	inf	#NAME?	0.05365	0.999437	no
Ppp2r5a	Cor-SSC	PF-SSC	OK	49.0987	17.9697	-1.45012	-1.83283	0.0551	0.999437	no
Ranbp3	Cor-SSC	PF-SSC	OK	40.2722	34.9575	-0.204179	-0.293822	0.76385	0.999437	no
Wnt3a	Cor-SSC	PF-SSC	NOTEST	0	0	0	0	1	1	no
Apc	Cor-SSC	Sag-SSC	OK	3.58613	3.27493	-0.130963	-0.173784	0.8568	0.999437	no
Axin1	Cor-SSC	Sag-SSC	OK	17.9576	16.6719	-0.107172	-0.153193	0.87115	0.999437	no
Cav1	Cor-SSC	Sag-SSC	OK	79.3851	9.11597	-3.1224	-3.84094	0.0005	0.14157	no
Csnk1g1	Cor-SSC	Sag-SSC	OK	3.15943	3.06606	-0.0432788	-0.0482765	0.96145	0.999437	no
Ctnnb1	Cor-SSC	Sag-SSC	OK	263.262	158.011	-0.736473	-0.764792	0.4347	0.999437	no
Cul3	Cor-SSC	Sag-SSC	OK	31.3487	35.7674	0.19024	0.284252	0.7743	0.999437	no
Dvl1	Cor-SSC	Sag-SSC	OK	35.8943	22.0485	-0.703076	-1.02812	0.2988	0.999437	no
Dvl2	Cor-SSC	Sag-SSC	OK	13.5715	17.8064	0.39182	0.529011	0.57575	0.999437	no
Dvl3	Cor-SSC	Sag-SSC	OK	15.5098	11.291	-0.458016	-0.58914	0.56275	0.999437	no
Fzd5	Cor-SSC	Sag-SSC	OK	1.24898	0.276249	-2.1767	-1.12047	0.28715	0.999437	no
Gsk3a	Cor-SSC	Sag-SSC	OK	21.3924	16.1521	-0.405373	-0.517169	0.5961	0.999437	no
Gsk3b	Cor-SSC	Sag-SSC	OK	7.58337	9.48996	0.323564	0.459416	0.6377	0.999437	no
Klhl12	Cor-SSC	Sag-SSC	OK	5.37203	6.44858	0.263514	0.264779	0.7934	0.999437	no
Lrp6	Cor-SSC	Sag-SSC	OK	6.33434	7.55588	0.254404	0.363834	0.7119	0.999437	no
Nkd2	Cor-SSC	Sag-SSC	OK	4.26733	3.15469	-0.435836	-0.376303	0.6974	0.999437	no
Pi4k2a	Cor-SSC	Sag-SSC	OK	16.5009	12.9618	-0.348283	-0.492161	0.61245	0.999437	no
Pip5k1b	Cor-SSC	Sag-SSC	OK	0	0.413808	inf	#NAME?	0.12225	0.999437	no
Ppp2r5a	Cor-SSC	Sag-SSC	OK	49.0987	19.7826	-1.31146	-1.67886	0.0796	0.999437	no
Ranbp3	Cor-SSC	Sag-SSC	OK	40.2722	41.1722	0.0318886	0.0459557	0.9613	0.999437	no
Wnt3a	Cor-SSC	Sag-SSC	NOTEST	0	0.0359574	inf	0	1	1	no
Apc	PF-SSC	Sag-SSC	OK	1.5154	3.27493	1.11177	1.22016	0.22505	0.999437	no
Axin1	PF-SSC	Sag-SSC	OK	16.504	16.6719	0.0146054	0.0204215	0.9832	0.999437	no

Response to Reviewer#3

Thank you for giving me an opportunity to read a revised version of the manuscript. The manuscript has been improved and clarified over the revision. However, I still have some concerns. Now authors clarified that they have employed the same surface antigens to isolate SSCs by FACS (Chan et al., Cell 2015) and it becomes clear that cells isolated with this protocol from different anatomical locations are substantially different in their transcriptional profile (Fig. 2 and Figs S24). They are also transcriptionally different from the originally characterized SSCs obtained from the mouse growth plate (Chan et al., Cell 2015). So, can we really be sure that they are skeletal stem cells? They do express similar surface antigens, but quite different in their transcriptional profile. Would it be enough relying only on the surface antigens to claim them being SSCs? For me this assumption does not look reliable without supporting functional tests.

Several of my previous concerns were also not addressed and remain to be valid (see below).

Dear Reviewer, we thank you very much for the time dedicated to review our manuscript. Your comments and suggestions are greatly appreciated.

Reviewer: 1. The authors explore CD45-, Ter119-, Tie2-, Thy1.1-, Thy1.2-, 6C3-, CD105-, Alpha V Integrin+ population of cells (as clarified in the methods section) and name them skeletal stem cells (SSCs) based on the previous publication (Chan CKF et al., Cell 2015, 160:285-298). However, in that publication SSCs are defined as CD45-, Ter119-, Tie2-, Thy1.1-, Thy1.2-, 6C3-, CD105-, Alpha V Integrin+, CD200+, whereas the Alpha V+, CD200- population is referred as pre-bone cartilage and stromal progenitors arising from AlphaV+CD200+ stem cells (Chan CKF et al., Cell 2015, 160:285-298). Thus, I conclude that the manuscript is focused not on SSCs, but a population of alphaV integrin positive multi-potent transient progenitors or a mixture of different subpopulations. Please clarify this issue

Our Answer: We do apologize for the apparent confusion about the nomenclature. We have isolated SSCs from sutures using exactly the FACS procedure as originally described by Chan, C. K. et al. Identification and specification of the mouse skeletal stem cell. Cell 160, 285-298, (2015); (we have only one shared profile, which is currently in use to isolate mouse SSCs from several sources (e.g. long bone, calvarial bone, calvarial suture, mandible). Dissociated cells from PF, COR, SAG sutures and GP were stained with fluorochrome-conjugated antibodies for Ter119, CD51, CD105, Thy1.1, Thy1.2, Tie2, CD45, 6C3 and CD200. CD200 was mistakenly left off the methods section and has been revised appropriately, (please see Methods section page 2, paragraph Fluorescence assisted cell sorting (FACS)). SSCs isolated from sutures share the same immunophenotype profile as that of SSCs previously described by our group in several published papers (Chan, C. K. et al. Identification and specification of the mouse skeletal stem cell. Cell 160, 285-298, 2015; Gulati, G. S. et al. Isolation and functional assessment of mouse skeletal stem cell lineage. Nat Protoc 13, 1294-1309, 2018; Tevlin R. et al., Pharmacological Rescue of Diabetic Skeletal Stem Cell Niches. Sci Transl Med 9, 1-23, 2017; Mechanoresponsive stem cells acquire neural crest fate in jaw regeneration. Nature, 563, 514-521, 2018; Jones R.E. et al., Skeletal Stem Cell-Schwann Cell Circuitry in Mandibular Repair. Cell Rep, 11, 2757-2766, 2019; Murphy MP et al., Activation of endogenous skeletal stem cell in an injured osteoarthritic mouse model (Nature Med, 2020, accepted for publication). To provide further clarification on our FACS isolation and gating method we have added in our revised manuscript a new Supplementary Figure 1 illustrating the gating strategy to isolate SSCs.

Thank you for the clarification. This makes the entire manuscript more exciting.

Reviewer: 2. The manuscript provides only a correlation between the number of integrinV alpha+ (below I call them CD51+, coding gene is ITGAV) cells and a suture fate. There is no causal relation proven in any single experiment.. Accordingly, many claims presented throughout the manuscript are overstating and the title “Skeletal stem cells powering calvarial suture fate: An answer to craniosynostosis” is by far overstating taking the correlative nature of the manuscript into account.

Our Answer: We believe that all the data presented in Figure 3, 4, 5 indicate a tight correlation between the decrease and/or increase in SSCs and suture fate. We show clearly that a decrease in SSCs parallels/correlates to suture fusion, viceversa increase in SSCs leads to suture patency.

I am glad that the authors agreed with my point of view that the nature of the manuscript is correlative, not causative (in relation to authors’ claim “*data presented in Figure 3, 4, 5 indicate a tight correlation*” I would believe that “*tight*” should be further supported by calculating correlation coefficient between the number of SSCs and the degree of fusion). Otherwise, I would keep suggesting softening the conclusions and claims throughout the manuscript (e.g., page 12, line 297, page 10, lines 260-262, page 9, line 228, etc.)

It was not our intention to overstate the study by using the title: “Skeletal stem cells powering calvarial suture fate: An answer to craniosynostosis” but rather to pose a question. We have modify the title as it follows: “Skeletal stem cells powering calvarial suture fate: An answer to craniosynostosis”? Now, it is like asking a question without being bold or pretentious. We hope that the Reviewer would be “kindly permissive” giving us the opportunity to keep the title as it is. [-:],Thanks!].

I do not think that adding a question mark makes the title sounding better, albeit a bit less provocative. However, I still have a problem with the claim in the title “SSCs powering sutures fate”, I do not see direct evidences for that, only correlations.

Reviewer: 3. Single cell transcriptomics provides a powerful instrument to get insight into the mechanisms regulating CD51+ cells in various sutures and well appreciated here. However, the analysis presented is very immature. (i) There is no pathway analysis provided showing regulation of Wnt pathway in CD51+ cells, the pathway previously proposed by authors to explain fusion of a posterior-frontal (PT) suture, but not a coronal (CAR) or sagittal (SAG) sutures (Behr et al., Dev Biol 2010, 344:922-940). (ii) The role of hedgehog pathway, previously implicated in suture physiology (Zhao et al., Nat Cell Biol 2015, 17:386-396), is also not explored. (iii) Furthermore, cellular heterogeneity is not presented per suture, but as a pool of all cells including those from long bones, precluding comparison of different sutures and composition of their CD51+ population. (iv) Finally, a comparison with recently published single cell data sets for other skeletal stem cells (Debnath et al., Nature 2018, 562:133-139; Tikhonova et al., Nature 2019, 569:222-228; Mizuhashi et al., Nature 2018, 563:254-258 with single cell dataset presented in Mizuhashi et al., JBRM 2019, 34:1387-1392) is also lacking.

Our Answer: Our study focuses on two major points: 1). To provide evidence that a proper balance of skeletal stem cells is a crucial player in controlling patency and/or fusion of a cranial suture. 2). The relevant role of cWnt signaling in this process. In our previous study by Behr et al., we indeed demonstrated that a sustained endogenous active cWnt is a fundamental “prerequisite” for maintaining suture patency. Herein, we nail down our previous observation providing evidence that cWnt signaling triggers an enhancement of SSCs population resident in the cranial sutures.

Geneset enrichment analysis (GSEA) data obtained from bulk-RNAseq (performed on postnatal pN3 mice) revealed no significant difference in the endogenous level/regulation of cWnt signaling between SSCs derived from different sutures (PF, COR and SAG). These data indicate that the endogenous activation of cWnt signaling in SSCs mirrors that of the suture mesenchyme of pN3 mice as we have previously observed using the Axin2 transgenic, in which the Xgal staining is a read-out of endogenous activated signaling (Behr et al., Dev Biol 2010, 344:922-940.) Therefore, data gained from GSEA further support our choice to perform the bulk RNA-seq analysis at time postnatal 3 (pN3) to ensure that any difference unveiled by transcriptomic analysis would not reflect a significant differential degree in the extent of endogenous activation of cWnt signaling between the three sutures, but rather intrinsic characteristics of SSCs resident in different sutures (perhaps due to their different embryonic tissue origin). The GSEA data further validate and support what we stated in the Results paragraph on page 5, lines 3-5 as it follows: “Day pN3 was chosen based on two features that all three sutures share at this

timepoint: first, they are patent; and second, they exhibit a comparable degree of active endogenous cWnt signaling.”

Another important aspect emerging from GSEA is that SSCs are endowed by low endogenous active cWnt signaling. This finding would suggest an absence of saturated activation of cWnt signaling that would make cells more responsive to the exogenous Wnt-stimuli applied. This is supported by our observation that when we culture SSCs in vitro, they promptly respond to exogenous added Wnt3a protein by proliferating highly, more than untreated cells. Of note, SSCs express frizzled (fzd) and Lrp5 receptors of cWnt signaling.

We would like to share with the reviewer the GSEA report illustrating a cWnt signaling profile in pN3 suture-derived SSCs. The data can be found as Appendix Figure 1 and Appendix Table 1 and Table 2 attached below.

The transcriptomic analysis was performed on SSCs derived from each suture with the aim to highlight the degree of heterogeneity between SSCs resident in each of three cranial sutures formed by tissues of different embryonic origin as PF, COR and SAG sutures are.

Definitively, the transcriptomic analysis does not represent a major focus of this study. We are aware and indeed acknowledge the intellectual beauty of the several other papers mentioned by the reviewer. Unfortunately, due to space constraint we are not able to either discuss them or outline comparisons between our cells and those reported by other studies. However, we cited in the introduction section of our manuscript the works by Zhao et al. and Maruyama et al. (Reference 27 and 28) since they are the most closely related to our study.

In this context, we would like to anticipate that a comparison and relationship between our mSSCs and additional skeletal stem cells reported by other investigators should be accomplished in the near future. We have an independent ongoing study focusing on a comparative analysis between mouse and human suture-SSCs. This ongoing study is exclusively centered on an in-depth transcriptomic and epigenetic profiling (omics profiling) of the different cranial suture SSC populations in mouse and humans and comparative analysis to others skeletal stem cells.

The authors did not address my concern – the data related to single cell transcriptomics are poorly analyzed and therefore are of a limited value. As far as I understand, single cell transcriptomes from 3 sutures + growth plate are pooled together in Fig. 2d,g,e (please correct me if I am wrong in interpreting figure legend and methodological description). From my point of view, it is important to present such data for each individual suture and the growth plate separately, i.e., principal component analysis for each suture separately and subsequent heat map per cluster, gene distribution and gene ontology categories enrichment for each identified cluster. This will allow comparing composition of the SSCs in each individual suture and may shed light on their differential behavior/regulation.

I think the authors contradict themselves saying “*Definitively, the transcriptomic analysis does not represent a major focus of this study.*” at the same time dedicating 1 major and 2 supplementary figures to this analysis. Furthermore, I believe the transcriptomic data are important for this study – they may provide an insight in the mechanism why some sutures stay patent while others fuse.

I agree with the authors that comparing their scRNA-seq data with other studies may go beyond the scope of the current manuscript albeit sutures were single-cell sequenced (i.e. in Debnath et al., Nature 2018, 562:133-139) and it is of high curiosity to compare. At the same time I disagree with the argument that the space is limited – the current data can be presented much more efficiently without losing information, i.e., Fig. 3d,e,f can be presented as a single panel, Figures S3 and S4 can be combined, S6 presented in text, etc.

I would also suggest adding GSEA analysis and clarifications about Wnt levels mentioned by the authors to the manuscript in order to decrease confusion among the readers about authors' current and previous works.

Reviewer: 4. The only attempt to prove functional role of CD51+ cells is presented in Figure 6, but the authors compared combination of Wnt3a plus CD51+ cells versus untreated mice. Wnt3a alone is capable to prevent fusion of a suture (Behr et al., Dev Biol 2010, 344:922-940) and therefore the role of CD51+ cells in this experiment remains unanswered. I believe in this experiment the appropriate comparison would be Wnt3a vs Wnt3a plus CD51+ cells or all lin- cells vs CD51+ cells.

Our Answer: Thank you very much for this valuable comment. We performed parallel experiments having also group of mice treated either with Wnt3a or SSCs, alone, but decided to not present due to space constraint. We also anticipated (to be sincere) that perhaps one of the reviewers would have asked. Therefore, we are glad that the reviewer is giving us the opportunity to show these additional data, which can be found in the new Supplementary Figure 11.

Additionally, we would like to remark that in our previous study (Behr et al., Dev Biol 2010, 344:922-940.) the experiment with Wnt3a treatment was performed in very different context, such as a patent PF suture (postnatal day 4), which later (without any treatment) would undergo to physiological fusion. In the current study we have performed the experiment in a "COR suture craniosynostotic context", a suture already fused to which we apply Wnt3a upon its ablation. Our new results (Supplementary Figure 11) clearly demonstrated that Wnt3a alone in this context is not sufficient to prevent resynostosis upon COR suture ablation. These finding highlights novel aspects that definitively differ from those previously contemplated in Behr et al. work (Wnt3a treatment on non-fused PF suture). Importantly, we also show herein, that transplantation of SSCs alone is not sufficient to prevent resynostosis upon COR suture ablation. This strongly suggests that Wnt3a may trigger a proliferative response of SSCs and possibly of other neighboring cell populations, which in a concerted manner reconstitute a proper suture mesenchyme

Thank you for addressing my concern. This is an important piece of missing information.

Reviewer: 5. All the treatments and genetic manipulations employed are not-specific and may act via other cell populations. This applies to experiments with Twist1 and Axin2 gene manipulations as well as Wnt3a or SB431542 treatments. Accordingly, it is impossible to conclude if the gene/treatment affects CD51+ cells directly, via other cells or simply regulates ITGAV gene expression level.

Our Answer: Our FACS analyses clearly indicate that SSCs are responsive to each treatment we have performed, and their increase or decrease in percentage/ number tightly correlate to patency or fusion of sutures. Of course, we expect (and can't rule out) that in addition to SSCs, other mesenchymal cells may respond to the stimuli we apply and therefore participate to the suture patency and/or fusion program. A clear suggestion of this emerges from earlier experiments performed by Behr showing that Wnt3a treatment expands markedly the PF suture mesenchyme which, of course is composed not only by "our identified SSCs".

We know that these skeletal stem cells are not universal and the "ONLY ONE" resident in the suture mesenchyme, there are others. For example, Gli1+ and Axin2 stem cell populations have also been identified in cranial sutures. However, our study is performed through the lens of this specific SSCs population and we believe that it highlights a "consistent" correlation between activation of cWnt signaling- increased-SSCs-suture patency; and conversely, inhibition of cWnt-decreased- SSC-suture fusion.

We don't believe that a treatment like Wnt3a simply targets the expression of ITGAV(CD51) and CD200 (instead of enhancing their proliferation) because we have always observed that SSCs cultured in presence of Wnt3a proliferate at higher rate than untreated cells. Furthermore, the concomitant occurrence of two independent events such as, up-regulation of ITGAV(CD51) and CD200 expression by Wnt3a protein is quite unlikely.

Additionally, our results illustrated in Figure 3c-g also provide good evidence of a correlation between inhibition of TGF β signaling-decreased-SSCs-suture fusion. Taken together, these findings demonstrate a direct effect of cWnt and TGF β signaling on SSCs.

Well, in any of the mentioned experiments it is still possible that Wnt (or Tgfb) acts on osteoblasts or adipocytes or any other cells in the surrounding tissues, which in turn signal to SSCs. That is what I mean under “indirect action via other cell types”. I think this possibility has to be discussed. The mentioned effect of Wnt3a on proliferation of cultured SSCs would provide direct evidence, but either it is not reflected in the manuscript or I missed it.

Reviewer: 6. The number of CD51+ cells is presented being normalized to their number at postnatal day 3. This normalization looks rather strange to me. It looks reasonable only for Figure 1C where aged dynamics is assessed, but in this case why is it 4-9% at day 3, but not 100%? For experiments presented in Figs. 4 and 5 such normalization is rather misleading and the number of CD51+ cells better be presented as a percentage of all lineage-negative cells in a suture. In Fig. 3 the number of CD51+ is not normalized at all. It is therefore unclear if the observed decrease in CD51+ cells in Twist1 +/- mice reflects overall decrease of cell number in the suture or reflects specific regulation of CD51+ cells.

Our Answer: We apologize for the confusion, however as described either under the Materials section or Figure legend, P3 does not refer to postnatal day 3 (pN3). P3 population represents viable, non-hematopoietic (or RBCs) cells isolated from each region (sutures or growth plate) and used to normalize to the number of SSCs isolated. Percentage of P3 is the method we use for normalizing the representation of SSCs isolated from the three calvarial sutures and the growth plate. During the developmental timeline of our study there is a dynamic change in the size and structure of the calvarial sutures. To account for the differences in suture size and animal size over the time course and to provide a fair comparison to the growth plate, the number of SSCs isolated from each region (PF, COR, SAG or GP) were normalized to the P3 population in our FACS isolation scheme (described in methods - Fluorescence assisted cell sorting (FACS)). In our isolation scheme after gating out debris and doublets (FSC vs. SSCs) the remaining population is fractionated for CD45/Ter119 vs Pi. Hematopoietic (CD45⁺/Ter119⁺) and dead cells (Pi⁺) were excluded and the remaining population, denoted P3, is further fractionated as described in Supplementary Figure 1 and Materials section. Just to clarify one more time, P3 is not post-natal 3 (pN3), P3 is a population representing viable, non-hematopoietic (or RBCs) cells isolated from each suture and used to normalize to the number of SSCs isolated.

Thank you for clarification and sorry for being confused between P3 and pN3. I believe other readers may also get confused and it would be good to clarify it more clearly throughout the manuscript. Along the same lines, are the data on Figs. 3b and 3g normalized or just reflect the total number? It is not clear from the figures or the legends.

Reviewer: 7. The authors claim increased proliferation upon Wnt3a treatment and decreased proliferation upon blocking the pathway. This is based on clonal analysis with brainbow mouse strain (Fig. 4d and g). However, clonal size is not quantified and therefore conclusions are not supported. Furthermore, it is unclear how clones were defined. Utilization of non-specific Actin-CreER for clonal analysis also precludes concluding if the clone size reflects activity of CD51+ cells or any other cellular population in a suture. The authors interpreted the data as that “treatment (Wnt3a or Dkk-1) regulates proliferation within a suture”. Similar conclusion was achieved by EdU labeling in their previous work (Behr et al., Dev Biol 2010, 344:922-940) and therefore experiments presented in Fig 4d and g provide no new insights.

Our Answer: We have quantified the clonality in mice treated with Wnt3a and sFrp1 and Dkk1 inhibitors. These new data can be found in the new revised Figure 4. We used these mice to show difference in clonality and therefore proliferative activity upon each treatment (Wnt3a and sFrp1+Dkk1). As illustrated in Figure 4e Wnt3a significantly increased the clonality whereas,

sFrp1+Dkk1 treatment decreased the clonality Figure 4i. Additionally, we have also quantified the clonality in rainbow/*Twist1*^{+/-}, rainbow/*Axin2*^{lacZ/+} and rainbow wild-type mice Supplementary Figure 7. We would like to clarify that the aim of generating the transgenic rainbow/*Twist1*^{+/-} and rainbow/*Axin2*^{lacZ/+} mice was to highlighting differences in clonality and therefore an overall proliferative activity between the suture mesenchyme of a *Twist1*^{+/-} craniosynostotic COR suture and the COR suture mesenchyme of *Axin2*^{lacZ/+} mice that were the mice used to rescue normal COR suture phenotype in *Twist1*^{+/-}. Supplementary Figure 7 illustrates a significant difference in clonality, with decreased clonality in COR suture of rainbow/*Twist1*^{+/-} as compared to COR suture of rainbow/*Axin2*^{lacZ/+} mice. Thus, the clonality profile of COR suture in these double transgenic mice mirrors that observed in PF and/or SAG suture of rainbow mice treated with Wnt3a and sFrp1+Dkk1 (Figure 4e and 4i) respectively.

The procedure for quantification of clonality shown in Figure 4e and 4i and Supplementary Figure 7 is described under Methods.

Furthermore, we would like to remark that the generation of these rainbow mice was not meant to show a specific increase of SSCs as unfortunately, this is not feasible due to the lack of a mouse transgenic specific marker, which could allow a lineage tracing of SSCs. Therefore, we could only use the ubiquitous Actin-Cre^{ER} for clonal analysis of suture mesenchyme in fused versus patent suture, and/or upon activation and inhibition of cWnt signaling. Again, the scope of the rainbow double mutant mouse was to highlight differences in clonality between a COR craniosynostotic suture a normal phenotype COR suture through enhancing activation of cWnt signaling within the suture mesenchyme.

In our previous study (Behr et al., Dev Biol 2010, 344:922-940) Wnt3a was applied to a patent PF suture, which physiologically fuses, herein we have performed an experiment using a genetically double mutant and analyzed a craniosynostotic COR suture that otherwise would resynostose upon ablation. In our previous study we have shown that Wnt3 treatment enlarged the suture mesenchyme, suppressed the chondrogenic master regulator SOX9 leading to inhibition of chondrogenesis within the suture, ultimately preventing the endochondral ossification process through which the PF suture closes physiologically (please see: Figure 4A-C, Behr et al., Dev Biol 2010, 344:922-940). Here we show a “de novo” reconstruction of the suture mesenchyme by transplanting SSCs coaxed with Wnt3. Moreover, we demonstrated that Wnt3a alone is not sufficient in preventing resynostosis of a craniosynostotic COR suture upon its ablation (please see: Supplementary Figure 11). Therefore, we believe that the two experimental contexts as well as outcomes are quite different.

I am convinced by the authors' argumentation that the data presented in this manuscript are different from the previous work as the use of differently behaving sutures may provide some new insights.

I am not satisfied by the data analysis and interpretation. First, it is clone size not the number of clones, which reflects cell proliferation. The latter reflects efficiency of labeling (if clones stay unbroken). However, the authors presented the number of clones instead (Fig. 4e,I, S7d). Second, for clonal definition the authors say: “Cells that were visually determined to be clones were traced...”. I think visual determination of clones being clones is not rigorous enough to be reproduced and requires somewhat more sophisticated definitions, e.g. mathematical.

There is likely a typo in the method, surface area of 3.2m² sounds quite much.

Reviewer: 8. The experiments with genetic combination of *Twist1* heterozygous and *Axin2*-LacZ/+ mice are very interesting and abrogation of COR synostosis in double heterozygous mice is impressive. The authors attribute the observed phenotype to the increased activity of b-catenin pathway in *Axin2*-LacZ/+ mice. I believe that elevated activity of b-catenin pathway in *Axin2*-LacZ/+ has to be proven - elevated b-catenin activity would imply numerous pathologies but these mice are grossly normal, which rise a suspicion that their b-catenin signaling is rather normal.

Our Answer: Immunofluorescence staining for activated β -catenin is higher in $Axin2^{lacZ/+}$ COR suture than in wild-type COR suture thus, suggesting an increased activation of endogenous cWnt signaling in $Axin2^{lacZ/+}$ COR suture. Importantly, staining for activated β -catenin performed on COR suture of $Twist1^{+/-};Axin2^{lacZ/+}$ reveals levels of active cWnt signaling similar to wild-type and $Axin2$ - $LacZ/+$ COR sutures. In contrast, activated β -catenin staining is very poor in $Twist1^{+/-}$ COR sutures. These finding demonstrated high level of endogenous active cWnt signaling in the COR suture of double mutant resembling that of an unfused COR suture. These new data are presented in the new Supplementary Figure 10.

We would like to remark that an additional confirmation of substantial endogenous active cWnt signaling in the COR suture of double mutant mice is provided by the large number of X-gal staining observed in COR suture of these mice (Figure 5d and Supplementary Figure 9).

Targeted disruption of $Axin2$ gene in mice induces malformations of skull structures only in homozygotic $Axin2^{-/-}$ mice as reported in the article originally describing this transgenic mouse (Yu HM et al, 2005, Dev Biol). Moreover, our long-time experience working with these mice confirms this as well. In contrast, heterozygotic $Axin2^{+/-}$ mice display a normal phenotype without any pathological features.

The staining of active β -catenin provided in the new Fig. S10 looks very much cytoplasmic. At the same time, upon activation β -catenin translocates to the nucleus and, accordingly, active β -catenin usually looks as a nucleus-localized signal. I think the staining provided is not reflecting β -catenin activity.

REVIEWER COMMENTS

Reviewer #1 (Remarks to the Author):

Whilst the authors wrote extensive rebuttals to all three referees, and clarified some details through explanations and 3 new supplementary figures (1, 10, 11), they have changed the text of their manuscript remarkably little. I have the sense that they have not really listened to the referees' significant scientific concerns. These go all the way back to the question of what cells they are really studying in this work.

I understand that these authors have previously produced excellent work defining a strategy to isolate putative skeletal stem cells from long bones, but this does not automatically mean that the same strategy will yield optimal populations of such cells from other tissues such as calvarial sutures. A particular concern is that the calvarial bones ossify by a different process from long bones. As raised by reviewer 3, in 2018 Debnath et al published work (Nature 562:133-139) specifically reporting that stem cells giving rise to periosteum (whether in long bones or calvarial sutures) are different from those yielding "endosteal osteoblasts". Reviewer 3 reasonably asked how "calvarial SSC" compared with the cells published by Debnath and in other papers, which the authors have not addressed at all in their rebuttal (in fact, remarkably, they failed to cite the Debnath paper altogether in the manuscript and use "space constraint" as an excuse to avoid the issue, despite the "intellectual beauty" of the work, and the fact that they quote data from that paper in the rebuttal to reviewer 1).

Given the reasonable concerns that the strategy for optimally isolating genuine stem cells from calvarial sutures might differ from that for long bones, the authors' attempts to characterise the quality and purity of the cell populations obtained from the sutures is remarkably cursory. In supplementary Fig 2, parts a and b show osteogenic and chondrogenic differentiation from "a pool of SSCs isolated from PF, SAG and COR sutures at day pN3" [it is not stated how many cells were used, this might be a property of only a few cells from pone of the sutures]; part c shows a primary CFU assay starting from 500 "SSCs", indicating that only between 10 and 35 of these are able to form colonies [a very low efficiency; why is the essential control using long bone growth-plate-derived cells missing from this?]; part d shows a illustrative example of successful secondary and tertiary colony formation, but provides no data on the efficiency of this process using the "SSC" isolated. In a response to reviewer 2, the authors state: "We felt comfortable to combine SSCs derived from two different bone-sources, because they share the same FACS profile as well as fate lineages specification and CFU ability". The appropriate level of evidence to support this statement has NOT been presented in the materials made available to the referees.

I also agree with reviewer 2's point that the assay shown in Fig 6 is looking at "a failure of damage healing rather than maintenance of suture patency". The extent of coronal suture that has been removed (as shown on post-operative day 1, Fig.6b) does not represent the full course of the suture at either the medial or lateral ends, therefore growth would continue to be mechanically locked between the frontal and parietal bones. The authors' response that they were assisted by a skilled plastic surgeon does not address this issue.

Unfortunately the authors did not heed the independent concerns of two reviewers about the overstatement in the title of the paper. This title represents pure journalism rather than a true scientific description conveying the content of this work. Moreover, they did not take the opportunity to improve the manuscript by incorporating other suggestions of the reviewers, for example by discussing the issue of genetic background of mice in the Twist1-Axin2 cross (reviewer 1) or insights from the analysis of single cell data (reviewer 3).

Additional comments:

Fig 1e, pN day3, GP, x-axis incorrectly labelled as CD15

Fig2f: the percentages in the diagrams add up to 1%. It seems that the use of proportions and percentages has been confused.

Supp Fig 11 legend: It is nowhere stated that this illustrates Twist1+/- mice

Reviewer #2 (Remarks to the Author):

The authors appropriately responded the critiques raised by this reviewer and others. Adding new results in supplemental figure 11 further reinforces their conclusion.

Reviewer #3 (Remarks to the Author):

Thank you for giving me an opportunity to read a revised version of the manuscript. The manuscript has been improved and clarified over the revision. However, I still have some concerns. Now authors clarified that they have employed the same surface antigens to isolate skeletal stem cells (SSCs) by FACS (Chan et al., Cell 2015) and it becomes clear that cells isolated with this protocol from different anatomical locations are substantially different in their transcriptional profile (Fig. 2 and Figs S4). They are also transcriptionally different from the originally (Chan et al., Cell 2015) characterised SSCs obtained from the mouse growth plate (Fig 2 and FigS4). So, can we really be sure that they are skeletal stem cells? They do express similar surface antigens, but quite different in their transcriptional profile. Would it be enough relying only on the surface antigens to claim them being SSCs? For me this assumption does not look reliable without supporting functional tests.

Reviewer #1 (Remarks to the Author):

Reviewer: Whilst the authors wrote extensive rebuttals to all three referees, and clarified some details through explanations and 3 new supplementary figures (1, 10, 11), they have changed the text of their manuscript remarkably little. I have the sense that they have not really listened to the referees' significant scientific concerns. These go all the way back to the question of what cells they are really studying in this work.

Our Answer: We agree that our initial revisions may not have addressed the scientific concerns raised by the reviewers as thoroughly as we would have liked. We hope that the significant changes made to our paper in this current iteration better reflect these concerns.

Reviewer: I understand that these authors have previously produced excellent work defining a strategy to isolate putative skeletal stem cells from long bones, but this does not automatically mean that the same strategy will yield optimal populations of such cells from other tissues such as calvarial sutures. A particular concern is that the calvarial bones ossify by a different process from long bones.

Our Answer: We thank the reviewer for their positive appraisal of our prior work. We would like to discuss four points in relation to the above concerns:

1. To address the reviewer's concern regarding the process of ossification of cranial sutures, we have added significant background text to our manuscript that cites prior work examining SSCs in the setting of mandibular distraction osteogenesis, where bone forms by intramembranous ossification, and in the setting of mandible fracture (endochondral) and long bone (endochondral) healing¹⁻⁴. All SSCs in these studies were isolated using the immunophenotype described by *Chan et al.*^{5,6} as we have done in this study.
2. In order to demonstrate the multipotent (osteogenic and chondrogenic) potential of SSCs, we have added **New Supplementary Figure 2a, b**. Here, we validated the osteogenic and chondrogenic potential of the SSCs isolated from both the growth plate and the PF, COR, and SAG cranial sutures independently as requested by the Reviewer.

3. Furthermore, we have improved the manuscript with the addition of **New Supplementary Figure 2c-d**, which demonstrates the capacity of the SSCs from the PF, COR, and SAG cranial sutures to self-renew in comparison to the GP by examining the colony forming unit potential of these cells.
4. Finally, while we did not expect the transcriptomes of SSCs derived from long bone and sutures to share transcriptional signatures given their differing sites of origin, function, and weight bearing status, we compared the transcriptomes of SSCs derived from PF, COR, and SAG sutures versus SSCs derived from long bone growth plates on a single cell level (**New Supplementary Figure 6a**). Data from this analysis indicated that PF and COR-SSCs clustered with long bone GP-SSCs.

Reviewer: 1. As raised by reviewer 3, in 2018 Debnath et al published work (Nature 562:133-139) specifically reporting that stem cells giving rise to periosteum (whether in long bones or calvarial sutures) are different from those yielding ‘endosteal osteoblasts’. Reviewer 3 reasonably asked how ‘calvarial SSC’ compared with the cells published by Debnath and in other papers, which the authors have not addressed at all in their rebuttal (in fact, remarkably, they failed to cite the Debnath paper altogether in the manuscript and use ‘space constraint’ as an excuse to avoid the issue, despite the ‘intellectual beauty’ of the work, and the fact that they quote data from that paper in the rebuttal to reviewer 1).

Our Answer: Following the recommendations of Reviewers 1 and 3, we have compared the transcriptomic (scRNA sequencing) data sets from the study by *Debnath et al*⁷ with our transcriptomic data sets. The resulting analysis is now included in the new **Supplementary Figure 6b**. In brief, we demonstrated that the transcriptomes of SSCs derived from the respective PF, SAG, and COR sutures shared features with those of the periosteal stem cells identified by *Debnath et al.*, with the highest degree of similarity among clusters with gene features linked to bone fate. The manuscript has also been updated to include citation of *Debnath et al.*

Reviewer: 2. Given the reasonable concerns that the strategy for optimally isolating genuine stem cells from calvarial sutures might differ from that for long bones, the authors' attempts to characterize the quality and purity of the cell populations obtained from the sutures is remarkably cursory.

Our Answer: We agree with the reviewer that this concern is reasonable and should be addressed. The revised manuscript has been updated to include a **new Supplementary Figure 2** in order to characterize both the quality and purity of these cell populations. These findings further validate the osteogenic and chondrogenic differentiation of the SSCs isolated from both the growth plate and cranial sutures.

Reviewer: In supplementary Fig 2, parts a and b show osteogenic and chondrogenic differentiation from 'a pool of SSCs isolated from PF, SAG and COR sutures at day pN3' [it is not stated how many cells were used, this might be a property of only a few cells from one of the sutures].

Our Answer: Thank you for this recommendation. **New Supplementary Figure 2a and 2b** now demonstrate the osteogenic and chondrogenic potential of SSCs isolated from the 3 sutures in comparison to the growth plate. For the osteogenic differentiation assay in **new Supplementary Figure 2a**, 3×10^3 SSCs were seeded/well and composed of equal numbers (1×10^3) of SSCs from each suture, and this is clarified in the accompanying figure legend.

Due to the scarcity of these cells, it would require sacrificing more than 60 pN3 pups to prospectively isolate $\sim 1 \times 10^3$ double sorted SSCs from each suture by FACS. For this reason we first utilized the entire skull for isolation of SSCs to perform a chondrogenic differentiation assay at an appropriate seeding density of 1×10^6 cells/well (**New Supplementary Figure 2a**). To assay the individual suture chondrogenic potential, we also performed chondrogenic differentiation assays for the individual sutures at a density of 1×10^3 cells/well as shown in **New Supplementary Figure 2b**. These details have been included in the revised **Methods**.

Reviewer: Part c shows a primary CFU assay starting from 500 ‘SSCs’, indicating that only between 10 and 35 of these are able to form colonies [a very low efficiency; why is the essential control using long bone growth-plate-derived cells missing from this?]

Our Answer: We agree with the reviewer that data from long bone growth-plate-derived cells would serve as an important control, and this has now been added in our **new Supplemental Figure 2c-d**, which demonstrates that SSCs isolated from the growth plate have similar CFU capacity to that of SSCs isolated from the PF and COR sutures but less than those isolated from the SAG suture. With regard to the colony-forming assay, the colony forming efficiency is in keeping with prior publications addressing stem cells, including those derived from cranial sutures.^{8,9}

Reviewer: Part d shows an illustrative example of successful secondary and tertiary colony formation but provides no data on the efficiency of this process using the ‘SSC’ isolated.

Our Answer: **New Supplemental Figure 2d** now provides SSC efficiency data which are also described in the revised in figure legend section.

Reviewer: 3. In a response to reviewer 2, the authors state: ‘We felt comfortable to combine SSCs derived from two different bone-sources, because they share the same FACS profile as well as fate lineages specification and CFU ability’. The appropriate level of evidence to support this statement has NOT been presented in the materials made available to the referees.

Our Answer: We thank the reviewer for their comment. We have now performed additional experiments to demonstrate the osteogenic and chondrogenic potential of SSCs isolated from both the growth plate and the PF, COR, and SAG cranial sutures independently as requested by the Reviewer. These results are shown in the new **Supplementary Figure 2a-b**. We have also added colony forming unit experiments demonstrating the capacity of SSCs from the PF, COR and SAG cranial sutures for self-renewal in a new **Supplementary Figure 2c-d**.

Reviewer: I also agree with reviewer 2's point that the assay shown in Fig 6 is looking at 'a failure of damage healing rather than maintenance of suture patency'. The extent of coronal suture that has been removed (as shown on post-operative day 1, Fig.6b) does not represent the full course of the suture at either the medial or lateral ends, therefore growth would continue to be mechanically locked between the frontal and parietal bones. The authors' response that they were assisted by a skilled plastic surgeon does not address this issue.

Our Answer: We thank the reviewer for highlighting this concern and agree that it is important to address, even though Reviewer 2 has now signed off. We did indeed previously attempt to perform a complete suturectomy in our animal model using the operating microscope with the assistance of a fellowship-trained Pediatric Microsurgeon and Craniofacial Surgeon. However at postnatal day 16-18, the animals were unfortunately unable to consistently survive this technically challenging procedure (likely due to complications related to sagittal sinus hemorrhage and /or thrombosis, dural injury and/or central nervous system infection) and thus, we had to modify our model to ensure feasibility. The modified model represents a proof of concept suggesting that SSCs coaxed with activator of Wnt can prevent bone formation or re-synostosis. From the clinical perspective, the goal is to maintain patency until normal development can occur rather than to perform a complete suturectomy and this is addressed by our model.

We observed re-synostosis (complete or near complete) in all of our controls groups (no treatment, Wnt3a treatment alone, SSCs treatment alone and sponge alone) but not in our group treated with Wnt3a and SSCs together. These data suggest that this model results in maintenance of suture patency, rather than failure of healing. Finally, the suturectomy defect measured ~3.1mm (length) x 0.72mm (width). In our prior work we noted that in juvenile mice, a cranial defect either of 3mm (non-critical defect) or 4/5mm (critical cranial defect) heals spontaneously whereas in adult mice (postnatal day 60 and older) a critical cranial defect does not.¹⁰ This defect could be considered analogous to a non-critical-size defect with the potential to regenerate.

Reviewer: 4. Unfortunately, the authors did not heed the independent concerns of two reviewers about the overstatement in the title of the paper. This title represents pure journalism rather than a true scientific description conveying the content of this work.

Our Answer: We thank the reviewer for their suggestion: The Title has been changed to one that is more appropriate to the conclusions made in the manuscript:

**Skeletal Stem Cells Maintain Cranial Suture Patency and Prevent Pathologic Fusion in
Mouse Models of Craniosynostosis**

Reviewer: 5. Moreover, they did not take the opportunity to improve the manuscript by incorporating other suggestions of the reviewers, for example by discussing the issue of genetic background of mice in the Twist1-Axin2 cross (reviewer 1) or insights from the analysis of single cell data (reviewer 3).

Our Answer: We apologize for this omission. The issue of mouse genetic background is now discussed in the edited figure legend of **Supplemental Figure 9** and accompanying Results. The Methods have also been edited to address this concern.

Additionally, we have added significantly greater detail to the analysis of our single cell RNA-seq data. As requested by the reviewers, we compared the transcriptome of the “periosteal stem cell” as characterized by *Debnath et al*⁷ with the SSCs derived from each of the sagittal (SAG), posterior frontal (PF), and coronal (COR) cranial sutures at a single cell level. Interestingly, we illustrated that the transcriptome of SSCs derived from the respective SAG, PF and COR shared transcriptional signatures with the “periosteal stem cell” characterized by Debnath and colleagues at a single cell level. These new data are shown in **New Supplementary Figure 6b**.

We also attempted to compare the transcriptomes of the SSCs derived from long bone, as published by *Chan et al*^{5,6} with the SSCs derived from our sutures. Unfortunately, this was not successful due to multiple factors.

- First, the Chan scRNA sequencing data were obtained from a small number of cells (n=37). At this limited number, most well-validated pre-integration scaling approaches are unstable, and we found that these data failed both anchor transform-based integration as well as traditional scaling (in the absence of significant parameter optimization to the point where the comparison could no longer reasonably be considered unbiased).
- The second limitation of this comparison is that the scRNA sequencing technique described in the study by *Chan et al* was novel at the time but is technically limited in comparison to current methods, including those used in the present study, further compounding the “batch effect” correction problem described above.
- The third limitation is that the *Chan et al* scRNA sequencing data were derived from the parent “SSC population”, which the authors were subsequently able to further separate into SSCs and downstream progenitors (the so-called pre-BCSP), further muddling the above comparison. Thus, in order to address the reviewer’s concerns, we compared the transcriptomes of SSCs isolated from murine long bone (using the methods and immunophenotype as described by *Chan et al*) with those of suture-derived SSCs. These SSCs were specifically isolated from the long bone growth plate (rather than the entire long bone), as it plays an analogous role to the suture mesenchyme in the lower extremity. These data are now shown in the new **Supplementary Figure 6a**.

Reviewer: 6. Additional comments:

- Fig 1e, pN day3, GP, x-axis incorrectly labelled as CD15 - Thank you for this correction. We have now correctly labeled the figure x-axis to read CD51 as shown in updated **Figure 1e**.
- Fig2f: the percentages in the diagrams add up to 1%. It seems that the use of proportions and percentages has been confused - Thank you for this correction. **Figure 2F** has now been revised in full: Pie charts in panel F represent the number of cells from each suture that comprise that cluster.
- Supp Fig 11 legend: It is nowhere stated that this illustrates Twist1+/- mice - Thank you for this correction. The legend for **Supplementary Figure 12** (previously Supplementary Figure 11) has now been updated to include a description of the Twist1+/- mice.

References:

- 1 Ransom, R. C. *et al.* Mechanoresponsive stem cells acquire neural crest fate in jaw regeneration. *Nature* **563**, 514-521, doi:10.1038/s41586-018-0650-9 (2018).
- 2 Jones, R. E. *et al.* Skeletal Stem Cell-Schwann Cell Circuitry in Mandibular Repair. *Cell Rep* **28**, 2757-2766 e2755, doi:10.1016/j.celrep.2019.08.021 (2019).
- 3 Tevlin, R. *et al.* Pharmacological rescue of diabetic skeletal stem cell niches. *Sci Transl Med* **9**, doi:10.1126/scitranslmed.aag2809 (2017).
- 4 Murphy, M. P. *et al.* Articular cartilage regeneration by activated skeletal stem cells. *Nat Med*, doi:10.1038/s41591-020-1013-2 (2020).
- 5 Chan, C. K. *et al.* Identification and specification of the mouse skeletal stem cell. *Cell* **160**, 285-298, doi:10.1016/j.cell.2014.12.002 (2015).
- 6 Gulati, G. S. *et al.* Isolation and functional assessment of mouse skeletal stem cell lineage. *Nat Protoc* **13**, 1294-1309, doi:10.1038/nprot.2018.041 (2018).
- 7 Debnath, S. *et al.* Discovery of a periosteal stem cell mediating intramembranous bone formation. *Nature* **562**, 133-139, doi:10.1038/s41586-018-0554-8 (2018).
- 8 Zhao, H. *et al.* The suture provides a niche for mesenchymal stem cells of craniofacial bones. *Nat Cell Biol* **17**, 386-396, doi:10.1038/ncb3139 (2015).
- 9 Yang, Z. *et al.* CD49f Acts as an Inflammation Sensor to Regulate Differentiation, Adhesion, and Migration of Human Mesenchymal Stem Cells. *Stem cells (Dayton, Ohio)* **33**, 2798-2810, doi:10.1002/stem.2063 (2015).
- 10 Aalami, O. O. *et al.* Applications of a mouse model of calvarial healing: differences in regenerative abilities of juveniles and adults. *Plastic and reconstructive surgery* **114**, 713-720, doi:10.1097/01.prs.0000131016.12754.30 (2004).

Reviewer #2 (Remarks to the Author):

The authors appropriately responded the critiques raised by this reviewer and others. Adding new results in supplemental figure 11 further reinforces their conclusion.

The Authors thank reviewer 2 for their thoughtful comments and contribution to this Manuscript.

Reviewer #3 (Remarks to the Author):

Reviewer: Thank you for giving me an opportunity to read a revised version of the manuscript. The manuscript has been improved and clarified over the revision. However, I still have some concerns.

Our Answer: We thank the reviewer for their comments and are pleased that the reviewer comments that our manuscript has been improved and clarified compared to the initial submission. Please find your concerns addressed in full below.

Reviewer: 1. Now authors clarified that they have employed the same surface antigens to isolate skeletal stem cells (SSCs) by FACS (*Chan et al.*, Cell 2015) and it becomes clear that cells isolated with this protocol from different anatomical locations are substantially different in their transcriptional profile (Fig. 2 and Figs S4). They are also transcriptionally different from the originally (*Chan et al.*, Cell 2015) characterized SSCs obtained from the mouse growth plate (Fig 2 and FigS4).

- So, can we really be sure that they are skeletal stem cells?
- They do express similar surface antigens, but quite different in their transcriptional profile. Would it be enough relying only on the surface antigens to claim them being SSCs? For me this assumption does not look reliable without supporting functional tests.

Our Answer: We thank the reviewer for these comments. We have provided further key data supporting the stem phenotype of these cells and would like to discuss four points in relation to your concerns.

We thank the reviewer for their positive appraisal of our prior work. We would like to discuss four points in relation to the above concerns:

1. To address the reviewer's concern regarding the transcriptional variation among SSCs isolated from various anatomical locations, we have added significant background text to our manuscript that cites prior work examining SSCs in the setting of SSCs from long bone^{1,2}, long bone fractures^{3,4}, and mandibles.^{5,6} All SSCs in these studies were isolated using the immunophenotype described by *Chan et al.*^{1,2} as performed in this study.

2. In order to demonstrate the multipotent (osteogenic and chondrogenic) potential of SSCs, we have added **New Supplementary Figure 2a, b**. Here, we validated the osteogenic and chondrogenic potential of the SSCs isolated from both the growth plate and the PF, COR, and SAG cranial sutures independently as requested by the Reviewer.
3. Furthermore, we have improved the manuscript with the addition of **New Supplementary Figure 2c-d**, which demonstrates the capacity of the SSCs from the PF, COR, and SAG cranial sutures to self-renew by examining the colony forming unit potential of these cells in comparison to SSCs from the GP.
4. Finally, while we did not expect the transcriptomes of SSCs derived from long bone and sutures to share transcriptional signatures given their differing sites of origin, function, and weight bearing status, we compared the transcriptomes of SSCs derived from PF, COR, and SAG sutures versus SSCs derived from long bone growth plates on a single cell level (**New Supplementary Figure 6a**). Data from this analysis indicated that PF and COR-SSCs clustered with long bone GP-SSCs.

Reviewer: 2. I am glad that the authors agreed with my point of view that the nature of the manuscript is correlative, not causative (in relation to authors' claim "*data presented in Figure 3, 4, 5 indicate a tight correlation*" I would believe that "*tight*" should be further supported by calculating correlation coefficient between the number of SSCs and the degree of fusion). Otherwise, I would keep suggesting softening the conclusions and claims throughout the manuscript (e.g., page 12, line 297, page 10, lines 260-262, page 9, line 228, etc.)

Our Answer: We agree with the reviewer, and each of those statements has been softened accordingly.

Reviewer: 2. I do not think that adding a question mark makes the title sounding better, albeit a bit less provocative. However, I still have a problem with the claim in the title “SSCs powering sutures fate”, I do not see direct evidences for that, only correlations.

Our Answer: We agree with this assessment and have changed the title to:

**Skeletal Stem Cells Maintain Cranial Suture Patency and Prevent Pathologic Fusion in
Mouse Models of Craniosynostosis**

Reviewer: 3. The authors did not address my concern – the data related to single cell transcriptomics are poorly analyzed and therefore are of a limited value. As far as I understand, single cell transcriptomes from 3 sutures + growth plate are pooled together in Fig. 2d,g,e (please correct me if I am wrong in interpreting figure legend and methodological description). From my point of view, it is important to present such data for each individual suture and the growth plate separately, i.e., principal component analysis for each suture separately and subsequent heat map per cluster, gene distribution and gene ontology categories enrichment for each identified cluster. This will allow comparing composition of the SSCs in each individual suture and may shed light on their differential behavior/regulation.

I think the authors contradict themselves saying “*Definitively, the transcriptomic analysis does not represent a major focus of this study.*” at the same time dedicating 1 major and 2 supplementary figures to this analysis. Furthermore, I believe the transcriptomic data are important for this study – they may provide an insight into the mechanism why some sutures stay patent while others fuse.

Our Answer: We thank the reviewer for this comment. Taking the reviewer’s comments into consideration, we strengthened our authorship team with the addition of a new co-author with expertise in scRNA transcriptomics to optimize our analyses. We have completely re-analyzed our single cell RNA-seq data incorporating the reviewer’s suggestions as shown in the revised **Figure 2** and three new supplementary figures. A brief description of each new analysis is described below:

- **Revised Figure 2d-e** provides new analysis of scRNA-seq data from the three cranial sutures (independently of the long bone growth plate cells), both in aggregate and also evaluated individually.
- **New Supplementary Figure 4** compares PCA analysis of gene features, violin plots, gene ontology and pathway analysis of the clusters derived from the cranial suture SSCs when analyzed together.
- **New Supplementary Figure 5** demonstrates the PCA plots, heat maps, pathway analyses and gene ontology enrichment for the PF, SAG, COR and growth plate SSCs analyzed separately.
- **New Supplementary Figure 6a** demonstrates the comparison of the PF, SAG and COR SSCs to the growth plate SSCs derived from long bone building on the work of *Chan et al.*¹
- **New Supplementary Figure 6b** demonstrates the comparison of the PF, SAG and COR SSCs to the “periosteal stem cell” as characterized by *Debnath et al.*⁷

Reviewer: I agree with the authors that comparing their scRNA-seq data with other studies may go beyond the scope of the current manuscript albeit sutures were single-cell sequenced (i.e. in Debnath et al., Nature 2018, 562:133-139) and it is of high curiosity to compare. At the same time I disagree with the argument that the space is limited – the current data can be presented much more efficiently without losing information, i.e., Fig. 3d,e,f can be presented as a single panel, Figures S3 and S4 can be combined, S6 presented in text, etc.

Our Answer: We have now added a new analysis comparing the SSC isolated from the cranial suture to previous publications. As requested by the reviewers, we compared the transcriptome of the “periosteal stem cell” as characterized by *Debnath et al.*⁷ with the SSCs derived from each of the posterior frontal (PF), sagittal (SAG), and coronal (COR) cranial sutures at a single cell level. Interestingly, we found that the transcriptome of SSCs derived from the respective PF, SAG and COR shared transcriptional signatures with the “periosteal stem cell” characterized by Debnath and colleagues at a single cell level. These new data are shown in the new **Supplementary Figure 6b.**

We also attempted to compare the transcriptome of the SSCs derived from long bone, as published by *Chan et al*^{1,2} with the SSCs derived from our sutures. Unfortunately, this was not successful due to multiple factors.

- First, the Chan scRNA sequencing data were obtained from a small number of cells. (n=37). At this limited number, most well-validated pre-integration scaling approaches are unstable, and we found that these data failed both anchor transform-based integration as well as traditional scaling (in the absence of significant parameter optimization to the point where the comparison could no longer reasonably be considered unbiased).
- The second limitation of this comparison is that the scRNA sequencing technique described in the study by *Chan et al* was novel at the time but is technically limited in comparison to current methods, including those used in the present study, further compounding the “batch effect” correction problem described above.
- The third limitation is that the *Chan et al* scRNA sequencing data were derived from the parent “SSC population”, which the authors were subsequently able to further separate into SSCs and downstream progenitors (the so-called pre-BCSP), further muddling the above comparison. Thus, in order to address the reviewer’s concerns, we compared the transcriptomes of SSCs isolated from murine long bone (using the methods and immunophenotype as described by *Chan et al.*) with those of suture-derived SSCs. These SSCs were specifically isolated from the long bone growth plate (rather than the entire long bone), as it plays an analogous role to the suture mesenchyme in the lower extremity. These data are now shown in the new **Supplementary Figure 6a**.

Reviewer: I would also suggest adding GSEA analysis and clarifications about Wnt levels mentioned by the authors to the manuscript in order to decrease confusion among the readers about authors’ current and previous works.

Our Answer: We thank the reviewer for these suggestions, which have been incorporated into our revised manuscript. A **new Supplementary Figure 3a** has been added to the manuscript and includes GSEA analysis. Additionally, to decrease confusion and provide clarification of Wnt levels among the 3 sutures at pN3, we have now added **New Supplementary 3a**.

Reviewer: 4. Well, in any of the mentioned experiments it is still possible that Wnt (or Tgfb) acts on osteoblasts or adipocytes or any other cells in the surrounding tissues, which in turn signal to SSCs. That is what I mean under “indirect action via other cell types”. I think this possibility has to be discussed. The mentioned effect of Wnt3a on proliferation of cultured SSCs would provide direct evidence, but either it is not reflected in the manuscript or I missed it.

Our Answer: We apologize for the confusion and agree that this is an important point to address. The revised manuscript now discusses the possibility that the treatment could act through other cell types modulating the SSC niche. Additionally, we now provide *in vitro* evidence of the proliferative effect of Wnt3a on SSCs through a proliferation assay of Wnt3a treated and untreated SSCs from the skull and GP (**Supplementary Figure 7e**). The observation that Wnt3a induces proliferation of SSCs *in vitro* provides direct evidence for the role of cWnt on SSC expansion independent of other potential paracrine signaling transmitted by SSCs-neighborhood cells within the suture.

Reviewer: 5. Thank you for clarification and sorry for being confused between P3 and pN3. I believe other readers may also get confused and it would be good to clarify it more clearly throughout the manuscript. Along the same lines, are the data on Figs. 3b and 3g are normalized or just reflect the total number? It is not clear from the figures or the legends.

Our Answer: We agree that this may have been unnecessarily confusing and have modified the manuscript to better clarify this distinction. We have also incorporated further explanation into the manuscript regarding the distinction between P3 and pN3 as shown on **Page 5**.

For **Figure 3**, SSCs are presented as total number per 500,000 events. This was used for the *in vitro* explant system for normalization and has been clarified in the **updated Figure 3** legend.

Reviewer: 6. I am convinced by the authors' argumentation that the data presented in this manuscript are different from the previous work as the use of differently behaving sutures may provide some new insights. I am not satisfied by the data analysis and interpretation. First, it is clone size not the number of clones, which reflects cell proliferation. The latter reflects efficiency of labeling (if clones stay unbroken). However, the authors presented the number of clones instead (Fig. 4e,I, S7d). Second, for clonal definition the authors say: "Cells that were visually determined to be clones were traced..." I think visual determination of clones being clones is not rigorous enough to be reproduced and requires somewhat more sophisticated definitions, e.g. mathematical.

Our Answer: Thank you for this recommendation. Unfortunately, due to limitations of the Rainbow system, we do not have a free channel to add a nuclear stain that will not overlap with the current fluorophores used in the rainbow construct. Therefore with a ubiquitous promoter such as ActinCre we are unable to accurately quantify the number of cells/clone and can only count the number and measure the size of clones. We have thus acted on the recommendations of the reviewer and have moved the rainbow data to **Supplementary Figures 7d&f** and **8a-c**. In addition, we have also scaled back the conclusions drawn from all Rainbow data in the manuscript.

Reviewer: There is likely a typo in the method, surface area of 3.2m2 sounds quite much.

Our Answer: We agree with the reviewer that this surface area is non-physiologic and have corrected our measurement unit (mm^2) in the revised paper.

Reviewer: 7. The staining of active b-catenin provided in the new Fig. S10 looks very much cytoplasmic. At the same time, upon activation b-catenin translocates to the nucleus and, accordingly, active b-catenin usually looks as a nucleus-localized signal. I think the staining provided is not reflecting b-catenin activity.

Our Answer: Thank you for this comment. For our immunofluorescence experiments we used a well-established monoclonal antibody, which specifically recognizes the signaling isoform of β -catenin, which is dephosphorylated at Ser37 or Thr4, and often referred to as the signaling form of β -catenin, or Active β -Catenin (ABC). This signaling isoform of β -catenin is not susceptible to either ubiquitination or degradation, therefore the employed monoclonal antibody should detect an active form of β -catenin mediating an cWnt signaling. To our knowledge, activated cytoplasmic and nuclear β -catenin cannot be distinguished by any antibody. To address the reviewer's concern of nuclear localization, we implemented the EzColocalization plugin on ImageJ.⁸ The EzColocalization plugin was developed to visualize the localization of signals and measure colocalization in cells, tissues and whole organisms, in an easy to use graphical interface. We implemented EzColocalization to map the cells within the suture mesenchyme in which β -catenin staining was nuclear by measuring the colocalization of DAPI (nuclear stain) and GFP (β -catenin). The accompanying heatmap is shown for the images. High intensity, yellow colors, represent areas of colocalization of DAPI and GFP, representing nuclear B-Catenin. These results are shown in a **new Supplementary Figure 11 a-d right panel.**

References:

- 1 Chan, C. K. *et al.* Identification and specification of the mouse skeletal stem cell. *Cell* **160**, 285-298, doi:10.1016/j.cell.2014.12.002 (2015).
- 2 Gulati, G. S. *et al.* Isolation and functional assessment of mouse skeletal stem cell lineage. *Nat Protoc* **13**, 1294-1309, doi:10.1038/nprot.2018.041 (2018).
- 3 Tevlin, R. *et al.* Pharmacological rescue of diabetic skeletal stem cell niches. *Sci Transl Med* **9**, doi:10.1126/scitranslmed.aag2809 (2017).
- 4 Murphy, M. P. *et al.* Articular cartilage regeneration by activated skeletal stem cells. *Nat Med*, doi:10.1038/s41591-020-1013-2 (2020).
- 5 Ransom, R. C. *et al.* Mechanoresponsive stem cells acquire neural crest fate in jaw regeneration. *Nature* **563**, 514-521, doi:10.1038/s41586-018-0650-9 (2018).
- 6 Jones, R. E. *et al.* Skeletal Stem Cell-Schwann Cell Circuitry in Mandibular Repair. *Cell Rep* **28**, 2757-2766 e2755, doi:10.1016/j.celrep.2019.08.021 (2019).
- 7 Debnath, S. *et al.* Discovery of a periosteal stem cell mediating intramembranous bone formation. *Nature* **562**, 133-139, doi:10.1038/s41586-018-0554-8 (2018).
- 8 Stauffer, W., Sheng, H. & Lim, H. N. EzColocalization: An ImageJ plugin for visualizing and measuring colocalization in cells and organisms. *Scientific reports* **8**, 15764, doi:10.1038/s41598-018-33592-8 (2018).

REVIEWER COMMENTS

Reviewer #3 (Remarks to the Author):

I am happy to see that the authors considered the reviewers' concern more seriously this time. The manuscript is significantly improved and most of my concerns and many of the reviewer #1 were addressed in the revised manuscript.

However, the main concern (both mine and the one of reviewer #1) is still there – are we discussing the skeletal stem cells (SSCs)?

The SSCs were defined as AlphaV+CD200+ cells obtained from the growth plate area and capable to produce all other lineages of skeletal progenitors in in vivo renal capsule assay of bone formation (Chan et al., Cell 2015).

In their rebuttal letter the authors say that sutures-derived SSCs are clustering together with those obtained from the growth plate (GP). In fact, only a small fraction of GP-SSC clusters together with sutures-derived SSCs (cluster #0, Fig S6), whereas most of GP-SSCs cluster into a separate cluster (cluster #3). This observation indicates that they are substantially different. Thus, it remains unclear what skeletogenic potential suture-derived SSCs have. The authors addressed the stemness of suture-derived SSCs in in vitro assays: colony formation and 3-lineage differentiation (Fig. S2). However, I think the stemness of suture-derived SSCs is not sufficiently proven without in vivo experiments (i.e., renal capsule assay) and demonstration that they are capable to generate all other lineages and progenitors.

It has to be taken into account that Chan et al (Cell 2015) has isolated SSCs from the neonatal growth plate area. This area contains chondrocytes and perichondrium/periosteum and it is logical to assume that GP-SSCs are either chondrocytes or perichondral/periosteal cells or a combination. Since there are no chondrocytes in the sutures it is likely that some of suture-derived SSCs may have similarities with the putative periosteal fraction of those GP-SSCs (cluster #0) whereas another one (cluster #3) may well represent growth plate chondrocytes.

This assumption is further supported by the substantial transcriptional similarities between sutures-derived SSCs and periosteal stem cells (Debnath et al, Nature 2018) (Figure S6b). Based on this, there is a good chance that the authors are dealing here with periosteal stem cells. In relation to this, mapping of the identified subpopulations on tissue-sections would be very helpful.

Additionally, transcription profiling indicates that there are several different sub-populations within the FACS-isolated fraction of suture-derived SSCs. It is unclear if they are all skeletal stem cells or just some sub-populations? What is the hierarchy within these subpopulations?

Taking all these considerations into account, this heterogeneous population of sutures-derived cells should be either renamed (i.e., Skeletal Stem and Progenitor Cells) throughout the manuscript or rigorous in vivo evidences of their stemness provided.

Response to Reviewer #3:

Reviewer #3 (Remarks to the Author):

1. **Reviewer:** I am happy to see that the authors considered the reviewers' concern more seriously this time. The manuscript is significantly improved and most of my concerns and many of the reviewer #1 were addressed in the revised manuscript.

Our Response: We thank the Reviewer for their ongoing contributions to our manuscript, and in particular for agreeing to take on the additional responsibilities resulting from the absence of Reviewer 1. Moreover, we are pleased to see that they found our revised paper to be significantly improved.

2. **Reviewer:** However, the main concern (both mine and the one of reviewer #1) is still there – are we discussing the skeletal stem cells (SSCs)? The SSCs were defined as AlphaV+CD200+ cells obtained from the growth plate area and capable to produce all other lineages of skeletal progenitors in in vivo renal capsule assay of bone formation (Chan et al., Cell 2015).

Our Response: To clarify, this study uses the same strategy for isolation of SSCs that was originally developed by our group in 2015 (Chan, C.K. & Longaker M.T. *et al. Identification and specification of the mouse skeletal stem cell. Cell* 160, 285-298, 2015); and since then, further employed to isolate SSCs from other bones including facial skeleton bones, such as the mandible. In these studies, the characterization of the newly isolated SSCs has been performed using *in vitro* lineage specification and CFU capacity assays. These are well established and acknowledged assays in the field and currently used by other investigators to define a skeletal stem cell resident in the cranial sutures. (Please see: Zhao H. *et al., Nat Cell Biol*, 2015; Maruyama T. *et al., Nat Cell Biol*, 2016; Wilk K. *et al., Stem Cells Reports*, 2017). Please also see our responses to comments #5 and #8 below.

3. **Reviewer:** In their rebuttal letter the authors say that sutures-derived SSCs are clustering together with those obtained from the growth plate (GP). In fact, only a small fraction of GP-SSC clusters together with sutures-derived SSCs (cluster #0, Fig S6), whereas most of GP-SSCs cluster into a separate cluster (cluster #3). This observation indicates that they are substantially different. Thus, it remains unclear what skeletogenic potential suture-derived SSCs have.

Our Response: We agree with the Reviewer that our discussion of suture-derived SSCs clustering with a small fraction of GP-SSCs may have been somewhat misleading and overly suggestive of functional distinctions among cell populations isolated from the sutures versus the long bone/growth plate. We would not expect the transcriptomes of GP-SSCs and three Suture-SSCs to share identical transcriptional programs given the significant differences in embryonic origin and load bearing status. We believe that this transcriptional heterogeneity is likely reflective of these differences in the niche environment in which GP-SSCs and Suture-SSCs reside. However, we agree with the Reviewer that the UMAP embedding provided in the previous Supplementary Figure 6a was potentially misleading to that effect, and it has been removed to avoid confusion. Regarding the skeletogenic potential of Suture-derived

SSCs the lineage specification assays indicate that Suture-SSCs have both, osteo- and chondrogenic potential.

4. **Reviewer:** The authors addressed the stemness of suture-derived SSCs in in vitro assays: colony formation and 3-lineage differentiation (Fig. S2). However, I think the stemness of suture-derived SSCs is not sufficiently proven without in vivo experiments (i.e., renal capsule assay) and demonstration that they are capable to generate all other lineages and progenitors.

Our Response: This is addressed above in our response to the Reviewer comment #2.

5. **Reviewer:** It has to be taken into account that Chan et al (Cell 2015) has isolated SSCs from the neonatal growth plate area. This area contains chondrocytes and perichondrium/periosteum and it is logical to assume that GP-SSCs are either chondrocytes or perichondral/periosteal cells or a combination.

Our Response: We thank the Reviewer for their comments. The growth plate zone contains multiple cell types included chondrocytes and perichondral/periosteal cells. However, it is improbable that the SSCs isolated from the GP are chondrocytes, considering that *Chan & Longaker et al.*, have demonstrated that these are Skeletal Stem Cells. Regarding perichondral/periosteal cells, our isolation method from both the GP and Sutures includes periosteal tissue, and of course it cannot be ruled out that a fraction of our isolated SSCs are derived either from perichondrium or periosteum.

The *in vitro* evidence provided in the current paper is in line with the most recent four publications (three of which are Nature family journals). (See: Appendix 1) While we agree that a fundamental recapitulation of the original 2015 *in vivo* study set would be ideal, this is not practical in the context of the relative harvest size for the tissues evaluated here, where 60 mouse cranial sutures (PF, SAG or COR) can provide approximately 750-2000 cells depending on suture. To isolate ~20,000 double sorted, pure GFP⁺ SSCs needed to recapitulate the original 2015 *in vivo* study, would require the coordinated breeding of ~500+ animals while significantly fewer mouse's long bones can provide sufficient cells for this assay.

In the absence of said experiments, we agree with the Reviewer that selective language modulation would be appropriate, which has been done in the revised manuscript, but nevertheless feel confident in referring to these cells as SSCs while providing proper context.

Furthermore, in response to the Reviewer's concern regarding chondrocytes, we also conducted an additional meta-analysis of our data in conjunction with the recently published *Mizuhashi et al.*, *JBMR 2019* study examining Col2a1-creER;R26R^{tdTomato} chondrocytes isolated from pN2 mouse femoral growth plates (See: Figure 1 below). After batch correction using Seurat's anchor-based cross-platform scRNA-seq integration technique, we found that our Suture- and GP-SSCs exhibited only minimal overlap with this comprehensive dataset, suggesting that our cells, both GP-SSCs and Suture-SSCs, are unlikely to include a significant chondrocyte subset.

6. **Reviewer:** Since there are no chondrocytes in the sutures it is likely that some of suture-derived SSCs –may have similarities with the putative periosteal fraction of those GP-SSCs (cluster #0) whereas another one (cluster #3) may well represent growth plate chondrocytes.

Our Response: We thank the reviewer for this comment and would like to provide some additional clarification. The PF suture fuses by endochondral ossification. As early as pN8 chondrocytes (and strong alcian blue staining for proteoglycans, as well expression of chondrogenic markers) are present in the PF suture mesenchyme (*Sahar et al., Developmental Biology, 2005 PMID:15882577, Behr et al., Developmental Biology 2010*

PMID:20547147). Therefore, similar to the long bones, the PF suture goes through an endochondral developmental program and explains the transcriptomic similarities observed between GP-SSCs and PF-SSCs.

From our secondary scRNA-seq analysis (previously included in Supplemental Figure 6a, i-ii), we had posited that cluster 3 with SSCs derived from both the PF and GP might be indicative of transcriptional priming for an endochondral ossification fate of these SSCs – unique to the PF suture and growth plate. We understand how this discussion might have prompted confusion and have removed this in our revised manuscript.

7. **Reviewer:** This assumption is further supported by the substantial transcriptional similarities between sutures-derived SSCs and periosteal stem cells (Debnath et al, Nature 2018) (Figure S6b). Based on this, there is a good chance that the authors are dealing here with periosteal stem cells. In relation to this, mapping of the identified subpopulations on tissue-sections would be very helpful.

Our Response: We agree with the reviewer that these transcriptional similarities/differences can potentially muddle the underlying biological differences. To be clear, we do not believe we are isolating Periosteal Stem Cells (PSCs), as described by *Debnath et al., Nature, 2018*. Debnath defined PSCs as Cathepsin K-GFP⁺/CD200⁺/CD51_{dim}, whereas the mouse Skeletal Stem Cell (mSSC) is defined as CD200⁺/CD51⁺. This difference in CD51 expression would make it unlikely that we are isolating PSCs under the mSSC marker profile.

The intent of our label-transfer mapping of Debnath periosteal stem cell data onto our current scRNA-seq data was to highlight that some SSCs exhibited comparatively more or comparatively less similar transcriptional programs to those of PSCs. However, we should have better clarified that the scale of the anchor-mapping metric was fixed at 0→1 for the floor and ceiling of the respective similarities, such that the absolute values mean nothing, but rather the relative distribution is a reflection of the comparative similarity in transcriptional program.

To better represent the relationship between cells in these two datasets, we have provided a replacement figure panel that employs Seurat's anchor-based cross-platform scRNA-seq integration technique to project all cells into the same manifold space. From this analysis, we see clear separation between cells from *Debnath et al.* versus our, Suture-derived SSCs data set, with minimal *Menon et al.* to *Debnath et al.* promiscuity. These data have been incorporated into the new Supplementary Figure 6 a, ii.

Moreover, we would not expect the transcriptomes of GP-SSCs and three Suture-SSCs to share identical transcriptional programs given the significant differences in embryonic origin and load bearing status. We believe that this transcriptional heterogeneity is likely reflective of these differences in the niche environment in which GP-SSC and Suture-SSCs reside. We also agree with the Reviewer that confirming these differences would require additional assays in line with spatial mapping (Visium, Codex, etc). However, at present, those technologies are not compatible with interrogation of rare cell populations such as SSCs, whose signal would be washed out at the current resolution of 5-20 cells per amplification

“spot”. Given these limitations, we agree with the Reviewer’s concerns and have modified our verbiage accordingly.

8. **Reviewer:** Additionally, transcription profiling indicates that there are several different sub-populations within the FACS-isolated fraction of suture-derived SSCs. It is unclear if they are all skeletal stem cells or just some sub-populations?

Our Response: As discussed above, we agree with the Reviewer that our earlier usage of the terms “subpopulations”, “subsets”, “subgroups”, and “clusters” was unnecessarily confusing and has been refined in the updated manuscript. Although our gene expression profiling identified several transcriptionally-defined subgroups of suture-derived SSCs, we are not trying to assert that these are necessarily functionally-distinct subgroups (a higher bar for use of the term “subpopulation”).

Debate regarding the transcriptional definition of a “homogeneous” cell population (predicted to have no functionally-distinct subpopulations) or a heterogeneous population (predicted to have multiple functionally-distinct subpopulations) has long preceded the development of scRNA-seq technologies, and our group wrote one of the earliest articles attempting to address this in 2011 (*Glotzbach et al., PLoS One, 2011 PMID: 21731674*). Our approach then was far from perfect, and other groups have since advanced this field further and more elegantly (*Jiang et al., Genome Biol, 2016 PMID: 27368803, Liu et al., Nat Commun, 2020 PMID: 32572028*). However, even at present there is no real substitute for actually isolating putative ‘subpopulations’ for functional profiling. This is not our intent

now, nor was it our intent previously, to have proposed these transcriptionally-defined cell subgroups as ‘subpopulations’, although we acknowledge that some of our earlier verbiage may have suggested that.

However, we agree with the Reviewer that the question of SSC purity is important and that our manuscript would benefit from addressing this. We have now included additional assessments of transcriptional purity using the recently-developed ROGUE software packages (*Liu et al., Nat Commun, 2020* PMID: 32572028) and provided the results in a new Supplementary Figure 6b. These findings demonstrate similar levels of ‘homogeneity’ for our suture populations in comparison to the original SSCs prospectively isolated by our group in 2015 (*Chan & Longaker et al. Cell 2015*) and those transcriptionally-defined by Debnath (*Debnath et al., Nature, 2018*), providing further support that we are in fact isolating comparatively pure SSC populations.

9. **Reviewer:** What is the hierarchy within these subpopulations?

Our Response: Given the significant modifications to the revised manuscript’s handling of heterogeneity and putative “cell sub-populations”, in accordance with the Reviewer and Editor’s helpful suggestions above, we now feel that a discussion of candidate hierarchies among transcriptionally-defined subsets is no longer central to the paper’s narrative and may itself be misleading or overly-speculative relative to the study’s findings.

10. **Reviewer:** Taking all these considerations into account, this heterogeneous population of sutures-derived cells should be either renamed (i.e., Skeletal Stem and Progenitor Cells) throughout the manuscript or rigorous *in vivo* evidences of their stemness provided.

Our Response: We again thank the reviewer for highlighting these potential incongruities within our nomenclature, which we agree needs to be more precise in order to appropriately convey the significance of our findings in this study. Based on the Editors recommendation to provide additional *in vivo* evidence or provide the clarifications and modification to our manuscript described above in response to the reviewer’s very helpful suggestions, we believe that concerns regarding the purity versus heterogeneity of the SSCs isolated from cranial sutures in this work have been largely allayed. In fact, our Suture-SSC population ROGUE purity score was comparable to *Debnath et al.*, and *Chan & Longaker et al.*, as seen in **Supplementary Figure 6b**. These points are addressed in our revised manuscript, which we believe provides appropriate context for our qualified application of the “Skeletal Stem Cell” designation, in keeping with the body of prior literature in this field. We again thank the reviewer for their persistent efforts to improve this article.

Please find these following changes, addressing all of the above concerns in red font. **Abstract: MS page 2, lines 51-53, 59, Results: MS page 6-7, lines 159, 161-167, Discussion: MS page 13, lines 321-326.** Red font only represents new changes in our revised manuscript. Notably, many sentences from these sections have been removed in their entirety that were thought to be too strong and/or unclear.

Appendix 1

1. Chan, C. K. *et al.* Identification and specification of the mouse skeletal stem cell. *Cell* **160**, 285-298, doi:10.1016/j.cell.2014.12.002 (2015)
2. Tevlin, R. *et al.* Pharmacological rescue of diabetic skeletal stem cell niches. *Sci Transl Med* **9**, doi:10.1126/scitranslmed.aag2809 (2017).
3. Gulati, G. S. *et al.* Isolation and functional assessment of mouse skeletal stem cell lineage. *Nat Protoc* **13**, 1294-1309, doi:10.1038/nprot.2018.041 (2018).
4. Ransom, R. C. *et al.* Mechanoresponsive stem cells acquire neural crest fate in jaw regeneration. *Nature* **563**, 514-521, doi:10.1038/s41586-018-0650-9 (2018).
5. Jones, R. E. *et al.* Skeletal Stem Cell-Schwann Cell Circuitry in Mandibular Repair. *Cell Rep* **28**, 2757-2766 e2755, doi:10.1016/j.celrep.2019.08.021 (2019).
6. Murphy, M. P. *et al.* Articular cartilage regeneration by activated skeletal stem cells. *Nat Med*, doi:10.1038/s41591-020-1013-2 (2020).

REVIEWERS' COMMENTS

Reviewer #3 (Remarks to the Author):

Dear editor and authors,

My conceptual concern is not addressed by the authors – I do not see sufficient evidence to claim the cells, characterized by the authors in the sutures, as skeletal stem cells. It seems we have principal differences with the authors on this issue, which cannot be resolved in the current discussion. To justify their point of view, the authors refer to their own publications, which makes such a justification bias. So, I am insisting that there is no strong ground to do approximation that if two markers (AlphaV and CD200) were used to identify stem cells in the growth plate, they will automatically identify stem cells elsewhere. Thus, either direct evidence needs to be provided, or stem cells must be renamed to “potential stem cells” or “stem and progenitor cell population”.

Thank you again for giving me an opportunity to participate in the revision process. I do not wish to review this manuscript further.

Best regards,

comment to the authors: ROGUE score should be applied to the clusters. If it shows the purity of the entire population being high (Fig. S6b) - this would imply that clustering is rather artificial.